# Optimal Control under Multiplicative and Internal Noise with Model Mismatch

## Abstract

Natural agents interact with their environment through noisy and continuous sensorimotor loops. Stochastic optimal control provides a principled framework for this problem, but existing analytical solutions are restricted to linear dynamics with Gaussian observations and additive noise. They cannot address scenarios with multiplicative noise in control or observations, and with internal noise affecting estimation – features central to biological and robotic systems. We provide a provably convergent algorithm that computes fixed-point controller–filter solutions for linear dynamics with quadratic costs under multiplicative and internal noise. Our method overcomes the limitations of prior analytical approaches and improves the efficiency of state-of-the-art gradient-based methods by more than three orders of magnitude in realistic tasks. Importantly, it also optimizes internal dynamics, relaxing the classical assumption that internal models must match external dynamics. Allowing such model mismatch yields substantially better performance under internal noise. In sum, we provide the first full solution to stochastic optimal linear control under multiplicative and internal noise, covering both matched and mismatched internal models.

## 1 Introduction

Understanding the computational mechanisms that govern the sensorimotor system in humans and other animals is a long-standing goal in systems and computational neuroscience (Wolpert et al., 1995; Shadmehr & Krakauer, 2008; Franklin & Wolpert, 2011; Todorov, 2004). Yet, developing formal and mathematically tractable models that accurately capture these mechanisms remains an open problem, with far-reaching implications for fields such as artificial intelligence and robotics. In this context, stochastic optimal control theory provides a powerful mathematical framework for explaining behavior in terms of optimality principles, accounting for uncertainty and variability inherent in biological systems (Todorov & Jordan, 2002; Todorov, 2005; Straub & Rothkopf, 2022; Schultheis et al., 2021; Faisal et al., 2008). The seminal work in Todorov (2005) extended the classic Linear-Quadratic-Additive-Gaussian – LQAG – framework (usually referred to as Linear-Quadratic-Gaussian – LQG – problem (Davis, 2013)) to incorporate a more biologically realistic noise model of the sensorimotor system. This includes control-dependent noise (Schmidt et al., 1979; Todorov, 2002), signal-dependent sensory feedback noise (Todorov & Jordan, 2002; Harris & Wolpert, 1998), and internal neural noise (Faisal et al., 2008; Moreno-Bote et al., 2014; Churchland et al., 2006) – all of which are essential for reproducing key signatures of human motor behavior (Todorov, 2005; Flash & Hogan, 1985; Harris & Wolpert, 1998; Todorov, 2002; Schmidt et al., 1979). However, explaining behavior through optimal control requires first obtaining optimal solutions to the underlying problem (Todorov, 2005; Schultheis et al., 2021).

The study of Todorov (2005) provided the first analytically-derived algorithm for optimal linear control under multiplicative and internal noise. Despite its wide applicability (Schultheis et al., 2021; Straub & Rothkopf, 2022; Sensinger & Dosen, 2020; Liu & Todorov, 2007; Izawa et al., 2008; Takei et al., 2021; Shanechi et al., 2013), Damiani et al. (2024) demonstrated that this solution fails to yield truly optimal results in the presence of internal noise, due to the incorrect assumption of unbiased estimators and its connection with the orthogonality principle (Appendix A.1). More recent theoretical work has continued to assume unbiased estimation in extended applications, including iterative LQG (iLQG) and differential dynamic programming (DDP) (Li & Todorov, 2007). To address this limitation, Damiani et al. (2024) introduced a numerical gradient-based algorithm that achieves op-

timal performance, in terms of cost-minimization, under multiplicative and internal noise, albeit at high computational cost, making it impractical for inverse optimal control applications. They also proposed an analytical counterpart, the FPOMP algorithm, which solves the problem in the one-dimensional case and, in higher dimensions, only under additive noise, due to the increased mathematical complexity of the full setting. Consequently, no previous work provides a general analytical solution or formal convergence guarantees.

In this work, we derive an algorithm that fully solves the stochastic control problem of Todorov (2005); our algorithm exploits coordinate descent, and we prove its monotonic improvement and convergence to a critical point (Appendix A.2). This overcomes prior analytical limitations and, unlike the state-of-the-art numerical methods, yields an analytically-derived algorithm for the full problem with speedups of more than three orders of magnitude in realistic tasks. Our framework thus provides both a conceptual advance and a major efficiency gain over existing approaches.

A further limitation of current theoretical work on stochastic optimal control is the reliance on two core assumptions: (1) a strict separation between estimation and control, and (2) the matched-dynamics assumption, i.e., that the internal model used for estimation and control perfectly matches the dynamics of the external environment. These limitations underlie both Todorov (2005) and Damiani et al. (2024), where noisy sensory feedback is first processed by a Kalman filter to produce a state estimate – based on the same forward model of the environment – which then guides linear control actions. Within the classical LQAG problem, this methodology is mathematically justified by the separation principle (Davis, 2013). However, once multiplicative and internal noise are included, the separation principle no longer holds, making estimation and control inherently coupled (Todorov, 2005). Moreover, the assumption that the agent's internal model exactly matches the external dynamics strongly limits the realism of this approach, overlooking a substantial body of research emphasizing the role of internal models in motor control (Wolpert et al., 1995; Shadmehr et al., 2010; Körding & Wolpert, 2004; Kawato, 1999; Golub et al., 2015).

Our second main contribution is to relax these assumptions by considering the more general case where the internal dynamics – used by the agent to process sensory stimuli and generate motor outputs – need not match the dynamics of the external world and must themselves be optimized (Sec. 4). We refer to the classical case as Model Match (M-Match), and to our extension as Model Mismatch (M-Mis). We extend the algorithm developed for the M-Match case (Appendix A.2.2) to this scenario, providing an analytical solution for mismatched internal models. In Sec. 5, we demonstrate that this additional flexibility leads to improved solutions relative to M-Match, particularly in the presence of internal noise. Finally, we illustrate the generality of our framework by applying it to the steering of linear neural populations, which connects directly to computational principles underlying reservoir computing (Jaeger & Haas, 2004; Maass et al., 2002) and, more broadly, to recurrent neural network models that generate task-relevant outputs (Sussillo & Abbott, 2009).

## 2 STOCHASTIC LINEAR OPTIMAL CONTROL: PROBLEM FORMULATION

We first review the standard Linear-Quadratic-Additive-Gaussian (LQAG) problem, then extend the noise model, following Todorov (2005), to include multiplicative observation, control noise, and internal noise, yielding the Linear-Quadratic-Multiplicative-Internal (LQMI) formulation. In both LQAG and LQMI, internal and state dynamics are matched; the more general mismatched case is discussed in Sec. 4.

### 2.1 STOCHASTIC OPTIMAL CONTROL UNDER MULTIPLICATIVE AND INTERNAL NOISE

In the standard LQAG formulation, an agent receives noisy observations $y_t \in \mathbb{R}^k$ ($t = 0, 1, ..., T$) from a state variable $x_t \in \mathbb{R}^m$,

$$y_t = Hx_t + \omega_t , \tag{1}$$

where $H \in \mathbb{R}^{k \times m}$ is the observation matrix and $\omega_t \in \mathbb{R}^k$ is a zero-mean noise with covariance $\Sigma_\omega$. The control problem consists in finding the optimal control signal $u_t(y_{t-1}, ..., y_0) \in \mathbb{R}^p$ that steers the stochastic linear dynamical system

$$x_{t+1} = Ax_t + Bu_t + \xi_t , \tag{2}$$

so as to minimize the expected cumulative quadratic cost

$$C = \sum_{t=0}^{T} \mathbb{E}\left[ x_t^\top Q_t x_t + u_t^\top R_t u_t \right] . \tag{3}$$

The dynamics of the state variable, Eq. 2, is assumed to be linear in state and control with matrices $A \in \mathbb{R}^{m \times m}$ and $B \in \mathbb{R}^{m \times p}$ and corrupted by zero-mean noise $\xi_t \in \mathbb{R}^m$ with covariance $\Sigma_\xi$. All noises are uncorrelated in time and are not required to be Gaussian. We observe that time-dependent matrices in the dynamics or noise can be trivially incorporated. The initial condition of the dynamics is $x_0$, usually drawn from a Gaussian distribution. The control signal $u_t(y_{t-1}, ..., y_0)$ at time $t$ is allowed to depend only on previous observations, but not on the state nor on future observations to enforce partial observability and causality, respectively. The expectation in Eq. 3 is over the realizations of the noise and the initial conditions. Each term in the sum is the expected instantaneous cost at time $t$. The total expected cost $C$ penalizes large control signals – reflecting energetic or metabolic constraints – as well as deviations from desired trajectories or targets, through the symmetric positive semidefinite matrices $R_t \in \mathbb{R}^{p \times p}$, $R_t \geq 0$, and $Q_t \in \mathbb{R}^{m \times m}$, $Q_t \geq 0$, respectively.

The LQAG problem admits an analytical solution (Davis, 2013), which is the combination of a linear Kalman filter, providing optimal estimates $\hat{x}_t \equiv z_t$ of the partially observable state $x_t$, and a linear feedback controller defined by $u_t = L_t z_t$, which are computed independently, without mathematical dependence between control and filter gains – the so-called separation principle (Davis, 2013). We return to this point in Appendix A.4.7, where we empirically examine the consequences of relying on this principle. The internal variable becomes a state estimate evolving according to

$$z_{t+1} = A z_t + B u_t + K_t(y_t - H z_t) , \tag{4}$$

where $K_t \in \mathbb{R}^{m \times k}$ is the Kalman gain at time $t$. Solving the optimal control problem therefore consists in computing both the optimal filter and control gains, respectively $K_t$ and $L_t \in \mathbb{R}^{p \times m}$, under the constraint that the internal dynamics follow the same forward dynamics as the state variable (matrices $A$ and $B$; see Appendix A.2.3 for the well-known solutions).

While the analytical tractability of the LQAG framework is a key advantage, it comes at the expense of reduced biological realism. In particular, the noise model does not account for multiplicative noise, also neglecting internal sources of variability (Faisal et al., 2008; Moreno-Bote et al., 2014; Churchland et al., 2006; Franklin & Wolpert, 2011). To consider a more general and realistic noise model, following Todorov (2005), we first introduce multiplicative noise – both control-dependent and observational – into the system and observation dynamics in Eqs. 1,2. This leads to the modified equations

$$x_{t+1} = A x_t + B u_t + \xi_t + \sum_i \varepsilon_t^i C_i u_t \tag{5}$$

$$y_t = H x_t + \omega_t + \sum_i \rho_t^i D_i x_t . \tag{6}$$

In this framework, executing a control input $u_t$ adds noise whose magnitude scales with the input itself (Sutton & Sykes, 1967; Schmidt et al., 1979; Harris & Wolpert, 1998), Eq. 5. Conversely, sensing the partially observable state $x_t$ introduces sensory noise whose magnitude scales with the state itself (Burbeck & Yap, 1990; Whitaker & Latham, 1997), Eq. 6. The matrices $C_i \in \mathbb{R}^{m \times p}$ and $D_i \in \mathbb{R}^{k \times m}$ define fixed gain patterns for the multiplicative noise components, while $\varepsilon_t \in \mathbb{R}^c$ and $\rho_t \in \mathbb{R}^d$ represent zero-mean noise vectors, each with identity covariance, $\Sigma_\varepsilon = \mathbb{I}_{c \times c}$ and $\Sigma_\rho = \mathbb{I}_{d \times d}$. As in the LQAG problem, control and observation noises are assumed to be mutually independent, and also independent from both the additive and multiplicative noise components. Finding the optimal control signal $u_t(y_{t-1}, ..., y_0)$ that minimizes the cost in Eq. 3 with system and observation dynamics given by Eqs. 5,6 is a challenging problem with no known solutions, even in the case of Gaussian noise. In particular, no sufficient statistic, analogous to $\hat{x}_t \equiv z_t$, is known that would allow for a Kalman filter-like recursion. Following Todorov (2005), we assume that the control signal $u_t$ can only linearly depend on the estimate $z_t \in \mathbb{R}^m$, that is, $u_t = L_t z_t$, with $L_t \in \mathbb{R}^{p \times m}$, and that the state estimate obeys the *matched* dynamical equation

$$z_{t+1} = A z_t + B u_t + K_t(y_t - H z_t) + \eta_t , \quad u_t = L_t z_t , \tag{7}$$

with the same terminology as in Eq. 4, but where we have introduced an internal additive noise term $\eta_t \in \mathbb{R}^m$, with zero mean and covariance $\Sigma_\eta$. The internal noise may represent internal neural variability (Faisal et al., 2008; Moreno-Bote et al., 2014; Churchland et al., 2006; Franklin & Wolpert,

2011) or flaws in the filtering process itself, and it is introduced here to obtain a more realistic and general model (Todorov, 2005). Taken together, incorporating multiplicative and internal noise with the assumptions of a linear Kalman filter for state estimation and a linear control policy based on an internal estimate whose forward dynamics match those of the state (matrices $A$ and $B$) gives rise to the more general Linear–Quadratic–Multiplicative–Internal (LQMI) problem. Solving this problem involves determining the optimal control gains $L_{0,...,T}$ and filter gains $K_{0,...,T}$ that minimize the quadratic cost function in Eq. 3 under the system, observation and estimate dynamics in Eqs. 5,6,7.

## 3 SOLVING THE LQMI PROBLEM

We derive an algorithm that is guaranteed to converge to a critical point of the cost function in Eq. 3, under the dynamics in Eqs. 5, 6, and 7. Importantly, this guarantee holds even though the problem is non-convex: indeed, the global LQAG problem in the fully observable setting – which is a special case of our LQMI formulation – is itself non-convex (Fazel et al., 2018). Our algorithm yields improved pairs of control and filter gains, fully solving the LQMI problem. Complete derivations and pseudocode appear in Appendices A.2 and A.3.1 – Algorithm 1. Below, we summarize the main ideas and corresponding equations.

Assuming a linear control signal $u_t = L_t z_t$, we first rewrite the cost function in Eq. 3 as $C = \sum_{t=0}^{T} \left( \text{tr}(Q_t S_t^{xx}) + \text{tr}(L_t^\top R_t L_t S_t^{zz}) \right)$, where we introduce the 2nd-order moment matrices $S_t^{xx} = \int dx dz \, p_t(x,z) xx^\top$, $S_t^{zz} = \int dx dz \, p_t(x,z) zz^\top$, and $S_t^{xz} = \int dx dz \, p_t(x,z) xz^\top$, with $p_t(x,z)$ being the joint distribution of $x$ and $z$ at time $t$ generated by previous control and filter gains and averaging over noises and initial conditions following the distribution $p_0(x,z)$. To find the conditions for extrema on the control $L_{0,...,T}$ and filter $K_{0,...,T}$ gains we add Lagrange multipliers and define the new objective

$$C_\mathcal{L} = \sum_{t=0}^{T} \left( \text{tr}(Q_t S_t^{xx}) + \text{tr}(L_t^\top R_t L_t S_t^{zz}) \right) - \sum_{t=1}^{T+1} \left( \text{tr}(\Lambda_t G_t^{xx}) + \text{tr}(\Omega_t G_t^{zz}) + \text{tr}(\Gamma_t G_t^{xz}) \right) , \quad (8)$$

where $\Lambda_t$, $\Omega_t$ and $\Gamma_t$ are $\mathbb{R}^{m \times m}$ matrices of Lagrange multipliers (see Eqs. 16 in Appendix A.2). The constraints $G_t^{xx} = G_t^{zz} = G_t^{xz} = 0$ are given by the temporal evolution of the 2nd-order moment matrices $S_t^{xx}$, $S_t^{zz}$ and $S_t^{xz}$, respectively, between two consecutive time steps $t$ and $t+1$, obtained from Eqs. 5,6,7 (see Appendices A.2 and A.2.4 for details). A crucial step in solving the LQMI problem is to observe that the total cost in Eq. 3 admits the decomposition

$$C = C_{<t} + C_t \quad (9)$$

for any $t$, where $C_{<t} = \sum_{\tau=0}^{t-1} \text{tr}(Q_\tau S_\tau^{xx} + L_\tau^\top R_\tau L_\tau S_\tau^{zz})$ and the cost-to-go from time $t$ onward is defined as $C_t = \text{tr}(\Lambda_t S_t^{xx} + \Omega_t S_t^{zz} + \Gamma_t S_t^{xz}) + \gamma_t$. Thus, $C_t$ depends on the Lagrange multipliers (given by Eqs. 16) and on the additional scalar parameter $\gamma_t$ (following Eq. 19). Given this structure, and since $L_t$ affects only the expected cost from time $t$ onward, we can locally optimize $L_t$ at each time step – as shown in Appendix A.2 – as

$$L_t^* = \arg\min_{L_t} C_t = E_t^{-1} \left( F_t S_t^{xz} (S_t^{zz})^{-1} + J_t \right) , \quad (10)$$

(with matrices $E_t$, $F_t$ and $J_t$ defined in Appendix A.2.6) while keeping the rest of gains fixed, i.e., $L_{0,...,t-1,t+1,...,T}$ and $K_{0,...,T}$ are held constant.

For each local subproblem (i.e., optimizing $L_t$ with all other gains held fixed), a global minimum for $L_t$ exists because $C_t$ is convex. As shown in Appendix A.2, starting from a set of gains $L^{(n)} \equiv L_{0,...,T}^{(n)}$ and $K^{(n)} \equiv K_{0,...,T}^{(n)}$, we can update the control gains by optimizing $L_t$ sequentially from $t = 0$ to $T$ using Eq. 10. This yields the new set of gains $L^{(n+1)}$, after which the Lagrange multipliers are recomputed backward in time using Eqs. 16. Because of the local optimization, we obtain that the cost is non-increasing, that is, $C(L^{(n+1)}, K^{(n)}) \leq C(L^{(n)}, K^{(n)})$. A full forward pass that sequentially optimizes the control gains, followed by a full backward pass of the multipliers is referred to as *control pass*. An analogous procedure can be applied to optimize $K_t$ (Eq. 26 in Appendix A.2), defining the corresponding *filter pass*.

In conclusion, starting from arbitrary $L^{(0)}$ and $K^{(0)}$ and distribution of initial conditions $p_0(x,z)$, we can alternate the control and filter passes, so that $C(L^{(0)}, K^{(0)}) \geq C(L^{(1)}, K^{(0)}) \geq$

$C(L^{(1)}, K^{(1)}) \geq ... \geq C(L^{(n+1)}, K^{(n)}) \geq C(L^{(n+1)}, K^{(n+1)}) \geq ... \geq C_{min} \geq 0$. Since the series is non-negative, it converges to a total cost no higher than the initial one with optimal filters $L^* = L^{(\infty)}$ and $K^* = K^{(\infty)}$. In summary, in our *coordinate descent* algorithm, each block update solves a convex quadratic subproblem exactly, which guarantees that the total cost decreases monotonically and therefore converges. Because the converged solution is also a stationary point of the Lagrangian, Eq. 8, it corresponds to a fixed point of the original cost function (see Appendix A.2). Following this reasoning, we prove

**Theorem 1.** *Starting with arbitrary $L^{(0)}$ and $K^0$ and distribution of initial conditions $p_0(x, z)$, the coordinate descent algorithm defined by iterating in alternation control and filter passes converges to an improved pair of control and filter gains $L^*$ and $K^*$. The improved pair corresponds to a critical point of the cost function in Eq. 3.*

We first remark that the Lagrange equations may admit multiple solutions. In practice, our algorithm converges to different critical points depending on the initialization, but when initializing the control and filter matrices trying to impose the orthogonality principle and then freely running the algorithm, the best critical point is found, empirically. Secondly, it is worth mentioning that in the derivation of our algorithm we do not assume the orthogonality principle (OP: $S_t^{xz} = S_t^{zz}$ for all $t$, equivalent to $\mathbb{E}[(x_t - z_t)z_t^\top] = 0$), which is shown (Sec. 3.1 and Appendix A.1) to be violated in the general case (specifically, whenever there is internal noise). Thirdly, we have not assumed any parametric form for initial distribution $p_0(x, z)$. Finally, as shown in Eqs. 16, 23, 26, and 32, only the first and second noise moments enter the moment propagation and optimality conditions. No further assumptions are required beyond finite second moments, so the method applies to any noise distribution with finite covariance. In AppendixA.4.8 we validate this empirically using non-Gaussian noise.

### 3.1 ORTHOGONALITY PRINCIPLE YIELDS A CRITICAL POINT AT ZERO INTERNAL NOISE

**Theorem 2.** *Take initial condition $p_0(x, z)$ such that $S_0^{zz} = S_0^{xz}$. A solution to the Lagrange equations 13,14,15,16 is given by the orthogonality principle $S_t^{zz} = S_t^{xz}$ for $t = 1, ..., T$, iff internal noise is zero, that is, $\Sigma_\eta = 0$. The solution corresponds to a critical point of the cost in Eq. 8*

See the proof in Appendix A.2.7. We note that OP is implied by the unbiasedness condition (Appendix A.1), but not vice versa. While unbiasedness was empirically shown to be violated in Damiani et al. (2024), we have now formally demonstrated that only the weaker OP condition is required to obtain a critical point of the cost. In Appendix A.2.8, we further show that, without multiplicative or internal noise, enforcing OP recovers the classical LQAG solution.

## 4 OPTIMAL CONTROL WITH MODEL MISMATCH

We have shown that an analytical solution to the LQMI control problem can be derived requiring only standard assumptions: linear Kalman filtering for estimation and linear control laws. However, a central assumption remains unaddressed. By optimizing estimation and control gains ($K_{0,...,T}$ and $L_{0,...,T}$) one implicitly assumes i) that the agent's internal model exactly matches the true dynamics, and ii) that optimal behavior emerges from optimizing estimation and control as a partially decoupled process. This formalization weakens the notion of partial observability by presuming full access to the external world's dynamics. Although such knowledge could, in principle, be learned, it imposes strong constraints on the agent's internal strategy, leaving little room for internal computations that are structurally independent from the environment.

This perspective also risks underestimating the role of internal representations, which are central to many motor control studies (Wolpert et al., 1995; Kawato, 1999; Shadmehr & Krakauer, 2008; Franklin & Wolpert, 2011; Golub et al., 2013; 2015). Beyond these classical formulations, a broader neuroscience literature has shown that internal models need not faithfully match the external dynamics. Frameworks such as optimal feedback control and forward-model learning posit that internal dynamics may be simplified, biased, or task-dependent (Kawato, 1999; Wolpert & Ghahramani, 2000; Shadmehr & Holcomb, 1997; Scott, 2004). Empirical work further demonstrates that neural population activity often reflects internally generated dynamics optimized for control or prediction rather than a veridical copy of the physical plant (Churchland et al., 2012; Gallego et al., 2017). These ideas align with the conceptual motivation behind our Model-Mismatch framework, intro-

duced next, where the internal model is optimized jointly with control rather than constrained to follow the true system dynamics.

Allowing internal models to differ from the laws governing the external world extends the flexibility of the stochastic optimal control framework, opening the door to a richer class of biologically plausible computations. In addition, this flexibility may lead to improved solutions in terms of cost minimization, particularly when internal representations are affected by noise (Hazon et al., 2022; Panzeri et al., 2022; Moreno-Bote et al., 2014).

We then consider a more general control problem where the internal dynamics are also optimized and may become mismatched with the actual forward dynamics of the state variables. We formalize the new *Model Mismatch* (M-Mis) framework over an even more general LQMI problem than the one described in Sec. 2, allowing fully generalized multiplicative noise: both the state and the internal dynamics may be affected by noise that depends on the state and on the internal variable. We define the control problem as

$$x_{t+1} = Ax_t + BL_tz_t + n_t^x \ , \quad y_t = Hx_t + n_t^y \ , \quad z_{t+1} = W_tz_t + P_ty_t + n_t^z \quad (11)$$

$$n_t^c = \epsilon_t^c + \sum_r \eta_t^c U_r^c x_t + \sum_l \xi_t^c V_l^c L_t z_t \ , \quad c \in \{x, y, z\} \ ,$$

where notation follows Eqs. 5-7, with appropriate matrix dimensions and noises with covariances $\mathbb{E}[\epsilon_t^c \epsilon_t^{c'}] = \Sigma_{\epsilon^c} \delta_{cc'}$, and i.i.d. one-dimensional noises $\eta_t^c$ and $\xi_t^c$ with unit variance. We introduce additive and multiplicative noises $n_t^c$ in the dynamics, observation and internal dynamics $z_t$. Sums over $r$ and $l$ can be $c$-dependent. We consider control-dependent noise, where the control is given by $u_t = L_t z_t$, rather than modeling the multiplicative noise as directly proportional to $z_t$. $P_t \in \mathbb{R}^{n \times m}$ is a pseudo-filter matrix that takes the observation $y_t$ and inputs it to the dynamics of the internal variable $z_t$, which follows a linear system with time-dependent forward dynamics $W_t \in \mathbb{R}^{n \times n}$.

Importantly, in the M-Mis framework, the internal variable $z_t$ integrates both control and estimation signals, unlike in the Model Match case where $z_t$ is constrained to represent a state estimate. In the former, since $W_t$ need not match the external dynamics, $z_t$ can evolve independently of $x_t$ and encode dynamics optimized for control rather than estimation. The internal variable $z_t$ has dimension $n$, while the control signal $u_t = L_t z_t$ is again $p$-dimensional, with $L_t \in \mathbb{R}^{p \times n}$. The problem consists in optimizing the time-dependent, forward dynamics $W_{0,...,T}$, pseudo-filter $P_{0,...,T}$ and control $L_{0,...,T}$ matrices so as to minimize the cost in Eq. 3, with initial condition $p_0(x, z)$. Using the same procedure as in the Model Match approach (Sec. 3) – since the two problems share the same underlying mathematical structure – we derive a coordinate-descent algorithm guaranteed to converge to a critical point of the cost (Appendix A.2.9; pseudocode in Appendix A.3.2, Algorithm 2).

## 5 EXPERIMENTS

### 5.1 COMPARISON WITH CURRENT NUMERICAL AND ANALYTICAL METHODS

To compare against the current state-of-the-art numerical approach for LQMI cost minimization – the gradient- descent (GD) method of Damiani et al. (2024) – we apply our algorithm to the same single-joint reaching task used in Todorov (2005) and Damiani et al. (2024) (problem details in Appendix A.4.1). Our M-Match algorithm (Algorithm 1) converges to a critical point of the cost function (Fig. 1a) and recovers the same optimal control and filter gains as the GD approach (control gains shown in Fig. 1b), while achieving a substantial computational speedup. On a standard laptop, our algorithm (Algorithm 1) converges in approximately 6 seconds, compared to more than 5 hours for the GD method, and achieves the same expected cost, $C = 0.32$. In Appendix A.4.2, we further evaluate computational scaling on increasingly high-dimensional systems (up to 100 state dimensions), demonstrating both robustness and a growing advantage over GD. In the largest setting tested, runtime decreases from more than two days to only 2.7 seconds. Moreover, we confirm the findings of Damiani et al. (2024), showing that the suboptimal (see the discussions in Sec. 1 and Appendix A.1) solutions obtained with the algorithm of Todorov (2005) diverge substantially once internal noise is present, yielding much larger (in absolute value) control gains (Fig. 1c) and significantly worse performance, with an expected cost of $C = 0.50$.

## 5.2 OPTIMAL CONTROL IN MULTIDIMENSIONAL MOTOR TASKS

We then apply the Model Match (M-Match), Model Mismatch (M-Mis), and (Todorov, 2005) (TOD) approaches to two additional motor-control tasks.

**3D Reaching Task** We first examine a 3D reaching task – a multidimensional extension of the classic single-joint paradigm of the previous Section – with a 6-dimensional state including positions and velocities ($m = n = p = k = 6$; see Appendix A.4.3 for details). The coordinate-descent algorithm for the M-Mis framework (Algorithm 2) converges reliably across a wide range of internal-noise levels $\sigma_\eta$ (Fig. 1d), achieving substantially lower cost as internal noise increases (Fig. 1e) when compared to the M-Match and TOD solutions. In Fig. 1f, $\tilde{W}_t = A + BL_t - P_tH$ (with $P_t$ corresponding to $K_t$ in Eq. 7) denotes the forward dynamics required for M-Mis to reduce to the classical M-Match case. Indeed, setting $W_t = \tilde{W}_t$ recovers the Kalman filter update for $z_t$ in Eq. 11, so that $z_t$ acts as a standard state estimate of $x_t$. As internal noise increases, however, the optimal $W_t$ deviates progressively from $\tilde{W}_t$ (Fig. 1f), indicating that internal representations no longer attempt to mirror the external dynamics. Instead, $z_t$ becomes an abstract internal variable that integrates sensory feedback and past information in a way that supports robust control rather than faithful state estimation. Consequently, the internal variable $z_t$ can no longer be interpreted as an estimate of the state $x_t$; instead, it becomes a more abstract representation that integrates sensory feedback and past information to support optimal control (Fig. 1g), yet drastically reducing the cost - Fig. 1e. Appendix A.4.3 provides additional analyses illustrating how sensory weighting, control readouts, and internal dynamics adapt to internal fluctuations in the M-Mis framework. To further illustrate the conceptual shift, Appendix A.4.4 outlines example behavioral and neural predictions that distinguish the Model Mismatch and Model Match approaches.

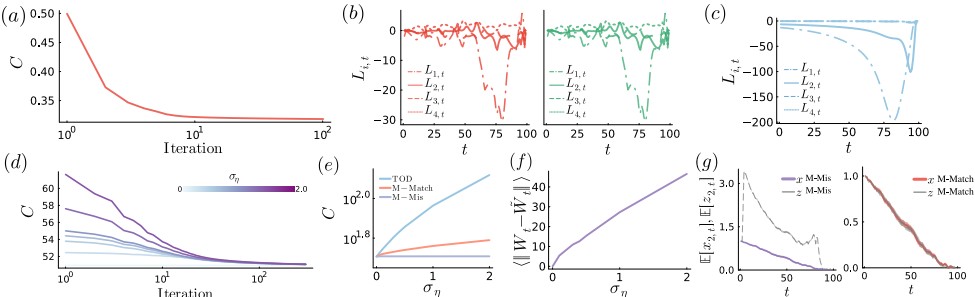

Figure 1: *Comparison With Current Methods and Cost Reduction via Model Mismatch.* **(a)** Expected accumulated cost $C$ (Eq. 3), during joint optimization of control and filter gains using Algorithm 1. **(b)** Optimal control gains $L_t$ obtained with the M-Match algorithm – Algorithm 1 – (red, left) and with the numerical gradient-descent approach of Damiani et al. (2024) (green, right). Here, $L_{i,t}$ denotes the $i$-th component of the 4-dimensional control-gain vector at time $t$. **(c)** Same as (b), but for the solutions obtained using the algorithm of Todorov (2005) **(d)** Convergence of the Model Mismatch algorithm – Algorithm 2 – for different internal noise levels $\sigma_\eta$. **(e)** Expected cost for TOD (Todorov, 2005) (blue), Model Match (red), and Model Mismatch (purple). **(f)** Time-averaged norm of $W_t - \tilde{W}_t$. **(g)** Second component of $x_t$ and $z_t$ (mean $\pm$ SEM, $\sigma_\eta = 0.1$) for M-Mis (left) and M-Match (right).

**Application to a Redundant Arm-Control Task** We next evaluate our algorithms on a more realistic and structurally complex motor-control problem: a 3-DOF planar arm performing a reaching movement around a stable reference posture. The arm is actuated by nine muscle-like control channels that map linearly onto three joint torques through a matrix $S$ (the full model and parameter choices are reported in Appendix A.4.5). This actuation redundancy (9 controls for 3 torques) is a hallmark of biological musculo-skeletal systems and is widely studied in robotics and computational motor control to analyze coordination under redundancy (Tahara et al., 2009).

As in the previous 3D reaching task, the M-Mis framework yields substantially more robust performance across internal-noise levels, consistently achieving lower cost than both M-Match and TOD (Todorov, 2005) (Fig. 2a, purple curve).

Because musculo-skeletal systems admit multiple muscle activation patterns that produce identical torques, a standard approach for understanding coordination is through muscle synergies, i.e., low-dimensional patterns of co-activation (d'Avella & Bizzi, 2005; Tresch et al., 2006; Valero-Cuevas et al., 2009; Kutch & Valero-Cuevas, 2012; Todorov & Jordan, 2002). Synergy analyses show that biological motor systems concentrate control effort along task-relevant directions, in line with the "minimal intervention principle" (Valero-Cuevas et al., 2009; Safavynia & Ting, 2012). Our solutions exhibit the same structure. We decompose the control signal $u_t$ using the standard pseudoinverse projection: $u_t^{\text{torque}} = S^\dagger S u_t$ and $u_t^{\text{null}} = (I - S^\dagger S)u_t$, where $S^\dagger$ is the pseudoinverse of $S$, yielding components in the torque-producing and muscle null spaces (with $S u_t^{\text{null}} = 0$ by construction). Computing the projected effort $\mathbb{E}[|u_t^{\text{proj}}|^2]$ for $\text{proj} \in \{\text{torque}, \text{null}\}$ shows that virtually all control effort lies in the torque-producing subspace, with negligible activation in the null space (Fig. 2b; identical results for M-Mis, not shown). Thus, redundancy is resolved by selecting minimal-effort torque-producing patterns rather than co-activating muscles along null directions – consistent with empirical observations in human and animal motor control (Valero-Cuevas et al., 2009) and widely used strategies in robotics (Dietrich et al., 2015).

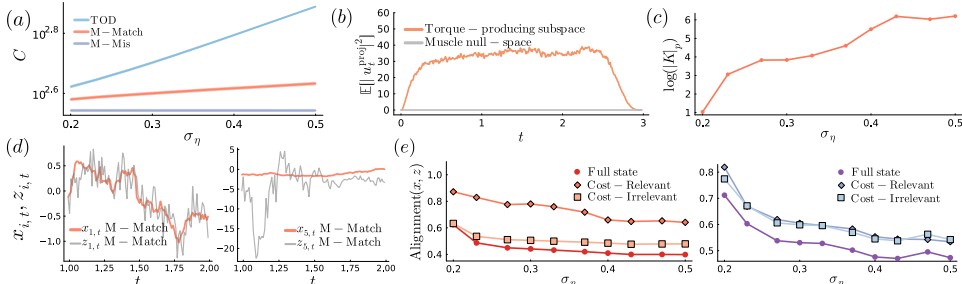

Figure 2: *Task-Aligned Control Under Model Match and Model Mismatch.* **(a)** Expected cost for the algorithm from Todorov (2005) (TOD, blue), Model-Match (M-Match, red) and Model-Mismatch (M-Mis, purple), averaged over $500$ Monte-Carlo trials (shaded areas show the standard error of the mean). **(b)** Squared magnitude of the time-dependent control signal projected onto the torque-producing subspace and onto the muscle null-space, with $\text{proj} \in \{\text{torque}, \text{null}\}$ defined as $u_t^{\text{torque}} = S^\dagger S u_t$ and $u_t^{\text{null}} = (I - S^\dagger S)u_t$. Curves are averaged over $500$ trials (standard error mean shading barely visible) for $\sigma_\eta = 0.23$ in the M-Match solution (M-Mis shows similar trends; not shown). **(c)** Time-averaged logarithm of the pseudodeterminant of the control gain matrices $K_t$ in the M-Match framework as a function of internal noise. The log of the pseudodeterminant is computed as the sum of the logarithms of all singular values of $K_t$ above a numerical tolerance ($10^{-12}$). **(d)** First (left panel) and fifth (right panel) component of the vectors $x_t$ and $z_t$ for a representative trial of the M-Match solution with $\sigma_\eta = 0.23$ (temporal window between 1–2 s shown for clarity). **(e)** Alignment between the state $x_t$ and the internal state $z_t$, averaged over time and over $500$ trials, in the Model-Match framework (left panel) and in the Model-Mismatch framework (right panel). Circles indicate alignment between the full vectors; squares indicate alignment restricted to cost-irrelevant dimensions (the last three components, i.e. the angular velocities, which are weakly penalized by the cost $Q_t$); and diamonds indicate alignment restricted to cost-relevant dimensions (the first three components, i.e. joint angles).

The performance gap between the M-Match and M-Mis frameworks in Fig. 2a stems from fundamentally different internal computations. In the M-Match case, the internal dynamics tend to channel variability into cost-irrelevant and unobserved state dimensions, thereby stabilizing the control output (in this task only joint angles are penalized and observed, as defined by $Q$ and $H$) – see Appendix A.4.5 for additional analyses. In parallel, sensory feedback gains increase with internal-noise magnitude (Fig. 2c shows the time-averaged log-pseudodeterminant of $K_t$, i.e. the sum of the logarithms of its non-zero singular values), allowing the system to compensate for internal fluctuations while maintaining accurate estimates of the cost-relevant state components. Consequently, on individual trials, the first three components of $z_t$ reliably track the corresponding components of the physical state (Fig. 2d, left panel), whereas the remaining components diverge and decouple from $x_t$ (Fig. 2d, right panel). This strategy remains stable across noise levels (see Appendix A.4.5). In the M-Mis framework, by contrast, the internal dynamics are no longer constrained to implement a

Kalman-like recursion. Instead, they reorganize to stabilize the entire control loop, producing internal representations that no longer track $x_t$ (consistent with Figs. 1f–g), but instead adapt to noise in a way that supports robust control (see Appendix A.4.5). Fig. 2e illustrates this difference: we plot the alignment (absolute cosine similarity) between $x_t$ and $z_t$ across state dimensions. In M-Match, the observed—and cost-relevant components remain strongly aligned with $z_t$, while the cost-irrelevant ones progressively decouple as internal noise grows. In M-Mis, all components show uniformly low alignment with $z_t$, indicating that the internal variable encodes representations optimized for control rather than for state estimation.

Taken together, these results show that our algorithm scales naturally to high-dimensional, redundant biomechanical systems and yields clear, testable predictions. In M-Match, internal noise drives a noise-suppression strategy that channels variability into unobserved, cost-irrelevant dimensions; in M-Mis, synergies remain stable while internal dynamics reorganize to preserve output stability. These contrasting computations lead to experimentally accessible signatures – such as EMG patterns, muscle-synergy adaptation, alignment or misalignment between neural and behavioral subspaces, and noise-dependent changes in sensory weighting – that can be directly probed in human motor control and robotics.

## 5.3 NEURAL POPULATION STEERING VIA MODEL MISMATCH CONTROL

Finally, we apply our framework to a neural population–steering task, where an unstable recurrent network is driven toward a target state via optimized linear readouts from another population – a setting reminiscent of biologically inspired machine-learning approaches (Jaeger & Haas, 2004; Maass et al., 2002; Sussillo & Abbott, 2009). This task connects to recent work using optimal control to study neural population dynamics (Costa et al., 2024; Kao et al., 2021; Slijkhuis et al., 2023; Athalye et al., 2023). Classical approaches (Todorov, 2005; Damiani et al., 2024) require the internal variable $z_t$ to behave as a Kalman filter estimate of $x_t$ by enforcing the structural constraint $W_t = A + BL_t - P_t H$ in Eq. 11, so that $z_t$ follows Eq. 7. In contrast, the Model Mismatch framework removes this constraint by allowing $W_t$ to be freely optimized, enabling $z_t$ and $x_t$ to represent distinct neural populations with independent connectivity matrices $W$ and $A$ (Fig. 3a). The M-Mis algorithm also supports partial optimization; for instance, $W$ and $P$ can be fixed (e.g., random or biologically plausible) while optimizing only $L_t$. Such configurations are incompatible with the Model Match framework, which ties $z_t$'s connectivity to $x_t$ and forces $W_t$ to vary over time, making it unsuitable for simulating interactions between distinct neural populations.

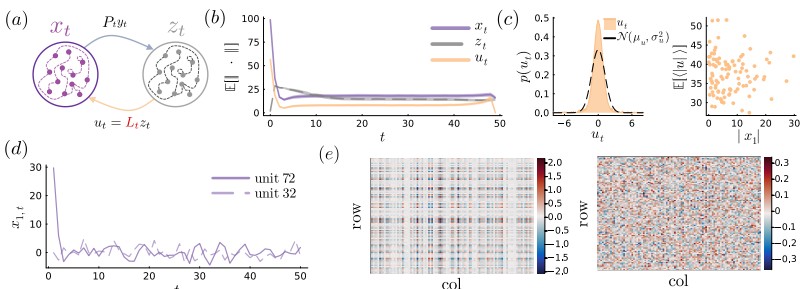

Figure 3: *Model Mismatch for Neural Population Steering*. **(a)** Sketch of the neural population steering task. **(b)** Average (over noise realizations) norm of $x_t$, $z_t$, and of the control signal $u_t = L_t z_t$ with error bars (standard error of the mean). **(c)** Distribution of the control signal over time and realizations with Gaussian fit (left), and average control magnitude (over time and realizations) received by each unit as a function of its initial absolute activity (right). **(d)** Activity of two units from the population vector $x_t$ in a single trial. **(e)** Heatmaps of the matrices $L_t$ at two time points: early (left) and mid-trial (right).

We consider two populations of $N_{\text{units}} = 100$ linear neurons, each with sparse, time-invariant random connectivity (Appendix A.4.6 for details). The activity of the population $z_t$ is read out through a time-varying matrix $L_t$, optimized to steer the population $x_t$ toward a target while minimizing control effort (Fig. 3a). The population $z_t$ receives inputs from $x_t$ through sparse random pro-

jections. The Gaussian-distributed recurrent and feedforward matrices ($A$, $W$, $P$) follow standard assumptions from dynamical mean-field theory (Sompolinsky et al., 1988; Rajan et al., 2010).

We optimize only the readout weights $L_{0,...,T}$ keeping all other parameters fixed. As a result, $x_t$ is reliably steered toward the target (Fig. 3b) through a distributed control strategy: all units in the $x$ population receive, on average, similar amounts of control (Fig. 3c). Despite this overall uniform drive, the control selectively targets the units initially farthest from the target (zero in this coordinate frame), as shown in Fig. 3d. This selective modulation likely reflects the interplay between the recurrent dynamics of $x$ and the structure of $L_{0,...,T}$. Early in the trial, $L_1$ is highly structured and low-rank (Mastrogiuseppe & Ostojic, 2018), strongly pulling activity toward the target; after a transient ($t \geq \tilde{t}$), $L_{\tilde{t},...,T}$ becomes sparse and high-rank, stabilizing the system around the target despite intrinsic instability and noise (Fig. 3e). This mirrors strategies observed when controlling recurrent networks with reinforcement learning (Mastrogiuseppe & Moreno-Bote, 2024).

The Model Mismatch framework therefore extends stochastic control beyond the standard agent–environment formulation and provides a tool for studying also neural computation. In this simplified setting, $z_t$ can be viewed as a premotor population driving a downstream motor population $x_t$, consistent with experimental findings where premotor activity initializes motor cortex before movement (Kao et al., 2021; Logiaco et al., 2021). While not intended as a detailed biological model, this example illustrates how the framework captures computational strategies – such as low-to-high rank transitions, selective modulation, and stabilization of unstable dynamics – that classical Model Match approaches cannot represent.

## 6 Conclusions

We have introduced a convergent iterative algorithm (Sec. 3) that fully solves stochastic optimal control problems under a general noise model with both multiplicative and internal noise, assuming linear control with a quadratic cost – the so-called LQMI problem. This goes beyond previous analytical approaches, which remained incomplete (Todorov, 2005; Damiani et al., 2024). Our algorithm also outperforms existing state-of-the-art gradient-based methods (Damiani et al., 2024) by more than three orders of magnitude in efficiency on realistic tasks, making it particularly well suited for inverse optimal control.

Moreover, the Model Mismatch framework relaxes two central assumptions in stochastic control: (1) the partial decoupling of estimation and control, and (2) the requirement that internal forward dynamics match the actual state dynamics. By allowing internal dynamics – used to generate control signals – to be optimized jointly with control and pseudo-filter gains, our framework broadens the solution space. Notably, we find that mismatched forward dynamics can outperform matched dynamics in the presence of internal noise. This suggests that internal representations need not faithfully track the state variable; instead, mixed representations of estimation and control signals can provide superior performance. Furthermore, the Model Mismatch framework extends the applicability of stochastic optimal control to the control of neural populations.

Overall, our work expands stochastic optimal control to a more general and realistic setting, with direct applications to neuroscience and robotics, while preserving analytical tractability and interpretability.

**Limitations and Future Work** We assume linear dynamics, linear control, and a quadratic cost, which yield closed-form second-order moments and analytical tractability but might not capture all problems of interest. Nevertheless, the framework accommodates time-varying dynamics, which can approximate nonlinearities. Another promising research direction is to combine our solutions with iLQG and DDP methods (Li & Todorov, 2007; Tassa et al., 2014; Van Den Berg et al., 2016; Liao & Shoemaker, 1991), which approximate optimal control in nonlinear systems under partial observability by locally linearizing the dynamics and using quadratic approximations to the value function. A potential advantage of our approach is that, by using the Model-Mismatch framework, we do not need to assume a model-matched extended Kalman filter – as is typically done in iLQG and DDP – and we can also avoid the unbiased-estimator assumption. Another relevant direction is that the Model Mismatch framework allows internal dimensionality to be freely chosen – a promising but unexplored direction that could support nonlinear strategies via linear representations (Korda & Mezić, 2018; Brunton et al., 2016).

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

## A  APPENDIX

### A.1  UNBIASEDNESS AND ORTHOGONALITY: CLARIFICATIONS AND IMPLICATIONS

Here we briefly review related work on stochastic optimal control in the presence of multiplicative and internal noise (LQMI problem, Sec. 2.1). The influential work of Todorov (2005) introduced an iterative algorithm that alternates between optimizing the control and filter gains until convergence. A key assumption in this derivation is *unbiased estimation*, i.e., $\mathbb{E}[x_t \mid z_t] = z_t$, used to constrain the control policy to depend solely on the internal estimate $z_t$, in line with the problem's partial observability.

However, Damiani et al. (2024) empirically showed that this unbiasedness condition is generally violated, with the discrepancy growing as internal noise increases. They also proposed an alternative numerical algorithm that avoids assuming unbiasedness and empirically outperforms the original approach under internal noise.

The reason the method in Todorov (2005) still performs optimally when internal noise is absent is that unbiasedness implies the *orthogonality principle* (Davis, 2013; Damiani et al., 2024), which characterizes the optimal estimator in that specific case. Importantly, orthogonality does not imply unbiasedness, so the converse does not hold. Thus, the success of Todorov (2005) in the zero internal noise regime stems from its implicit reliance on orthogonality, which breaks down otherwise.

In Appendix A.2.7, we provide a formal proof that the orthogonality principle corresponds to a critical point of the cost function in Eq. 3 only in the absence of internal noise, extending and mathematically validating the empirical observations in Damiani et al. (2024). Moreover, in Appendix A.2.8, we demonstrate that the orthogonality principle actually leads to the global optimum for the classic LQAG problem.

### A.2  SOLVING THE LQMI PROBLEM: FULL DERIVATIONS

Here we provide an algorithm guaranteed to converge to a critical point of the cost function in Eq. 3, under the dynamics in Eqs. 5,6,7. As shown in prior work (Fazel et al., 2018), the global LQAG problem is non-convex even in the fully observable, noise-free setting. This implies that the more general problem considered here - featuring multiplicative and internal noise - is also non-convex. The algorithm yields improved pairs of control and filter gains, fully solving the LQMI problem. The pseudocode is shown in Appendix A.3.1.

### A.2.1  FIXED-POINT EQUATIONS OF THE COST FUNCTION

Assuming a linear control $u_t = L_t z_t$, we first rewrite the cost function in Eq. 3 as $C = \sum_{t=0}^{T} \left( \text{tr}(Q_t S_t^{xx}) + \text{tr}(L_t^\top R_t L_t S_t^{zz}) \right)$, where we introduce the 2nd-order moment matrices $S_t^{xx} = \int dx dz\, p_t(x, z) x x^\top$, $S_t^{zz} = \int dx dz\, p_t(x, z) z z^\top$, and $S_t^{xz} = \int dx dz\, p_t(x, z) x z^\top$, with $p_t(x, z)$ being the joint distribution of $x$ and $z$ at time $t$ generated by previous control and filter gains and averaging over noises and initial conditions following $p_0(x, z)$. To find the conditions for extrema on the control $L_{0,\dots,T}$ and filter $K_{0,\dots,T}$ gains we add Lagrange multipliers and define the new objective

$$C_{\mathcal{L}} = \sum_{t=0}^{T} \left( \text{tr}(Q_t S_t^{xx}) + \text{tr}(L_t^\top R_t L_t S_t^{zz}) \right) - \sum_{t=1}^{T+1} \left( \text{tr}(\Lambda_t G_t^{xx}) + \text{tr}(\Omega_t G_t^{zz}) + \text{tr}(\Gamma_t G_t^{xz}) \right) , \quad (12)$$

where $\Lambda_t$, $\Omega_t$ and $\Gamma_t$ are $\mathbb{R}^{m \times m}$ matrices of Lagrange multipliers. The constraints $G_t^{xx} = G_t^{zz} = G_t^{xz} = 0$ are given by the temporal evolution of the 2nd-order moment matrices $S_t^{xx}$, $S_t^{zz}$ and $S_t^{xz}$, respectively, between two consecutive time steps $t$ and $t+1$, obtained from Eqs. 5,6,7 (see Appendix A.2.4 for details), as

$$G_{t+1}^{xx} = S_{t+1}^{xx} - A S_t^{xx} A^\top - A S_t^{xz} L_t^\top B^\top - B L_t (S_t^{xz})^\top A^\top - B L_t S_t^{zz} L_t^\top B^\top - \Sigma_t^{xx}$$

$$G_{t+1}^{zz} = S_{t+1}^{zz} - K_t H S_t^{xx} H^\top K_t^\top - K_t H S_t^{xz} M_t^\top - M_t (S_t^{xz})^\top H^\top K_t^\top - M_t S_t^{zz} M_t^\top - \Sigma_t^{zz}$$

$$G_{t+1}^{xz} = S_{t+1}^{xz} - A S_t^{xx} H^\top K_t^\top - B L_t S_t^{zz} M_t^\top - A S_t^{xz} M_t^\top - B L_t (S_t^{xz})^\top H^\top K_t^\top , \quad (13)$$

where we have introduced the short-hand notation $M_t = A + B L_t - K_t H$, showing up repetitively, and the noise matrices $\Sigma_t^{xx} = \Sigma_\xi + \sum_i C_i L_t S_t^{zz} L_t^\top C_i^\top$ and $\Sigma_t^{zz} = \Sigma_\eta + K_t \Sigma_\omega K_t^\top +$

$K_t \left( \sum_i D_i S_t^{xx} D_i^\top \right) K_t^\top$. Since the cost function is defined in terms of quadratic terms in $x$ and $z$ and the temporal evolution of moments is closed at 2nd-order, the 2nd-order moments matrices are sufficient statistics of the problem (i.e., $p_t(x, z)$ does not need to be explicitly known), and only the constraints in their temporal evolution suffice.

For convenience, we define the Lagrange multipliers at time $T + 1$ to be all equal to zero, $\Lambda_{T+1} = \Omega_{T+1} = \Gamma_{T+1} = 0$ (hereafter 0 meaning a matrix of zeros of consistent dimensions), so the constraints at that time are irrelevant. The introduction of Lagrange multipliers enables to take derivatives with respect the control and filter gains to find the fixed point conditions $\partial C_\mathcal{L}/\partial L_t = 0$ and $\partial C_\mathcal{L}/\partial K_t = 0$ for extrema without the need to propagate derivatives over the terms in the sum of the cost. The fixed point equations take the form

$$L_t = E_t^{-1} \left( F_t S_t^{xz} (S_t^{zz})^{-1} + J_t \right) \tag{14}$$

$$K_t = \left( S_{AH} + \tilde{\Omega}_{t+1}^{-1} \Gamma_{t+1} S_{LH} \right) S_{HH}^{-1} , \tag{15}$$

with matrices defined in Appendix A.2.6 – note that these equations express the control and filter gains as a function of themselves, and therefore they are implicit.

From the conditions $\partial C_\mathcal{L}/\partial S_t^{xx} = \partial C_\mathcal{L}/\partial S_t^{zz} = \partial C_\mathcal{L}/\partial S_t^{xz} = 0$, the Lagrange multipliers themselves obey the set of equations

$$\Lambda_t = Q_t + A^\top \Lambda_{t+1} A + H^\top K_t^\top \Omega_{t+1} K_t H + H^\top K_t^\top \Gamma_{t+1} A + \sum_i D_i^\top K_t^\top \Omega_{t+1} K_t D_i$$

$$\Omega_t = L_t^\top R_t L_t + L_t^\top B^\top \Lambda_{t+1} B L_t + M_t^\top \Omega_{t+1} M_t + M_t^\top \Gamma_{t+1} B L_t + \sum_i L_t^\top C_i^\top \Lambda_{t+1} C_i L_t$$

$$\Gamma_t = L_t^\top B^\top \tilde{\Lambda}_{t+1} A + M_t^\top \tilde{\Omega}_{t+1} K_t H + M_t^\top \Gamma_{t+1} A + L_t^\top B^\top \Gamma_{t+1}^\top K_t H . \tag{16}$$

These equations can be solved backwards given control and filter gains, and using the boundary conditions $\Lambda_{T+1} = \Omega_{T+1} = \Gamma_{T+1} = 0$. However, the full solution to Eqs. 14,15,16 would require simultaneously determining gains and multipliers. We bypass this by deriving an iterative algorithm to find fixed point solutions, as described in the next section.

It is worth mentioning that in the derivation of Eqs. 14,15,16 and main algorithm described below we have not assumed the orthogonality principle (OP: $S_t^{xz} = S_t^{zz}$ for all $t$, equivalent to $\mathbb{E}[(x_t - z_t)z_t^\top] = 0$), which is shown (Sec. 3.1, see also Appendix A.1) to be violated in the general case (specifically, whenever there is internal noise). Secondly, we have not assumed any specific initial distribution $p_0(x, z)$. Also, note that we have not assumed Gaussian noises nor Gaussian distribution on $x$ or $z$. Further, our algorithm is guaranteed to converge to a fixed-point pair of control and filter gains, and reduce the cost at every step (Sec. A.2.2). The algorithm in Todorov (2005) can actually increase the cost in the first iteration step because not for any arbitrary initial filter gain OP is obeyed. Finally, the model described in Eqs. 5,6,7 could be readily extended to the case where i) the internal noise is multiplicative in Eq. 7, ii) when there is $x$-dependent multiplicative noise in the state dynamics, Eq. 5, and iii) when there is $z$-dependent multiplicative noise in the feedback dynamics, Eq. 6. However, we refrain from doing so to avoid clutter and because a more general framework (Model Mismatch) is introduced in Sec. 4.

### A.2.2 COORDINATE-DESCENT ALGORITHM FOR JOINT CONTROL AND FILTER OPTIMIZATION

Here we derive the main algorithm of the paper, a coordinate-descent iterative algorithm that gives a pair of improved, fixed-point control and filter gains. We first start by showing the connection between the Lagrange multipliers and the cost-to-go incurred by starting at fixed $x$ and $z$.

We define the cost-to-go starting at $x$ and $z$ from time $t$ ($t = 0, ..., T$) up to time $T$ as $C_t(x, z) = \text{tr}(Q_t xx^\top + L_t^\top R_t L_t zz^\top) + \sum_{\tau=t+1}^{T} \mathbb{E}\left[x_\tau^\top Q_\tau x_\tau + u_\tau^\top R_\tau u_\tau\right]$, where the expectation is over the noises with initial conditions fixed at $x$ and $z$ at time $t$, and for specific control and filter gains from time $t$ onward. This definition is consistent with our definition of cost in Eq. 3, as $C = \int p_0(x, z)C_0(x, z)$, where $p_0(x, z)$ is the distribution of initial conditions over $x$ and $z$. The cost-to-go obeys the Bellman equation

$$C_t(x, z) = \text{tr}(Q_t xx^\top + L_t^\top R_t L_t zz^\top) + \int dx' dz' C_{t+1}(x', z') p_{x,t+1}(x'|x, z) p_{z,t+1}(z'|x, z) , \tag{17}$$

where the transition probability densities $p_{x,t+1}(x'|x,z)$ and $p_{z,t+1}(z'|x,z)$ are defined by equations 5,6,7 with $u_t = L_t z_t$, with means $\mathbb{E}[x'|x,z] = Ax + BL_t z$ and $\mathbb{E}[z'|x,z] = K_t H x + M_t z$, and conditional 2nd-order moments given by Eqs. 33.

The Bellman equation 17 can be solved backwards: noticing that the boundary condition is the final cost-to-go $C_T(x,z) = \text{tr}(Q_T xx^\top + L_T^\top R_T L_T zz^\top)$ and that the 2nd-order moments are closed (that is, no higher nor lower moments appear when propagating backwards the cost-to-go using Eq. 17), we find that the solution is given by

$$C_t(x,z) = \text{tr}(\Lambda_t xx^\top + \Omega_t zz^\top + \Gamma_t xz^\top) + \gamma_t , \tag{18}$$

where it can be seen that the coefficients $\Lambda_t$, $\Omega_t$ and $\Gamma_t$ are actually the Lagrange multipliers computed in Eqs. 16 with the same boundary conditions (see Appendix A.2.5), and where $\gamma_t$ can be recursively calculated as

$$\gamma_t = \text{tr}(\Lambda_{t+1}\Sigma_\xi + \Omega_{t+1}K_t\Sigma_\omega K_t^\top + \Omega_{t+1}\Sigma_\eta) + \gamma_{t+1} , \tag{19}$$

with boundary condition $\gamma_T = 0$. Eqs. 18,19 correctly captures the cost-to-go expression at time $T$, and it can be checked that recursively solve Eq. 17. From the definition of the Lagrange multipliers in Eqs. 16, one can see that e.g. higher noise levels or control costs enlarge the corresponding cost-to-go in Eq. 18, and these effects accumulate backwards, as expected.

While Eqs. 18,19 express the exact cost-to-go given control and filter gains if the exact world state $x$ is known, partial observability dictates that our choices of control and filter gains cannot depend on $x$. Indeed, our assumptions that the filter depends only on time and that the control law depends linearly on the current state estimate $z_t$, that is, $u_t = L_t z_t$, have already been used in our derivation and problem formalization, and they are subject to partial observability. Because of this, we integrate over the (generally unknown) joint probability density $p_t(x,z)$ given control and filter gains and initial condition $p_0(x,z)$ to define the averaged cost-to-go as

$$C_t = \int dx dz \, p_t(x,z) C_t(x,z) = \text{tr}(\Lambda_t S_t^{xx} + \Omega_t S_t^{zz} + \Gamma_t S_t^{xz}) + \gamma_t . \tag{20}$$

We can express the total cost in Eq. 3 as $C = C_0$, and therefore

$$C = C_{<t} + C_t \tag{21}$$

with $C_{<t} = \sum_{\tau=0}^{t-1} \text{tr}(Q_\tau S_\tau^{xx} + L_\tau^\top R_\tau L_\tau S_\tau^{zz})$ is valid for all $t$. In Eq. 21, $C_t$ is the only term depending on $L_t$, as $C_{<t}$ does not depend on it. Therefore, we locally optimize $L_t$ as

$$L_t^* = \arg\min_{L_t} C_t , \tag{22}$$

while keeping the rest of gains fixed, that is, $L_{0,...,t-1,t+1,...,T}$ and $K_{0,...,T}$ are held constant. A global minimum always exists because $C_t$ is non-negative. After noting that in $C_t$ (Eq. 20) only the Lagrange multipliers depend on $L_t$ (see Eqs. 16), while the 2nd-order moments at time $t$ only depend on previous $L_\tau$ with $\tau < t$ (see Eqs. 32), the minimization results in

$$L_t^* = E_t^{-1}\left(F_t S_t^{xz}(S_t^{zz})^{-1} + J_t\right) , \tag{23}$$

with matrices identical to those in Eq. 14 and Appendix A.2.6, and whenever matrix inverses exist.

If $L_{0,...,T}$ and $K_{0,...,T}$ are the values of the control and filter gains before the optimization in Eq. 22, clearly the cost is non-increasing after the optimization,

$$C(L_0, ..., L_{t-1}, L_t^*, L_{t+1}, ..., L_T) \le C(L_0, ..., L_{t-1}, L_t, L_{t+1}, ..., L_T) . \tag{24}$$

Note that after the optimization, the total cost in Eq. 21 becomes

$$C = C_{<t} + \text{tr}(Q_t S_t^{xx} + L_t^{*\top} R_t L_t^* S_t^{zz}) + \text{tr}(\Lambda_{t+1} S_{t+1}^{xx,*} + \Omega_{t+1} S_{t+1}^{zz,*} + \Gamma_{t+1} S_{t+1}^{xz,*}) + \gamma_{t+1} , \tag{25}$$

where the new 2nd-order moments at time $t + 1$, $S_{t+1}^*$, are computed from the moments at the previous time $t$ using Eqs. 32 with the optimal $L_t^*$ and noticing that the Lagrange multipliers from $t + 1$ onward have not changed. Redefining $L_t^*$ as $L_t$ and the $S_{t+1}^{ab,*}$ as $S_{t+1}^{ab}$, $ab \in \{xx, zz, xz\}$, we can now proceed to optimize $L_{t+1}$ using the same procedure as above (changing $t$ to $t + 1$) to

minimize again the total cost $C(L_0, ..., L_t, L_{t+1}^*, ..., L_T) \leq C(L_0, ..., L_t, L_{t+1}, ..., L_T)$ fixing all the gains except $L_{t+1}$, and consecutively for all $t$ up to $T$.

Therefore, starting from a set of gains $L^{(n)} \equiv L_{0,...,T}^{(n)}$ and $K^{(n)} \equiv K_{0,...,T}^{(n)}$, we can optimize $L_t$ in order from $t = 0$ up to time $T$ following the above steps to get a new set of control gains $L^{(n+1)}$, and clearly we have $C(L^{(n+1)}, K^{(n)}) \leq C(L^{(n)}, K^{(n)})$. After this, the Lagrange multipliers in Eq. 16 are recomputed backwards with the updated values of the control gains, $L^{(n+1)}$. In this way, we can express again the cost as in Eq. 21, but with updated values of control gains and multipliers. This represents a full forward pass to sequentially optimize control gains followed by a full backward pass of the multipliers, and we refer to this process as *control pass*.

We can proceed similarly for the filter gains by repeating the above steps but for $K_t$ instead of $L_t$. We optimize $K_t$ by keeping fixed the remaining filter gains and all control gains by minimizing the cost $C$ in Eq. 21, resulting in

$$K_t^* = \underset{K_t}{\arg \min} \, C_t = \left( S_{AH} + \tilde{\Omega}_{t+1}^{-1} \Gamma_{t+1} S_{LH} \right) S_{HH}^{-1} , \tag{26}$$

with matrices as in Eq. 15 and Appendix A.2.6. After updating the cost $C$ with the new $K_t^*$, we obtain an equation analogous to Eq. 25 having a new $\gamma_{t+1}$ term. This leads to a non-increasing cost change when going from the old $K_t$ to the optimized $K_t^*$, $C(K_0, ..., K_t^*, ..., K_T) \leq C(K_0, ..., K_t, ..., K_T)$. Therefore, starting from a set of gains $L^{(n+1)}$ and $K^{(n)}$, we optimize $K_t$ in order for $t = 0, ..., T$ to get a new set of filter gains $K^{(n+1)}$, which will obey $C(L^{(n+1)}, K^{(n+1)}) \leq C(L^{(n+1)}, K^{(n)})$. After this, the Lagrange multipliers are updated. This represents a *filter pass*: full forward pass to sequentially optimize filter gains followed by a full backwards pass to recompute the multipliers. Starting from arbitrary $L^{(0)}$ and $K^{(0)}$ and distribution of initial conditions $p_0(x, z)$, we can alternate now the control and filter passes, so that $C(L^{(0)}, K^{(0)}) \geq C(L^{(1)}, K^{(0)}) \geq C(L^{(1)}, K^{(1)}) \geq ... \geq C(L^{(n+1)}, K^{(n)}) \geq C(L^{(n+1)}, K^{(n+1)}) \geq ... \geq C_{min} \geq 0$. Since the series is non-negative, it converges to a total cost no higher than the initial one with optimal filters $L^* = L^{(\infty)}$ and $K^* = K^{(\infty)}$. In summary, each block update solves a convex quadratic subproblem exactly, which guarantees that the total cost decreases monotonically and therefore converges. The converged pair of control and filter gains obey the Lagrange Eqs. 32,14,15,16, because Eqs. 23,26, after convergence, are identical to the fixed point Eqs. 14,15. Therefore, the converged pair corresponds a to a fixed point solution of the Lagrangian in Eq. 12, and hence, they must be a critical point of the cost function in Eq. 3. We have thus proven the following

**Theorem 1.** *Starting with arbitrary $L^{(0)}$ and $K^0$ and distribution of initial conditions $p_0(x, z)$, the coordinate descent algorithm defined by iterating in alternation control and filter passes converges to an improved pair of control and filter gains $L^*$ and $K^*$. The improved pair corresponds to a critical point of the cost function in Eq. 3.*

As shown in Eqs. 16, 23, 26, and 32, only the first and second noise moments enter the moment propagation and optimality conditions. No further assumptions are required beyond finite second moments, so the method applies to any noise distribution with finite covariance. In Sec. A.4.8 we validate this empirically using non-Gaussian noise (Student-$t$ for heavy tails and Beta distributions for skewness). We also note that the Lagrange equations may admit multiple solutions. In practice, our algorithm converges to different critical points depending on the initialization, but when initializing the control and filter matrices trying to impose the orthogonality principle and then freely running the algorithm, the best critical point is found, empirically.

### A.2.3 Solutions of the Classic LQAG problem

The optimal $L_{0,...,T}$ and $K_{0,...,T}$, for the classic LQAG problem — defined in Sec. 2.1 – are given by (Doya, 2007; Davis, 2013; Todorov, 2005)

$$L_t = (2R_t + B^\top S_{t+1} B)^{-1} B^\top S_{t+1} A \tag{27}$$

$$S_t = 2Q_t + A^\top S_{t+1}(A + BL_t) \tag{28}$$

$$K_t = A\Sigma_t^e H^\top (H\Sigma_t^e H^\top + \Sigma_\omega)^{-1} \tag{29}$$

$$\Sigma_{t+1}^e = \Sigma_\xi + (A - K_t H)\Sigma_t^e A^\top . \tag{30}$$

A detailed derivation can be found in Doya (2007), Chapter 12, Sections 4 and 5. We observe that the only differences with the Eqs. in Doya (2007) arise from slightly different conventions: in the standard LQAG formulation, there is a prefactor of $1/2$ in front of the cost function, and the control signal is defined as $u_t = -L_t z_t$, meaning the control gain has the opposite sign compared to our convention.

In Appendix A.2.8, we prove that the solutions derived in Sec. A.2 recover these classical results in the absence of multiplicative and internal noise.

### A.2.4 DERIVING THE PROPAGATION OF SECOND-ORDER MOMENTS

Here we derive the temporal evolution of the 2nd-order moment matrices. We first rewrite Eqs. 5,6,7 in a more compact form by inserting the observation in the state estimate variable and grouping terms as

$$x_{t+1} = Ax_t + BL_t z_t + \xi_t + \sum_i \varepsilon_t^i C_i L_t z_t$$

$$z_{t+1} = M_t z_t + K_t H x_t + \eta_t + K_t \omega_t + K_t \sum_i \rho_t^i D_i x_t \tag{31}$$

with $M_t = A + BL_t - K_t H$.

The 2nd-order moments at time $t$ can be computed based on those in the previous time step $t$ by using the appropriate averages and interactions between terms in Eqs. 31. The result is

$$S_{t+1}^{xx} = AS_t^{xx} A^\top + AS_t^{xz} L_t^\top B^\top + BL_t(S_t^{xz})^\top A^\top + BL_t S_t^{zz} L_t^\top B^\top + \Sigma_t^{xx}$$

$$S_{t+1}^{zz} = K_t H S_t^{xx} H^\top K_t^\top + K_t H S_t^{xz} M_t^\top + M_t(S_t^{xz})^\top H^\top K_t^\top + M_t S_t^{zz} M_t^\top + \Sigma_t^{zz}$$

$$S_{t+1}^{xz} = AS_t^{xx} H^\top K_t^\top + BL_t S_t^{zz} M_t^\top + AS_t^{xz} M_t^\top + BL_t(S_t^{xz})^\top H^\top K_t^\top . \tag{32}$$

with $M_t = A + BL_t - K_t H$ and noise covariances $\Sigma_t^{xx} = \Sigma_\xi + \sum_i C_i L_t S_t^{zz} L_t^\top C_i^\top$ and $\Sigma_t^{zz} = \Sigma_\eta + K_t \Sigma_\omega K_t^\top + K_t \left( \sum_i D_i S_t^{xx} D_i^\top \right) K_t^\top$.

The conditional second-order moments at time $t+1$ conditioned on $x$ and $z$ at time $t$ are defined as

$$\hat{S}_t^{xx} = \int dx' dz' x' x'^\top p_{x,t+1}(x'|x,z) p_{z,t+1}(z'|x,z)$$

$$\hat{S}_t^{zz} = \int dx' dz' z' z'^\top p_{x,t+1}(x'|x,z) p_{z,t+1}(z'|x,z)$$

$$\hat{S}_t^{xz} = \int dx' dz' x' z'^\top p_{x,t+1}(x'|x,z) p_{z,t+1}(z'|x,z) ,$$

where the transition probabilities $p_{x,t+1}(x'|x,z)$ and $p_{z,t+1}(z'|x,z)$ are defined by equations 5,6,7 (with $u_t = L_t z_t$), or, equivalently, by Eqs. 31. The conditional second-order moments at time $t+1$ are obtained simply by replacing the second-order moments on the right hand side of Eqs. 32 by their corresponding non-averaged $x$ and $z$ as

$$\hat{S}_{t+1}^{xx} = Axx^\top A^\top + Axz^\top L_t^\top B^\top + BL_t zx^\top A^\top + BL_t zz^\top L_t^\top B^\top + \hat{\Sigma}_t^{xx}$$

$$\hat{S}_{t+1}^{zz} = K_t H xx^\top H^\top K_t^\top + K_t H xz^\top M_t^\top + M_t zx^\top H^\top K_t^\top + M_t zz^\top M_t^\top + \hat{\Sigma}_t^{zz}$$

$$\hat{S}_{t+1}^{xz} = Axx^\top H^\top K_t^\top + BL_t zz^\top M_t^\top + Axz^\top M_t^\top + BL_t zx^\top H^\top K_t^\top . \tag{33}$$

with conditional noise covariances $\hat{\Sigma}_t^{xx} = \Sigma_\xi + \sum_i C_i L_t zz^\top L_t^\top C_i^\top$ and $\hat{\Sigma}_t^{zz} = \Sigma_\eta + K_t \Sigma_\omega K_t^\top + K_t \left( \sum_i D_i xx^\top D_i^\top \right) K_t^\top$.

### A.2.5 CONSISTENCY OF THE COST-TO-GO SOLUTION

The cost-to-go obeys the Bellman equation

$$C_t(x,z) = \text{tr}(Q_t xx^\top + L_t^\top R_t L_t zz^\top) + \int dx' dz' C_{t+1}(x',z') p_{x,t+1}(x'|x,z) p_{z,t+1}(z'|x,z) , \tag{34}$$

identical to Eq. 17. The transition probability densities $p_{x,t+1}(x'|x,z)$ and $p_{z,t+1}(z'|x,z)$ are defined by equations 5,6,7 with $u_t = L_t z_t$, with means $\mathbb{E}[x'|x,z] = Ax + BL_t z$ and $\mathbb{E}[z'|x,z] =$

$K_t H x + M_t z$, and 2nd-order moments given by Eqs. 33. These will be important to compute averages as needed.

We propose a solution to the Bellman equation of the form

$$C_t(x, z) = \text{tr}(\Lambda_t x x^\top + \Omega_t z z^\top + \Gamma_t x z^\top) + \gamma_t \,, \tag{35}$$

identical to Eq. 18. Our goal is to show that it is possible to find a solution with such a form, and that the expression of the coefficients $\Lambda_t$, $\Omega_t$ and $\Gamma_t$ are actually identical to the Lagrange multipliers in Eqs. 16 with the same boundary conditions. In addition we want to show that $\gamma_t$ follows Eq. 19 with boundary condition $\gamma_T = 0$.

We first note that Eq. 35 is true for $t = T$, because $C_T(x, z)$ should be $C_T(x, z) = \text{tr}(Q_T x x^\top + L_T^\top R_T L_T z z^\top)$ and indeed this coincides with Eq. 35 when taking $\Lambda_T = Q_T$, $\Omega_T = L_T^\top R_T L_T$, $\Gamma_T = 0$ and $\gamma_T = 0$, which in turn are consistent with the Lagrange multiplier expression in Eq. 16 for $t = T$.

Now, assume that Eq. 35 is true for some $t + 1$. Let us show that then it is true for $t$. We insert Eq. 35 for $t + 1$ into Eq. 34 and use the expression of the conditional 2nd-order moments in Eqs. 33 to obtain

$$C_t(x, z) = \text{tr}(Q_t x x^\top + L_t^\top R_t L_t z z^\top)$$
$$+ \int dx' dz' \left( \text{tr}(\Lambda_{t+1} x' x'^\top + \Omega_{t+1} z' z'^\top + \Gamma_{t+1} x' z'^\top) + \gamma_{t+1} \right) p_{x,t+1}(x'|x,z) p_{z,t+1}(z'|x,z)$$
$$= \text{tr}(Q_t x x^\top + L_t^\top R_t L_t z z^\top)$$
$$+ \text{tr}[\Lambda_{t+1}(A x x^\top A^\top + B L_t z z^\top L_t^\top B^\top + A x z^\top L_t^\top B^\top + B L_t z x^\top A^\top + \hat{\Sigma}_t^{xx})]$$
$$+ \text{tr}[\Omega_{t+1}(K_t H x x^\top H^\top K_t^\top + M_t z z^\top M_t^\top + K_t H x z^\top M_t^\top + M_t z x^\top H^\top K_t^\top + \hat{\Sigma}_t^{zz})]$$
$$+ \text{tr}[\Gamma_{t+1}(A x x^\top H^\top K_t^\top + B L_t z z^\top M_t^\top + A x z^\top M_t^\top + B L_t z x^\top H^\top K_t^\top)]$$
$$+ \gamma_{t+1} \,. \tag{36}$$

Grouping terms proportional to $xx^\top$, $xz^\top$ and $zz^\top$ and constant, we find that the cost-to-go can be written as Eq. 35 where the coefficients obey the Lagrange multiplier equations in Eqs. 16 at time $t$. In addition, $\gamma_t$ is computed using Eq. 19.

By induction, then we have that Eq. 35 is true for all $t$ and that the coefficients are indeed the Lagrange multipliers defined in Eqs. 16 and Eq. 19.

### A.2.6 Fixed-Point Equations for Control and Filter Derivatives

The fixed point equations $\partial C_{\mathcal{L}} / \partial L_t = 0$ and $\partial C_{\mathcal{L}} / \partial K_t = 0$ for the extrema of the Lagrangian 8 take the form

$$\frac{\partial C_{\mathcal{L}}}{\partial L_t} = \left[ 2 R_t L_t + B^\top \left( \tilde{\Lambda}_{t+1} B L_t + \tilde{\Omega}_{t+1} M_t + \Gamma_{t+1} B L_t + \Gamma_{t+1}^\top M_t \right) + \sum_i C_i^\top \tilde{\Lambda}_{t+1} C_i L_t \right] S_t^{zz}$$
$$+ B^\top \left[ \tilde{\Lambda}_{t+1} A + \tilde{\Omega}_{t+1} K_t H + \Gamma_{t+1} A + \Gamma_{t+1}^\top K_t H \right] S_t^{xz} = 0 \,, \tag{37}$$
$$\frac{\partial C_{\mathcal{L}}}{\partial K_t} = \left[ \tilde{\Omega}_{t+1} K_t H + \Gamma_{t+1} A \right] S_t^{xx} H^\top - \left[ \tilde{\Omega}_{t+1} M_t + \Gamma_{t+1} B L_t \right] S_t^{zz} H^\top - \tilde{\Omega}_{t+1} K_t H S_t^{xz} H^\top$$
$$+ \tilde{\Omega}_{t+1} M_t (S_t^{xz})^\top H^\top - \Gamma_{t+1} A S_t^{xz} H^\top + \Gamma_{t+1} B L_t (S_t^{xz})^\top H^\top + \tilde{\Omega}_{t+1} K_t \Sigma_\omega$$
$$+ \tilde{\Omega}_{t+1} K_t \sum_i D_i S_t^{xx} D_i^\top = 0 \,, \tag{38}$$

with symmetric matrices $\tilde{\Lambda}_t = \Lambda_t + \Lambda_t^\top$ and $\tilde{\Omega}_t = \Omega_t + \Omega_t^\top$, after using elementary properties of the trace operator and its derivatives.

The fixed point equations can be further manipulated to express $L_t$ and $K_t$ as

$$L_t = E_t^{-1} \left( F_t S_t^{xz} (S_t^{zz})^{-1} + J_t \right) \,,$$

where

$$E_t = 2R_t + B^\top(\tilde{\Lambda}_{t+1} + \tilde{\Omega}_{t+1} + \Gamma_{t+1} + \Gamma_{t+1}^\top)B + \sum_i C_i^\top \tilde{\Lambda}_{t+1} C_i \,,$$

$$F_t = -B^\top(\tilde{\Lambda}_{t+1}A + \tilde{\Omega}_{t+1}K_t H + \Gamma_{t+1}A + \Gamma_{t+1}^\top K_t H) \,,$$

$$J_t = -B^\top(\tilde{\Omega}_{t+1} + \Gamma_{t+1}^\top)(A - K_t H) \,,$$

and

$$K_t = \left( S_{AH} + \tilde{\Omega}_{t+1}^{-1}\Gamma_{t+1}S_{LH} \right) S_{HH}^{-1}$$

with

$$S_{AH} = (A + BL_t)(S_t^{zz} - (S_t^{xz})^\top)H^\top \,,$$

$$S_{LH} = \left( -A(S_t^{xx} - S_t^{xz}) + BL_t(S_t^{zz} - (S_t^{xz})^\top) \right) H^\top \,,$$

$$S_{HH} = H(S_t^{xx} + S_t^{zz} - S_t^{xz} - (S_t^{xz})^\top)H^\top + \Sigma_\omega + \sum_i D_i S_t^{xx} D_i^\top \,.$$

Note that the equation for $L_t$ explicitly depends on $K_t$ on the right side, while the equation for $K_t$ depends on $L_t$ on the right side. This property enables the coordinate-descent algorithm described in the paper. The above expressions coincide with Eqs. 14,15.

### A.2.7 Orthogonality Principle Yields a Critical Point if and Only if Internal Noise Vanishes

**Theorem 2.** *Take the initial condition $p_0(x,z)$ such that $S_0^{zz} = S_0^{xz}$. A solution to the Lagrange equations 13,14,15,16 is given by the orthogonality principle $S_t^{zz} = S_t^{xz}$ for $t = 1, ..., T$, iff internal noise is zero, that is $\Sigma_\eta = 0$. The solution corresponds to a critical point of the cost in Eq. 8*

*Proof.* We first show that (1) assuming OP ($S_t^{xz} = S_t^{zz}$ for $t = 0, ..., T$) is true, we prove that the satisfaction of the Lagrange equations for the multipliers, Eqs. 16, and the equation for the fixed point of $L_t$, Eq. 14, for all $t$ implies that the *Lagrange equality*, $\Gamma_t = -\tilde{\Omega}_t$ for all $t$ ($\tilde{\Omega}_t \equiv \Omega_t + \Omega_t^\top$), is true, regardless of the value of internal noise. Next, we show that (2) OP and the Lagrange equality imply satisfaction of the fixed point equation for $K_t$, Eq. 15, and the 2nd-order moments equations, Eqs. A.2.4, if and only if internal noise is zero, $\Sigma_\eta = 0$. This will show that OP solves all Lagrange equations iff internal noise is zero, and therefore it will correspond to a critical point of the cost function in Eq. 8.

(1) Assume that OP holds. From the boundary condition of the Lagrange equations for the multipliers we have that $\Lambda_{T+1} = \Omega_{T+1} = \Gamma_{T+1} = 0$. Therefore, at time $T + 1$ the Lagrange equality $\Gamma_{T+1} = -\tilde{\Omega}_{T+1}$ is true. Let us prove by induction that the equality holds for all $t$. Assume that the Lagrange equality is true for some $t + 1$, that is, $\Gamma_{t+1} = -\tilde{\Omega}_{t+1}$ (note that $\Gamma_{t+1}$ is then symmetric). Then, from the Lagrange multipliers Eqs. 16 we can write

$$\Gamma_t = L_t^\top B^\top \tilde{\Lambda}_{t+1} A + M_t^\top \tilde{\Omega}_{t+1}K_t H + M_t^\top \Gamma_{t+1}A + L_t^\top B^\top \Gamma_{t+1}^\top K_t H$$

$$= L_t^\top B^\top \tilde{\Lambda}_{t+1} A - M_t^\top \Gamma_{t+1}K_t H + M_t^\top \Gamma_{t+1}A + L_t^\top B^\top \Gamma_{t+1}K_t H$$

$$\tilde{\Omega}_t = 2L_t^\top R_t L_t + L_t^\top B^\top \tilde{\Lambda}_{t+1}BL_t + M_t^\top \tilde{\Omega}_{t+1}M_t + M_t^\top \Gamma_{t+1}BL_t + L_t^\top B^\top \Gamma_{t+1}M_t$$

$$+ \sum_i L_t^\top C_i^\top \tilde{\Lambda}_{t+1}C_i L_t$$

$$= 2L_t^\top R_t L_t + L_t^\top B^\top \tilde{\Lambda}_{t+1}BL_t - M_t^\top \Gamma_{t+1}M_t + M_t^\top \Gamma_{t+1}BL_t + L_t^\top B^\top \Gamma_{t+1}M_t$$

$$+ \sum_i L_t^\top C_i^\top \tilde{\Lambda}_{t+1}C_i L_t \,,$$

where we have replaced $\tilde{\Omega}_{t+1}$ by $-\Gamma_{t+1}$ and using that $\Gamma_{t+1}$ is symmetric. Now, summing we have

$$\Gamma_t + \tilde{\Omega}_t = 2L_t^\top R_t L_t + L_t^\top B^\top \tilde{\Lambda}_{t+1}(A + BL_t) + M_t^\top \Gamma_{t+1}(A + BL_t - K_t H - M_t)$$

$$+ L_t^\top B^\top \Gamma_{t+1}^\top(A + BL_t) + \sum_i L_t^\top C_i^\top \tilde{\Lambda}_{t+1}C_i L_t$$

$$= L_t^\top \left[ 2R_t L_t + B^\top \tilde{\Lambda}_{t+1}(A + BL_t) + B^\top \Gamma_{t+1}^\top(A + BL_t) + \sum_i C_i^\top \tilde{\Lambda}_{t+1}C_i L_t \right] \,,$$

$$(39)$$

where we have realized that the last term in the first line is zero.

Now, the solution for which OP holds should satisfy all other Lagrange equations, in particular the one for the fixed point equation for $L_t$, Eq. 14. As OP is assumed to be true at all times, and in particular at time $t$, and the Lagrange equality is assumed to be true for $t+1$, Eq. 14 (see Sec. A.2.6) largely simplifies to

$$L_t = \bar{E}_t^{-1} \bar{F}_t \,, \tag{40}$$

with $\bar{E}_t = 2R_t + B^\top (\tilde{\Lambda}_{t+1} + \Gamma_{t+1})B + \sum_i C_i^\top \tilde{\Lambda}_{t+1} C_i$ and $\bar{F}_t = -B^\top (\tilde{\Lambda}_{t+1} + \Gamma_{t+1})A$. Then, it is clear that the bracket in the last line of Eq. 39 is zero, and therefore the Lagrange equality $\Gamma_t = -\tilde{\Omega}_t$ is true. Therefore, by induction we conclude that the Lagrange equality is true for all $t$ and that Lagrange equations for the multipliers and $L_t$ are solved. Notice that the above results are true regardless of the presence of internal noise.

(2) Still we have not used the Lagrange equation for $K_t$, Eq. 15, nor the Lagrange equations for the 2nd-order moments, Eqs. 32. These equations must also be satisfied by the OP condition. First, from OP (and the implied Lagrange equality shown in (1)) the expression for $K_t$ (see Sec. A.2.6) largely simplifies to

$$K_t = A\,(S_t^{xx} - S_t^{zz})\,H^\top \bar{S}_{HH}^{-1} \,, \tag{41}$$

with $\bar{S}_{HH} = H(S_t^{xx} - S_t^{zz})H^\top + \Sigma_\omega + \sum_i D_i S_t^{xx} D_i^\top$.

Now, this expression of $K_t$ must solve the Lagrange equations for the 2nd-order moments. The equation for $S_t^{xx}$ is trivially satisfied, but the equations for $S_t^{xz}$ and $S_t^{zz}$ should be such that $S_t^{xz} = S_t^{zz}$ for all $t$ – otherwise, our OP initial assumption would be inconsistent; no other restrictions are imposed by the Lagrange equations of the 2nd-order moments. This is only possible iff the difference $S_{t+1}^{zz} - S_{t+1}^{xz}$ equals zero:

$$S_{t+1}^{zz} - S_{t+1}^{xz} = \left[ -(A - K_t H)(S_t^{xx} - S_t^{zz})H^\top + K_t \Sigma_\omega + K_t \sum_i D_i^\top S_t^{xx} D_i \right] K_t^\top + \Sigma_\eta = 0 \,, \tag{42}$$

for all $t$ (this expression has been obtained using the 2nd-order moments in Eqs. 32 after several cancellations). In this expression, the bracket equals zero after using Eq. 41. Therefore, consistency of OP and satisfaction of the 2nd-order moments are satisfied if and only if internal noise is zero, $\Sigma_\eta = 0$.

This concludes the proof, because iff $\Sigma_\eta = 0$ we have a full satisfaction of all Lagrange equations for all $t$ under the sole assumption of OP for all $t$. $\qquad\square$

### A.2.8 RECOVERY OF CLASSICAL LQAG SOLUTIONS

In this section, we demonstrate that the solutions derived in Sec. A.2 exactly recover the classical analytical solutions of the standard LQAG problem (see Appendix A.2.3) when both multiplicative and internal noise terms vanish. To illustrate this, we examine the solutions presented in Appendix A.2.7. As empirically validated in Damiani et al. (2024), the optimal solutions, when internal noise is absent, satisfy the orthogonality principle (OP). Thus, by setting the multiplicative noise terms to zero, we can directly verify whether these solutions converge to the classic LQAG solutions. Additionally, this provides a proof that the orthogonality principle indeed corresponds to the global optimum of the cost function for the standard LQAG problem.

The optimal controller derived under the orthogonality principle in Appendix A.2.7 is given by Eq. 40. When both multiplicative and internal noise terms are turned off, we obtain

$$L_t = -[2R_t + B^\top (\tilde{\Lambda}_{t+1} + \Gamma_{t+1})B]^{-1}[B^\top (\tilde{\Lambda}_{t+1} + \Gamma_{t+1})A] \,, \tag{43}$$

which corresponds to the optimal $L_t$ for the classic LQAG case (see solutions in Sec. A.2.3) if $S_t = (\Gamma_t + \tilde{\Lambda}_t)$. Using Eq.16 and imposing the OP (setting $\Gamma_t = -\tilde{\Omega}_t$ – see Appendix A.2.7) we obtain

$$\Gamma_{t+1} + \tilde{\Lambda}_{t+1} = 2Q_t + (A + BL_t)^\top (\tilde{\Lambda}_t + \Gamma_t)A \,. \tag{44}$$

Now we observe, as discussed in Appendix A.2.7, that $\Gamma_t$ is symmetric and the same holds for $\tilde{\Lambda}_t$ (by definition), therefore we can rewrite Eq. 44 as

$$\Gamma_{t+1} + \tilde{\Lambda}_{t+1} = 2Q_t + A^\top (\tilde{\Lambda}_t + \Gamma_t)(A + BL_t) \,. \tag{45}$$

which corresponds to the formula for $S_t$ in Sec. A.2.3, therefore proving the equality between the two optimal solutions.

The optimal Kalman filter derived under the OP in Appendix A.2.7 is given by Eq. 41, corresponding to

$$K_t = A\left(S_t^{xx} - S_t^{zz}\right)H^\top[H(S_t^{xx} - S_t^{zz})H^\top + \Sigma_\omega]^{-1} \,, \tag{46}$$

when neither internal nor multiplicative noise is considered. We note that this solution corresponds to the one presented in Sec. A.2.3 when $\Sigma_t^e = S_t^{xx} - S_t^{zz}$, which is automatically satisfied when the OP, stating $S_t^{zz} = S_t^{xz}$, holds.

Therefore, the solutions derived in Appendix A.2.7 correspond to the globally optimal solutions of the classic LQAG problem in the absence of multiplicative and internal noise.

### A.2.9 JOINT OPTIMIZATION OF FORWARD DYNAMICS, PSEUDO-FILTER, AND CONTROL WITH MODEL MISMATCH: FULL DERIVATIONS

**Model and Moments** The Model Mismatch approach is defined by the equations

$$x_{t+1} = Ax_t + BL_t z_t + n_t^x \,, \quad y_t = Hx_t + n_t^y \,, \quad z_{t+1} = W_t z_t + P_t y_t + n_t^z \tag{47}$$
$$n_t^c = \epsilon_t^c + \sum_r \eta_t^c U_r^c x_t + \sum_l \xi_t^c V_l^c L_t z_t \,, \quad c \in \{x, y, z\} \,,$$

identical to Eqs. 11. The goal is to optimize the forward dynamics $W_t \in \mathbb{R}^{n \times n}$, pseudo-filter $P_t \in \mathbb{R}^{n \times m}$ and control $L_t \in \mathbb{R}^{p \times n}$ – where $p$ is the dimensionality of the control signal $u_t = L_t z_t$- matrices so as to minimize the expected cumulative quadratic cost

$$C = \sum_{t=0}^{T} \mathbb{E}\left[x_t^\top Q_t x_t + z_t^\top L_t^\top R_t L_t z_t\right] = \sum_{t=0}^{T} \left(\text{tr}(Q_t S_t^{xx}) + \text{tr}(L_t^\top R_t L_t S_t^{zz})\right) \,, \tag{48}$$

with initial condition $p_0(x, z)$.

Eqs. 47 can be put in a more compact form as

$$x_{t+1} = Ax_t + BL_t z_t + n_t^x \tag{49}$$
$$z_{t+1} = W_t z_t + P_t Hx_t + P_t n_t^y + n_t^z$$
$$n_t^c = \epsilon_t^c + \sum_r \eta_t^c U_r^c x_t + \sum_l \xi_t^c V_l^c L_t z_t \,, \quad c \in \{x, y, z\} \,,$$

from where it is more obvious that the system consists of two coupled linear dynamical systems with free parameters $W_t$, $P_t$ and $L_t$ chosen so as the minimize the cost. The sums $\sum_r$ and $\sum_l$ can run over different limits depending on the source $c$, but here we use the same symbol to avoid cluttered notation.

Note that the Model Mismatch framework is strictly more general than the Model Match one because one always is free to choose in Eqs. 49 $P_t = K_t$ and $W_t = A + BL_t - K_t H$, leading exactly to the Model Match approach in Eqs. 5,6,7. The reverse, mapping the Model Mismatch approach into the Model Match one, is in general not possible.

The 2nd-order moments, appearing in the cost 48, obey

$$S_{t+1}^{xx} = AS_t^{xx}A^\top + BL_t S_t^{zz} L_t^\top B^\top + AS_t^{xz} L_t^\top B^\top + BL_t(S_t^{xz})^\top A^\top + \Sigma_t^x$$
$$S_{t+1}^{zz} = P_t HS_t^{xx}H^\top P_t^\top + W_t S_t^{zz} W_t^\top + P_t HS_t^{xz} W_t^\top + W_t(S_t^{xz})^\top H^\top P_t^\top + P_t \Sigma_t^y P_t^\top + \Sigma_t^z$$
$$S_{t+1}^{xz} = AS_t^{xx}H^\top P_t^\top + BL_t S_t^{zz} W_t^\top + AS_t^{xz} W_t^\top + BL_t(S_t^{xz})^\top H^\top P_t^\top \,, \tag{50}$$

with $\Sigma_t^c = \Sigma_{\epsilon^c} + \sum_r U_r^c S_t^{xx}(U_r^c)^\top + \sum_l V_l^c L_t S_t^{zz} L_t^\top (V_l^c)^\top, c \in \{z, y, x\}$.

Even though the Model Mismatch approach is more general than the Model Match one, defined in Eqs. 5,6,7, it is already apparent that the equations for the second moments are simpler, more compact and transparent. This will be a recurrent theme in all next derivations and equations, so we will not repeat this below.

**Total Cost and Cost-to-Go** Let us define the cost-to-go at time $t$ starting from $x$ and $z$ as $C_t(x,z) = \text{tr}(Q_t xx^\top + L_t^\top R_t L_t zz^\top) + \sum_{\tau=t+1}^{T} \mathbb{E}\left[x_\tau^\top Q_\tau x_\tau + z_\tau^\top L_t^\top R_\tau L_t z_\tau\right]$, where the expectation is over the noises with initial conditions fixed at $x$ and $z$ at time $t$, and for specific $P$, $L$ and $W$ from time $t$ onward. The cost-to-go obeys the Bellman equation

$$C_t(x,z) = \text{tr}(Q_t xx^\top) + \text{tr}(L_t^\top R_t L_t zz^\top) + \int dx' dz' C_{t+1}(x',z') p_{x,t+1}(x'|x,z) p_{z,t+1}(z'|x,z) , \tag{51}$$

where the transition probability densities $p_{x,t+1}(x'|x,z)$ and $p_{z,t+1}(z'|x,z)$ are the transition probability functions over $x'$ and $z'$ at time $t+1$ when starting from $x$ and $z$ at time $t$, as defined by equations 47. Using backwards induction, and following similar steps to those in Secs. A.2.4 and A.2.5, it is not difficult to show that the cost-to-go can be written for all $t$ ($t = 0, ..., T$) as

$$C_t(x,z) = \text{tr}(\Lambda_t xx^\top) + \text{tr}(\Omega_t zz^\top) + \text{tr}(\Gamma_t xz^\top) + \gamma_t , \tag{52}$$

with matrices $\Lambda_t \in \mathbb{R}^{m \times m}$, $\Omega_t \in \mathbb{R}^{n \times n}$, and $\Gamma_t \in \mathbb{R}^{n \times m}$ and scalar $\gamma_t$ obeying equations

$$\Lambda_t = Q_t + A^\top \Lambda_{t+1} A + H^\top P_t^\top \Omega_{t+1} P_t H + H^\top P_t^\top \Gamma_{t+1} A$$
$$+ \sum_r (U_r^x)^\top \Lambda_{t+1} U_r^x + \sum_r (U_r^y)^\top P_t^\top \Omega_{t+1} P_t U_r^y + \sum_r (U_r^z)^\top \Omega_{t+1} U_r^z ,$$
$$\Omega_t = L_t^\top R_t L_t + L_t^\top B^\top \Lambda_{t+1} B L_t + W_t^\top \Omega_{t+1} W_t + W_t^\top \Gamma_{t+1} B L_t$$
$$+ \sum_r L_t^\top (V_r^x)^\top \Lambda_{t+1} V_r^x L_t + \sum_r L_t^\top (V_r^y)^\top P_t^\top \Omega_{t+1} P_t V_r^y L_t + \sum_r L_t^\top (V_r^z)^\top \Omega_{t+1} V_r^z L_t ,$$
$$\Gamma_t = L_t^\top B^\top (\Lambda_{t+1} + \Lambda_{t+1}^\top) A + W_t^\top (\Omega_{t+1} + \Omega_{t+1}^\top) P_t H + W_t^\top \Gamma_{t+1} A + L_t^\top B^\top \Gamma_{t+1}^\top P_t H ,$$
$$\gamma_t = \text{tr}(\Lambda_{t+1} \Sigma_{\epsilon^x}) + \text{tr}(P_t^\top \Omega_{t+1} P_t \Sigma_{\epsilon^y}) + \text{tr}(\Omega_{t+1} \Sigma_{\epsilon^z}) + \gamma_{t+1} , \tag{53}$$

with boundary conditions $\Lambda_T = Q_T$, $\Omega_T = L_T^\top R_T L_T$, $\Gamma_T = 0$ and $\gamma_T = 0$ (in this way the boundary condition that $C_T(x,z) = \text{tr}(Q_T xx^\top) + \text{tr}(L_T^\top R_T L_T zz^\top)$ is satisfied).

We now define the averaged cost-to-go at time $t$ as

$$C_t \equiv \int dx dz p_t(x,z) C_t(x,z) = \text{tr}(\Lambda_t S_t^{xx}) + \text{tr}(\Omega_t S_t^{zz}) + \text{tr}(\Gamma_t S_t^{xz}) + \gamma_t , \tag{54}$$

where $p_t(x,z)$ is the joint probability density over $x$ and $z$ given initial condition $p_0(x,z)$ and $W_\tau$, $L_\tau$, and $P_\tau$ for $\tau < t$. We note that the total cost $C$ in Eq. 48 can be written as

$$C = C_0 \equiv \int dx dz p_0(x,z) C_0(x,z) = \text{tr}(\Lambda_0 S_0^{xx}) + \text{tr}(\Omega_0 S_0^{zz}) + \text{tr}(\Gamma_0 S_t^{xz}) + \gamma_0 , \tag{55}$$

which it can also be expressed as

$$C = C_{<t} + C_t \tag{56}$$

with $C_{<t} = \sum_{\tau=0}^{t-1} \text{tr}(Q_\tau S_\tau^{xx} + L_\tau^\top R_\tau L_\tau S_\tau^{zz})$. It is important to note that Eq. 56 is valid for all $t$.

**Algorithm** Building an algorithm to find an improved triplet of time-dependent forward dynamics, pseudo-filter and control matrices is slightly simpler than in the case of the Model Match approach because $W_t$ and $P_t$ only appear in the internal variable dynamical equation and $L_t$ only appears in the state variable dynamics. In contrast, in the Model Match approach, $L_t$ appeared both in the state and state estimate dynamics, complicating the mathematical derivations.

Indeed, we note from Eqs. 53 that the coefficients $\Lambda_t$, $\Omega_t$, $\Gamma_t$ and $\gamma_t$ depend on $W_\tau$, $P_\tau$ and $L_\tau$ only for $\tau \geq t$, while $S_t^{ab}$, $ab \in \{xx, zz, xz\}$, only depend on those matrices for $\tau < t$, as it can be seen from Eqs. 50. Therefore, choosing an arbitrary $t$, in Eq. 56 only the term $C_t$ depends on $W_t$, and in that term, Eq. 54, only the coefficients $\Lambda_t$, $\Omega_t$, $\Gamma_t$ and $\gamma_t$ can depend on $W_t$. In conclusion, starting with a set of $W_{0,...,T}$, $P_{0,...,T}$ and $L_{0,...,T}$, we can improve the value of $W_t$ as

$$W_t^* = \arg\min_{W_t} C = \arg\min_{W_t} C_t , \tag{57}$$

while keeping the $W_\tau$ for $\tau \neq t$ and all $P_{0,...,T}$ and $L_{0,...,T}$ fixed. A global minimum exists because $C_t$ is always non-negative. Using elementary matrix operations, we find that

$$W_t^* = -P_t H S_t^{xz}(S_t^{zz})^{-1} - (\Omega_{t+1} + \Omega_{t+1}^\top)^{-1} \Gamma_{t+1}\left(B L_t + A S_t^{xz}(S_t^{zz})^{-1}\right) . \tag{58}$$

Note that if $S_0^{zz}$ is not invertible, then $W_0^*$ is not well defined, and thus we can take any arbitrary matrix. This might correspond to $z_0 = 0$. After the optimization, we must have

$$C^* = C(W_0, ..., W_t^*, ..., W_t) \le C(W_0, ..., W_t, ..., W_T) , \tag{59}$$

so that the total cost is non-increasing. After optimizing $W_t$, using the new $W_t^*$, the cost can be written as

$$C^* = C_{<t+1} + C_{t+1}^* = C_{<t+1} + \text{tr}(\Lambda_{t+1} S_{t+1}^{xx,*}) + \text{tr}(\Omega_{t+1} S_{t+1}^{zz,*}) + \text{tr}(\Gamma_{t+1} S_{t+1}^{xz,*}) + \gamma_{t+1} \tag{60}$$

where the coefficients at time $t + 1$ do not need to be updated (as they do not depend on $W_t^*$), but where the $S_{t+1}^{ab,*}$ need to be updated using Eqs. 50 with the new $W_t^*$.

Redefining $W_t^*$ as $W_t$ and the $S_{t+1}^{ab,*}$ as $S_{t+1}^{ab}$, we can now proceed to optimize $W_{t+1}$ using the same procedure as above (changing $t$ to $t + 1$) to minimize again the total cost $C(W_0, ..., W_t, W_{t+1}^*, ..., W_T) \le C(W_0, ..., W_t, W_{t+1}, ..., W_T)$ fixing $P_{0,...,T}$, $L_{0,...,T}$ and all $W_\tau$ except for $\tau = t$. This procedure can be repeated consecutively from $t = 0$ up to $T$.

After this forward pass, we would like to repeat the process for $P_t$ and $L_t$ instead of $W_t$. But before doing this, the value of the coefficients in Eqs. 53 have to be recomputed so that Eq. 55 is true again. The process of forward updating the $W_t$ from $t = 0$ up to time $T$ and, after this, recomputing the coefficients using a backwards pass is called $W$-pass. Note that in this process, the moments have been already recomputed. Starting from $W^{(n)} = W_{0,...,T}^{(n)}$, $P^{(n)} = P_{0,...,T}^{(n)}$ and $L^{(n)} = L_{0,...,T}^{(n)}$, the $W$-pass leads to a new set of forward dynamics matrices $W^{(n+1)}$ such that the cost is non-increasing, $C(W^{(n+1)}, P^{(n)}, L^{(n)}) \le C(W^{(n)}, P^{(n)}, L^{(n)})$. We define a $P$-pass as that consisting in exactly repeating the same procedure for the $P_{0,...,T}$ instead of the $W_{0,...,T}$ while keeping fixed $W_{0,...,T}$ and $L_{0,...,T}$, and using the expression (obtained after some calculations)

$$P_t^* = -\left[ W_t (S_t^{xz})^\top + (\Omega_{t+1} + \Omega_{t+1}^\top)^{-1} \Gamma_{t+1} \left( A S_t^{xx} + B L_t (S_t^{xz})^\top \right) \right] H^\top E_t^{-1} , \tag{61}$$

with $E_t = H S_t^{xx} H^\top + \sum_l U_l^y S_t^{xx} (U_l^y)^\top + \sum_r V_l^y L_t S_t^{zz} L_t^\top (V_l^y)^\top + \Sigma_{\epsilon^y}$. Starting from $W^{(n+1)} = W_{0,...,T}^{(n+1)}$, $P^{(n)} = P_{0,...,T}^{(n)}$ and $L^{(n)} = L_{0,...,T}^{(n)}$, the $P$-pass leads to a new set of pseudo-filter matrices $P^{(n+1)}$ such that the cost is non-increasing, $C(W^{(n+1)}, P^{(n+1)}, L^{(n)}) \le C(W^{(n+1)}, P^{(n)}, L^{(n)})$. Finally, we define an $L$-pass as that consisting in following similar steps to the previous ones to sequentially update the $L_{0,...,T}$ while keeping fixed $W_{0,...,T}$ and $P_{0,...,T}$, and using the expression (after some calculations)

$$L_t^* = -F_t^{-1} B^\top \left\{ \tilde{\Lambda}_{t+1} A S_t^{xz} (S_t^{zz})^{-1} + \Gamma_{t+1}^\top \left[ P_t H S_t^{xz} (S_t^{zz})^{-1} + W_t \right] \right\} , \tag{62}$$

with $F_t = 2R_t + B^\top \tilde{\Lambda}_{t+1} B + \sum_l (V_l^x)^\top \tilde{\Lambda}_{t+1} V_l^x + \sum_l (V_l^y)^\top P_t^\top \tilde{\Omega}_{t+1} P_t V_l^y + \sum_l (V_l^z)^\top \tilde{\Omega}_{t+1} V_l^x$, where we have defined $\tilde{\Lambda}_t = \Lambda_t + \Lambda_t^\top$ and $\tilde{\Omega}_t = \Omega_t + \Omega_t^\top$. Starting from $W^{(n+1)} = W_{0,...,T}^{(n+1)}$, $P^{(n+1)} = P_{0,...,T}^{(n+1)}$ and $L^{(n)} = L_{0,...,T}^{(n)}$, the $L$-pass leads to a new set of control matrices $L^{(n+1)}$ such that the cost is non-increasing, $C(W^{(n+1)}, P^{(n+1)}, L^{(n+1)}) \le C(W^{(n+1)}, P^{(n+1)}, L^{(n)})$.

Now, alternating $W$-, $P$- and $L$-passes from some initial arbitrary values $W^{(0)}, P^{(0)}, L^{(0)}$ we find

$$C(W^{(0)}, P^{(0)}, L^{(0)}) \ge C(W^{(1)}, P^{(0)}, L^{(0)}) \ge C(W^{(1)}, P^{(1)}, L^{(0)}) \ge ...$$
$$\ge C(W^{(n+1)}, P^{(n)}, L^{(m)}) \ge C(W^{(n+1)}, P^{(n+1)}, L^{(n)})$$
$$\ge C(W^{(n+1)}, P^{(n+1)}, L^{(n+1)}) \ge ... \ge C_{min} \ge 0 . \tag{63}$$

Since the series is non-negative, it converges to a total cost (not larger than the initial one) with optimal forward dynamics $W^* = W^{(\infty)}$, pseudo-filter $P^* = P^{(\infty)}$ and control $L^* = L^{(\infty)}$ matrices. We have thus proven the first part of the following

**Theorem 3.** *Starting with arbitrary $W^{(0)}$, $P^{(0)}$ and $L^{(0)}$ and distribution of initial conditions $p_0(x, z)$, the coordinate descent algorithm defined by iterating in alternation $W$-, $P$- and $L$-passes converges to an improved triplet of forward dynamics, pseudo-filter and control matrices $W^*$, $P^*$ and $L^*$. The improved triplet corresponds to a critical point of the cost function in Eq. 48.*

We remark that it is straightforward to extend our algorithm to the case where any of the matrices $W_t$, $P_t$ and $L_t$ are fixed simply by not updating the corresponding matrices using the above passes, still enjoying convergence properties.

**Lagrangian, Fixed-Point Equations, and Critical Points**   To complete the last part of the theorem, that is, that after convergence the triplet $W^*$, $P^*$ and $L^*$ is a critical point of the cost function 48, we must show that they solve all fixed points equations of the Lagrangian,

$$C_{\mathcal{L}} = \sum_{t=0}^{T} \left( \mathrm{tr}(Q_t S_t^{xx}) + \mathrm{tr}(R_t S_t^{zz}) \right) - \sum_{t=1}^{T+1} \left( \mathrm{tr}(\Lambda_t G_t^{xx}) + \mathrm{tr}(\Omega_t G_t^{zz}) + \mathrm{tr}(\Gamma_t G_t^{xz}) \right) , \quad (64)$$

where $\Lambda_t$, $\Omega_t$ and $\Gamma_t$ are matrices of Lagrange multipliers. The constraints $G_t^{xx} = G_t^{zz} = G_t^{xz} = 0$ are given by the temporal evolution of $S_t^{xx}$, $S_t^{zz}$ and $S_t^{xz}$, respectively, between two consecutive time steps $t$ and $t+1$, and can be computed using Eqs. 50 similarly as in Eqs. 13. Indeed, the fixed point equations of the Lagrangian $\partial C_{\mathcal{L}}/\partial W_t = 0$ and $\partial C_{\mathcal{L}}/\partial P_t = 0$ are identical to Eqs. 58,61,62, respectively, which must be satisfied after convergence by the improved triplet $W^*$, $P^*$ and $L^*$. After some work, the Lagrange equations $\partial C_{\mathcal{L}}/\partial S_t^{xx} = 0$, $\partial C_{\mathcal{L}}/\partial S_t^{xx} = 0$ and $\partial C_{\mathcal{L}}/\partial S_t^{xx} = 0$ can be seen to lead exactly to the coefficient Eqs. 53, which, again, are satisfied by the improved triplet. Finally, the derivatives of the Lagrangian with respect to the multipliers reduce to the second-order moment Eqs. 50, which are satisfied by the improved triplet. Thus, the improved triplet is a fixed-point solution of the Lagrangian 64 and therefore a critical point of the cost function 48.

### A.3   Algorithms Implementation: Pseudocodes

#### A.3.1   Pseudocode – Model Match framework

---

**Algorithm 1** Model Match (M-Match) approach

---

    **Input:** $S_0^{xx}, S_0^{xz}, S_0^{zz}$; initial guesses $L_{0,\ldots,T}^{(0)}, K_{0,\ldots,T}^{(0)}$; system parameters.

2: **Output:** Optimal gains $L_{0,\ldots,T}^*, K_{0,\ldots,T}^*$.

    **Steps:**

4: **for** each iteration $k = 1, \ldots,$ optimization steps **do**

      $\Lambda_{1,\ldots,T}, \Omega_{1,\ldots,T}, \Gamma_{1,\ldots,T} \leftarrow$ Eqs. 16 using $L_{0,\ldots,T}^{(k-1)}$ and $K_{0,\ldots,T}^{(k-1)}$ (backward equations)

6:     **for** each iteration $t = 0, \ldots, T-1$ **do**

        $L_t^{(k)} \leftarrow$ Eq. 14,

8:       $S_{t+1}^{xx}, S_{t+1}^{xz}, S_{t+1}^{zz} \leftarrow$ Eqs. 32 using $L_t^{(k)}$ and $K_t^{(k-1)}$

    **end for**

10:   $\Lambda_{1,\ldots,T}, \Omega_{1,\ldots,T}, \Gamma_{1,\ldots,T} \leftarrow$ Eqs. 16 using $L_{0,\ldots,T}^{(k)}$ and $K_{0,\ldots,T}^{(k-1)}$ (backward equations)

    **for** each iteration $t = 0, \ldots, T-1$ **do**

12:     $K_t^{(k)} \leftarrow$ Eq. 15,

      $S_{t+1}^{xx}, S_{t+1}^{xz}, S_{t+1}^{zz} \leftarrow$ Eqs. 32 using $L_t^{(k)}$ and $K_t^{(k)}$

14:   **end for**

    **end for**

16: $L_{0,\ldots,T}^* \leftarrow L_{0,\ldots,T}^{(k)}$; $K_{0,\ldots,T}^* \leftarrow K_{0,\ldots,T}^{(k)}$

---

The pseudocode above implements the algorithm of Sec. A.2.2, referred to as the Model Match (M-Match) approach, in contrast to the Model Mismatch (M-Mis) method of Sec. 4.

### A.3.2 PSEUDOCODE – MODEL MISMATCH FRAMEWORK

---

**Algorithm 2** Model Mismatch (M-Mis) approach

---

**Input:** $S_0^{xx}, S_0^{xz}, S_0^{zz}$; initial guesses $L_{0,\dots,T}^{(0)}, P_{1,\dots,T}^{(0)}, W_{1,\dots,T}^{(0)}$; system parameters.

2: **Output:** Optimal matrices $L_{0,\dots,T}^*, P_{1,\dots,T}^*, W_{1,\dots,T}^*$.

    **Steps:**

4: **for** each iteration $k = 1, \dots$, optimization steps **do**

    $\Lambda_{1,\dots,T}, \Omega_{1,\dots,T}, \Gamma_{1,\dots,T} \leftarrow$ Eqs. 53 using $P_{1,\dots,T}^{(k-1)}, W_{1,\dots,T}^{(k-1)}$ and $L_{0,\dots,T}^{(k-1)}$ (backward equations)

6:     **for** each iteration $t = 0, \dots, T-1$ **do**

      $P_t^{(k)} \leftarrow$ Eq. 61,

8:       $S_{t+1}^{xx}, S_{t+1}^{xz}, S_{t+1}^{zz} \leftarrow$ Eqs. 50 using $P_t^{(k)}, W_t^{(k-1)}$ and $L_t^{(k-1)}$

    **end for**

10:   $\Lambda_{1,\dots,T}, \Omega_{1,\dots,T}, \Gamma_{1,\dots,T} \leftarrow$ Eqs. 53 using $P_{1,\dots,T}^{(k)}, W_{1,\dots,T}^{(k-1)}$ and $L_{0,\dots,T}^{(k-1)}$ (backward equations)

    **for** each iteration $t = 0, \dots, T-1$ **do**

12:     $W_t^{(k)} \leftarrow$ Eq. 58,

      $S_{t+1}^{xx}, S_{t+1}^{xz}, S_{t+1}^{zz} \leftarrow$ Eqs. 50 using $P_t^{(k)}, W_t^{(k)}$ and $L_t^{(k-1)}$

14:   **end for**

    $\Lambda_{1,\dots,T}, \Omega_{1,\dots,T}, \Gamma_{1,\dots,T} \leftarrow$ Eqs. 53 using $P_{1,\dots,T}^{(k)}, W_{1,\dots,T}^{(k)}$ and $L_{0,\dots,T}^{(k-1)}$ (backward equations)

16:   **for** each iteration $t = 0, \dots, T-1$ **do**

      $L_t^{(k)} \leftarrow$ Eq. 62,

18:     $S_{t+1}^{xx}, S_{t+1}^{xz}, S_{t+1}^{zz} \leftarrow$ Eqs. 50 using $P_t^{(k)}, W_t^{(k)}$ and $L_t^{(k)}$

    **end for**

20: **end for**

    $P_{1,\dots,T}^* \leftarrow P_{1,\dots,T}^{(k)}; W_{1,\dots,T}^* \leftarrow W_{1,\dots,T}^{(k)}; L_{0,\dots,T}^* \leftarrow L_{0,\dots,T}^{(k)}$

---

The pseudocode above outlines the Model Mismatch (M-Mis) approach, introduced in Sec. 4 and detailed in Appendix A.2.9. While the order of optimization for $P$, $W$, and $L$ differs from that in Appendix A.2.9, all variants converge to a critical point of the cost function in Eq. 48.

### A.3.3 IMPLEMENTATIONS DETAILS

Here we report the algorithms' hyper-parameters, as selected for the experiments described in Sec. A.4.

For the single-joint reaching task used to evaluate Algorithm 1 – and to compare it with the gradient-based numerical method from Damiani et al. (2024) (referred to as GD) – we use the parameters listed in Table 1. Note that, in line with Damiani et al. (2024), the GD algorithm is implemented using the GradientDescent() function from the Optim.jl Julia package.

Table 1: Hyper-parameters of the algorithms used in the single-joint reaching task (Sec. A.4.1)

| Algorithm | Description | value |
|---|---|---|
| GD (Damiani et al., 2024) | Number of iterations of the "GradientDescent()" function | 50000 |
| M-Match (Algorithm 1) | Number of iterations of the estimation-control optimization | 100 |

For the 3D reaching task, detailed in Appendix A.4.3 and for the Redundant Arm-Control Task, detailed in Appendix A.4.5, we used

Table 2: Hyper-parameters of the algorithms used in the 3D reaching task and in the Redundant Arm-Control Task(Appendices A.4.3 and A.4.5)

| Algorithm | Description | value |
|---|---|---|
| TOD (Todorov, 2005) | Number of iterations of the estimation-control optimization | 100 |
| M-Match (Algorithm 1) | Number of iterations of the estimation-control optimization | 100 |
| M-Mis (Algorithm 2) | Number of iterations of the M-Mis optimization | 100 |

while for the neural population steering task of Appendix A.4.6 we selected the following hyper-parameters

Table 3: Hyper-parameters of the algorithm used in the neural population steering task (Appendix A.4.6)

| Algorithm | Description | value |
|---|---|---|
| M-Mis (Algorithm 2) | Number of iterations of the $L_{0,\dots,T}$ optimization | 20 |

## A.4 EXPERIMENTAL DETAILS AND SUPPLEMENTARY RESULTS

### A.4.1 SINGLE-JOINT REACHING TASK: MODEL AND PARAMETERS

In Sec. 5.1 we evaluated the M-Match algorithm – Algorithm 1 – on a single-joint reaching task, using the same problem formulation as in (Todorov, 2005; Damiani et al., 2024). The system features a four-dimensional state and one-dimensional control and sensory feedback, i.e., $m = 4, p = k = 1$. The discrete-time dynamics is given by Todorov (2005),

$$p(t + \Delta t) = p(t) + \dot{p}(t)\Delta t$$
$$\dot{p}(t + \Delta t) = \dot{p}(t) + f(t)\Delta t/m$$
$$f(t + \Delta t) = f(t)(1 - \Delta t/\tau_2) + g(t)\Delta t/\tau_2$$
$$g(t + \Delta t) = g(t)(1 - \Delta t/\tau_1) + u(t)(1 + \sigma_\varepsilon \varepsilon_t)\Delta t/\tau_1$$

with

$$A = \begin{pmatrix} 1 & \Delta t & 0 & 0 \\ 0 & 1 & \Delta t/m & 0 \\ 0 & 0 & 1 - \Delta t/\tau_2 & \Delta t/\tau_2 \\ 0 & 0 & 0 & 1 - \Delta t/\tau_1 \end{pmatrix}$$

$$B = \begin{pmatrix} 0 & 0 & 0 & \Delta t/\tau_1 \end{pmatrix}^\top$$

$$C = \begin{pmatrix} 0 & 0 & 0 & \sigma_\varepsilon \Delta t/\tau_1 \end{pmatrix}^\top$$

$$H = \begin{pmatrix} 1 & 0 & 0 & 0 \\ 0 & 0 & 0 & 0 \\ 0 & 0 & 0 & 0 \\ 0 & 0 & 0 & 0 \end{pmatrix}$$

$$D = \begin{pmatrix} \sigma_\rho & 0 & 0 & 0 \\ 0 & 0 & 0 & 0 \\ 0 & 0 & 0 & 0 \\ 0 & 0 & 0 & 0 \end{pmatrix}$$

$$Q_{1,\dots,T-1} = \begin{pmatrix} 0 & 0 & 0 & 0 \\ 0 & 0 & 0 & 0 \\ 0 & 0 & 0 & 0 \\ 0 & 0 & 0 & 0 \end{pmatrix}$$

$$Q_T = \vec{p}\vec{p}^\top + \vec{v}\vec{v}^\top + \vec{f}\vec{f}^\top$$

$$R_{1,\cdots,T-1} = \frac{r}{T-1}$$

$$R_T = 0$$

$$\vec{p} = \begin{pmatrix} 1 & 0 & 0 & 0 \end{pmatrix}$$

$$\vec{v} = \begin{pmatrix} 0 & w_v & 0 & 0 \end{pmatrix}$$

$$\vec{f} = \begin{pmatrix} 0 & 0 & w_v & 0 \end{pmatrix}$$

$$\Sigma_\xi = \begin{pmatrix} \sigma_\xi^2 & 0 & 0 & 0 \\ 0 & 0 & 0 & 0 \\ 0 & 0 & 0 & 0 \\ 0 & 0 & 0 & 0 \end{pmatrix}$$

$$\Sigma_\omega = \sigma_\omega^2$$

$$\Sigma_\eta = \begin{pmatrix} \sigma_\eta^2 & 0 & 0 & 0 \\ 0 & \sigma_\eta^2 & 0 & 0 \\ 0 & 0 & \sigma_\eta^2 & 0 \\ 0 & 0 & 0 & \sigma_\eta^2 \end{pmatrix}$$

with the initial conditions given by

$$\mathbb{E}[x_1] = \begin{pmatrix} x_1 & 0 & 0 & 0 \end{pmatrix}^\top$$
$$\mathbb{E}[z_1] = \mathbb{E}[x_1]$$

$$\Sigma_{x_1} = \begin{pmatrix} \sigma_x^2 & 0 & 0 & 0 \\ 0 & 0 & 0 & 0 \\ 0 & 0 & 0 & 0 \\ 0 & 0 & 0 & 0 \end{pmatrix}$$

$$\Sigma_{z_1} = \begin{pmatrix} 0 & 0 & 0 & 0 \\ 0 & 0 & 0 & 0 \\ 0 & 0 & 0 & 0 \\ 0 & 0 & 0 & 0 \end{pmatrix}.$$

The parameters of the problem are listed in Table 4 (std = standard deviation).

Table 4: Parameters of the single-joint reaching task

| Name | Description | Value |
|---|---|---|
| $\Delta t$ | time-step ($s$) | 0.010 |
| $m$ | mass of the hand ($Kg$) | 1 |
| $\tau_1$ | first time constant of the second order low pass filter | 0.04 |
| $\tau_2$ | second time constant of the second order low pass filter | 0.04 |
| $r$ | Auxiliary variable for control-dependent cost | $1e^{-5}$ |
| $w_v$ | Auxiliary variable for task-related cost | 0.2 |
| $w_f$ | Auxiliary variable for task-related cost | 0.01 |
| $T$ | time steps | 100 |
| $x_1$ | Target position | 0.15 |
| $\sigma_x$ | Target position standard deviation | 0.0 |
| $\sigma_\xi$ | std of dynamics noise $\xi_t$ | 0.1 |
| $\sigma_\omega$ | std of the sensory noise $\omega_t$ | 0.1 |
| $\sigma_\varepsilon$ | std of the control-dependent noise $\varepsilon_t$ | 0.5 |
| $\sigma_\rho$ | std of the sensory-dependent noise $\rho$ | 0.5 |
| $\sigma_\eta$ | std of the additive internal noise $\eta_t$ | 0.1 |

### A.4.2 COMPUTATIONAL EFFICIENCY AND DIMENSIONALITY SCALING: COMPARISON WITH PRIOR WORK

As additional evidence for computational efficiency of Algorithm 1, we present a dimensionality-scaling study comparing computation times with the numerical algorithm in Damiani et al. (2024), extending the analysis up to $m = 100$. This complements the results in Sec. 5.1, which already demonstrates a pronounced gap in runtime (6 s vs. 5 h).

To isolate the effect of dimensionality, we set $m = k = p = n_{\text{shared}}$. Matrices $A$, $B$, $C$, and $D$ are drawn from zero-mean, unit-variance Gaussian distributions and rescaled to ensure spectral radius $< 1$ for stability. We fix $T = 6$ and $\sigma_\xi = \sigma_\omega = \sigma_\rho = \sigma_\epsilon = \sigma_\eta = 0.2$, and vary $n_{\text{shared}} \in \{5, 10, 15, 40, 100\}$. We then compare the total computation time of our method (Algorithm 1) with the numerical approach in Damiani et al. (2024), initializing both with optimal gains from Todorov (2005) to ensure a fair comparison. All results were obtained on a MacBook Pro (Apple M1, 16 GB RAM).

Table 5: Comparison of runtime between this work and the numerical algorithm in Damiani et al. (2024) as a function of the number of shared dimensions $n_{\text{shared}}$.

| $n_{\text{shared}}$ | **This work** | **GD (Damiani et al., 2024)** |
|---|---|---|
| 5 | 1.15 s | 8.4 min |
| 10 | 1.25 s | 75.7 min |
| 15 | 1.40 s | 6.4 h |
| 40 | 2.7 s | > 2 days |
| 100 | 14 s | – |

Here, s = seconds, min = minutes, and h = hours. These results highlight the scalability of our method. Similar time gaps also emerge in lower-dimensional settings as trial duration $T$ increases, due to the linear growth in optimization parameters with $T$.

This computational advantage is critical for applying stochastic optimal control to real-world problems, particularly in Inverse Optimal Control (Schultheis et al., 2021; Straub & Rothkopf, 2022), which requires solving many control problems across parameter settings. The high cost of Damiani et al. (2024) renders it impractical for realistic tasks such as that in Sec. A.4.1, first described in Todorov (2005).

### A.4.3 3D REACHING TASK: MODEL, PARAMETERS, AND ADDITIONAL ANALYSES

The first problem studied in Sec. 5.2 is defined by the following matrices:

$$A = \begin{pmatrix} 1 & 0 & 0 & \Delta t & 0 & 0 \\ 0 & 1 & 0 & 0 & \Delta t & 0 \\ 0 & 0 & 1 & 0 & 0 & \Delta t \\ 0 & 0 & 0 & 1 & 0 & 0 \\ 0 & 0 & 0 & 0 & 1 & 0 \\ 0 & 0 & 0 & 0 & 0 & 1 \end{pmatrix}$$

$$B = I_6$$

$$C = \sigma_\varepsilon \cdot I_6$$

$$H = I_6$$

$$D = \sigma_\rho \cdot I_6$$

$$\Sigma_\xi = \sigma_\xi^2 \cdot I_6$$

$$\Sigma_\omega = \sigma_\omega^2 \cdot I_6$$

$$\Sigma_\eta = \sigma_\eta^2 \cdot I_6$$

$$Q_{1,\dots,T-1} = 0_{6 \times 6}$$

$$Q_T = \begin{pmatrix} 10 & 0 & 0 & 0 & 0 & 0 \\ 0 & 10 & 0 & 0 & 0 & 0 \\ 0 & 0 & 10 & 0 & 0 & 0 \\ 0 & 0 & 0 & 1 & 0 & 0 \\ 0 & 0 & 0 & 0 & 1 & 0 \\ 0 & 0 & 0 & 0 & 0 & 1 \end{pmatrix}$$

$$R_t = r \cdot I_6 \quad \text{for } t = 1, \ldots, T-1$$

$$R_T = 0 \,,$$

where $I_6$ denotes the $6 \times 6$ identity matrix, and $0_{6 \times 6}$ denotes the $6 \times 6$ zero matrix. The initial conditions are given by:

$$\mathbb{E}[x_1] = \begin{pmatrix} 1.5 & 1.0 & 2.5 & 10^{-5} & 10^{-5} & 10^{-5} \end{pmatrix}^\top$$

$$\mathbb{E}[z_1] = \mathbb{E}[x_1]$$

$$\Sigma_{x_1} = 0_{6 \times 6}$$

$$\Sigma_{z_1} = 0_{6 \times 6}$$

The parameters of the problem are listed in Table 6 (std = standard deviation).

Table 6: Parameters of the 3D reaching task

| Name | Description | Value |
|------|-------------|-------|
| $\Delta t$ | Time step (s) | 0.010 |
| $T$ | Time steps | 100 |
| $m$ | Dimension of state $x_t$ | 6 |
| $n$ | Dimension of internal state $z_t$ (for M-Mis) | 6 |
| $p$ | Dimension of observation $y_t$ | 6 |
| $k$ | Dimension of control $u_t$ | 6 |
| $r$ | Control cost scaling | 0.0001 |
| $\sigma_\xi$ | Std of dynamics noise $\xi_t$ | 0.5 |
| $\sigma_\omega$ | Std of additive sensory noise $\omega_t$ | 0.5 |
| $\sigma_\rho$ | Std of multiplicative sensory noise $\rho$ | 0.4 |
| $\sigma_\varepsilon$ | Std of multiplicative control noise $\varepsilon_t$ | 0.4 |
| $\sigma_\eta$ | Std of additive internal noise $\eta_t$ | $\{0.0, 0.1, 0.3, 0.4, 0.5, 1.0, 2.0\}$ |

In this experiment, we set the control matrix to $B = I_6$ and use a control signal with dimensionality equal to the state ($p = m = 6$), enabling full control of the system. This choice is primarily motivated by numerical considerations: it avoids instabilities in our Model Mismatch algorithm related to matrix inversions that arise when $B$ is not full-rank or poorly conditioned.

Although this means that control directly affects all state variables – including positions – this can be interpreted as an idealized feedback mechanism. The dynamics matrix $A$ still captures the physical structure, with positions evolving from velocities over time. Our focus is on assessing algorithmic performance under internal and multiplicative noise, rather than enforcing strict biomechanical realism. Nonetheless, the setup remains rich enough to support meaningful behavioral predictions and comparisons with biological control strategies.

**Additional Analyses** As internal noise grows, the internal variable becomes increasingly reliant on sensory feedback: the pseudo-filter matrices $P_{0,\ldots,T}$ induce stronger transformations to compensate for the unreliability of internal dynamics. In contrast, the control matrix $L_t$ induces weaker transformations (in terms of volume scaling) to suppress internal fluctuations when generating the control signal $u_t = L_t z_t$ (Fig. 4a).

Notably, this modulation impacts the scaling properties of the system but not the effective embedding dimensionality – i.e., the number of dimensions corresponding to dynamically relevant directions (see next paragraph) – of the matrices involved (Fig. 4b). Interestingly, the volume scaling of the internal dynamics ($W_t$), remains constant (Fig. 4a).

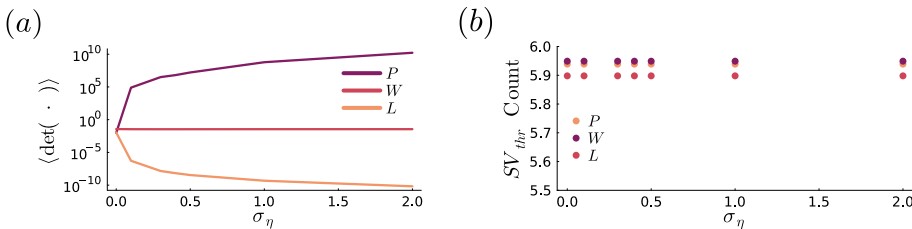

Figure 4: *3D Reaching Task: Additional Analyses.* **(a)** Time-averaged determinants of $P_t$, $W_t$, and $L_t$. **(b)** Time-averaged embedding dimensionality of the same matrices (see next paragraph for details).

**Embedding Dimensionality**   In Fig. 4b, we plot the embedding dimensionality of the matrices $P$, $W$, and $L$. For each time step $t$, we compute the number of singular values of $P_t$, $W_t$, and $L_t$ that are larger than $0.01 \cdot \max_{\sigma_i \in SV}\{\sigma_i\}$, where $SV$ denotes the set of singular values of the matrix under consideration. We then average this count across time steps to obtain a measure of effective dimensionality. Formally, we define:

$$SV_{thr}\text{Count} = \sum_{\sigma_i \in SV} \theta \left( \sigma_i \geq 0.01 \cdot \max_{\sigma_j \in SV} \sigma_j \right)$$

where $\theta(x)$ is the Heaviside step function. This quantity provides an estimate of the "effective" dimensionality of the transformation induced by the matrix, relative to its dominant singular values. This method accounts for changes in scale – such as reductions or increases in determinant magnitude due to varying levels of internal noise (Fig. 4a) – and thus provides a more meaningful estimate of dimensionality across different values of $\sigma_\eta$.

### A.4.4   DISTINCT NEURAL AND BEHAVIORAL SIGNATURES OF MODEL MATCH AND MODEL MISMATCH APPROACHES

While our main focus is to introduce an analytical solution to stochastic optimal control problems with multiplicative and internal noise, the two frameworks considered here – Model Match and Model Mismatch – also lead to distinct, experimentally testable predictions. Below we outline illustrative examples that highlight these differences and the importance of choosing between the two approaches.

**Divergence of internal dynamics**   In the 3D reaching task (Figs. 1d-g), the Model Mismatch approach exhibits qualitatively different strategies from the Model Match one. With internal noise, optimal control (Fig.1e) is achieved when internal dynamics diverge from external ones (Fig. 1f), leading to $z_t$ that no longer tracks $x_t$ (Fig. 1g). This suggests a fundamentally different way of handling internal fluctuations. Using inverse optimal control (Schultheis et al., 2021; Straub & Rothkopf, 2022), behavior can be fit under both Model Match and Model Mismatch approaches, allowing one to test whether neural activity aligns more closely with the inferred internal dynamics of one framework. If it resembles M-Match's $z_t$, it may reflect state estimation (e.g., posterior parietal cortex or cerebellum); if it resembles M-Mis's $z_t$, it may reflect control-optimized representations, possibly in premotor or motor areas.

**Noise-Dependent Control Magnitude**   From a behavioral perspective, in the same task as above, the magnitude of the control signal is strongly modulated by internal noise in the Model Match approach (Fig. 5a). In contrast, the Model Mismatch approach maintains a stable temporal profile of control magnitude across noise levels (Fig. 5a), likely due to flexible internal representations not constrained to track the external state (Figs. 1f,g). Internal fluctuations could in principle be experimentally influenced or estimated (Speed et al., 2020; Vinck et al., 2015), making this prediction possibly testable.

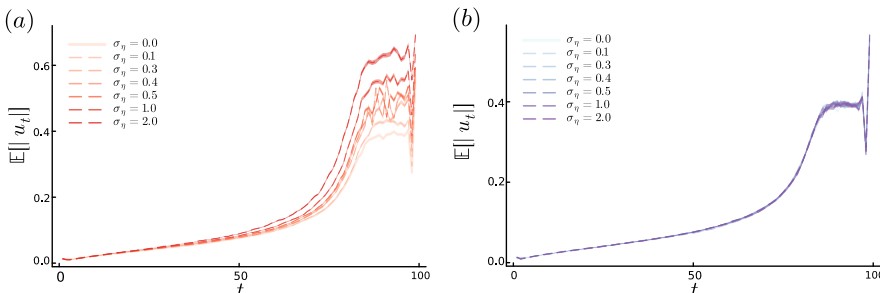

Figure 5: *Noise-dependent control magnitude in the two approaches.* **(a)** Expected control magnitude $|u_t|$, averaged over 10,000 realizations while varying internal noise $\sigma_\eta$ in the Model Match framework (shaded areas indicate the standard error of the mean). **(b)** Same as (a), but for the Model Mismatch framework.

**Perturbation Responses** To further probe the distinction between the Model Match and Model Mismatch approaches, we simulated the 3D reaching task from Figs. 1d-g with a transient bump of magnitude $d = 2.0$ applied to the second component of $x_t$ at $t = 20$, without reoptimizing. Both methods successfully compensate for the perturbation (Fig. 6a), as expected from their respective optimal solutions. Moreover, the behavioral output does not show visible qualitative differences across approaches (Fig. 6a). However, the internal dynamics diverge: in M-Mis, $z_t$ shows a non-linear, non-monotonic response with a slower return to baseline (Fig. 6b), strongly modulated by internal noise $\sigma_\eta$ (Fig. 6c). In contrast, M-Match displays a Kalman-like profile, where $z_t$ follows the perturbation magnitude and decays smoothly and monotonically (Fig. 6b), largely independent of noise (Fig. 6d). These findings suggest that M-Match and M-Mis could yield distinguishable neural signatures following perturbations, even when behavioral outputs remain similar.

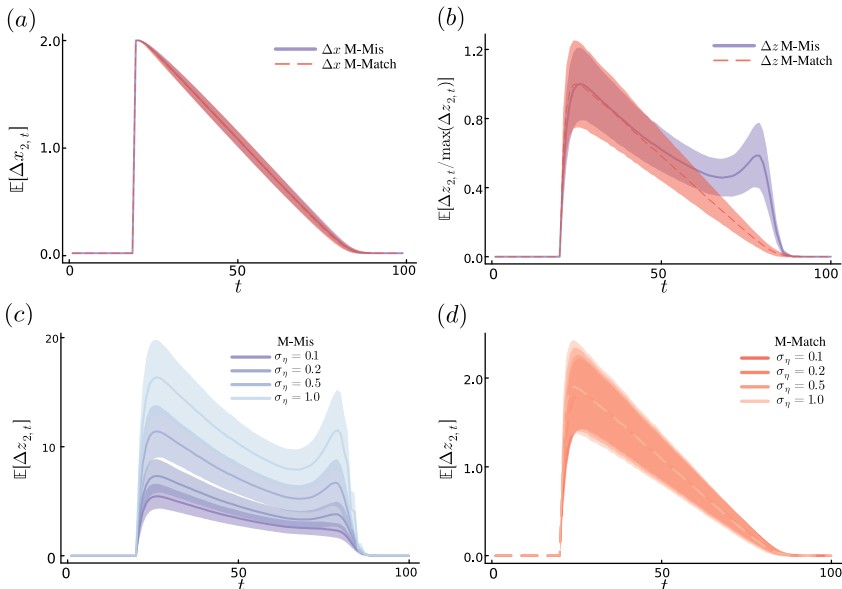

Figure 6: *Perturbation Responses in Model Match and Model Mismatch.* **(a)** Difference in the second component of the state ($y$-coordinate) between perturbed and unperturbed trials (same noise realization), averaged over 10,000 trials for the Model Match and Model Mismatch approaches, with $\sigma_\eta = 0.5$. **(b)**. Difference in the second component of the internal estimate between perturbed and unperturbed trials (same noise realization), averaged over 10,000 realizations for both approaches, normalized to their maximum, with $\sigma_\eta = 0.5$. **(c)**. Difference in the second component of the internal estimate between perturbed and unperturbed trials (same noise realization), averaged over 10,000, for the Model Mismatch approach at different levels of internal noise. **(d)**. Same as $(c)$, but for the Model Match approach. In all panels, shaded areas indicate the standard error of the mean.

### A.4.5 REDUNDANT ARM-CONTROL TASK: MODEL, PARAMETERS, AND ADDITIONAL ANALYSES

In Sec. 5.2, we also apply our algorithms to a 3-DOF planar arm performing a reaching movement around a stable reference posture. Below, we outline the full model, parameter choices, and additional analyses.

**Problem definition** We use a standard linear time-invariant (LTI) approximation around a fixed posture, as is common for moderate-amplitude reaching movements (Todorov & Jordan, 2002).

We consider a six-dimensional state (three joint angles and their angular velocities), a nine-dimensional control (muscle-like activations), and a three-dimensional observation (only joint angles are observed), i.e. $m = 6$, $p = 9$, $k = 3$. We denote by $\theta_t \in \mathbb{R}^3$ the joint-angle vector and by $\omega_t \in \mathbb{R}^3$ the corresponding angular velocities. The discrete-time dynamics with time step $\Delta t$ are

$$\theta_{t+1} = \theta_t + \Delta t \, \omega_t,$$
$$\omega_{t+1} = \left( I_3 - \Delta t \, M_{\text{joint}}^{-1} D_{\text{joint}} \right) \omega_t + \Delta t \, M_{\text{joint}}^{-1} S \, u_t,$$

where $I_3$ denotes the $3 \times 3$ identity matrix and $u_t \in \mathbb{R}^9$ is the control vector.

The muscle-to-joint map $S \in \mathbb{R}^{3 \times 9}$, which linearly converts muscle activations into joint torques, is

$$S = \begin{pmatrix} 1.2 & -1.0 & 0.0 & 0.8 & -0.6 & 0.0 & 0.5 & 0.0 & 0.0 \\ 0.0 & 0.0 & 1.0 & -0.4 & 0.6 & -0.5 & 0.0 & 0.5 & 0.0 \\ 0.0 & 0.0 & 0.0 & 0.0 & 0.0 & 1.0 & 0.0 & -0.3 & 0.6 \end{pmatrix}.$$

To construct the muscle-to-joint actuation matrix $S \in \mathbb{R}^{3 \times 9}$, we aimed to introduce a realistic and interpretable form of redundancy rather than an arbitrary high-dimensional control map. The structure of $S$ loosely mimics the organization of mono-articular and bi-articular muscles in the upper limb (e.g., Tahara et al. (2009)): each control channel acts as a simplified "muscle-like" actuator whose nonzero entries indicate which joints it spans, and whose signs emulate flexor versus extensor action. Although the exact numerical values are not intended to reproduce detailed biomechanics, the sparsity and sign patterns encode meaningful coupling across joints. This yields a redundant but structured control system in which multiple activation patterns can produce the same torque, preserving the essential geometric properties of musculo-skeletal redundancy while keeping the model analytically tractable.

The inertia and damping matrices are

$$M_{\text{joint}} = \text{diag}(m_1, m_2, m_3), \qquad D_{\text{joint}} = d_{\text{damp}} I_3,$$

with $m_1 = 1.2$, $m_2 = 0.8$, $m_3 = 0.5$ and $d_{\text{damp}} = 2.0$.

We define the state, control, and observation variables as

$$x_t = \begin{pmatrix} \theta_t \\ \omega_t \end{pmatrix} \in \mathbb{R}^6, \qquad \theta_t, \omega_t \in \mathbb{R}^3, \qquad u_t \in \mathbb{R}^9, \qquad y_t \in \mathbb{R}^3.$$

The matrices of the whole dynamical system are

$$A = \begin{pmatrix} I_3 & \Delta t \, I_3 \\ 0_{3\times 3} & I_3 - \Delta t \, M_{\text{joint}}^{-1} D_{\text{joint}} \end{pmatrix} \in \mathbb{R}^{6\times 6},$$

$$B = \begin{pmatrix} 0_{3\times 9} \\ \Delta t \, M_{\text{joint}}^{-1} S \end{pmatrix} \in \mathbb{R}^{6\times 9},$$

and the multiplicative control-noise matrix is

$$C = \sigma_\varepsilon \, B.$$

Only joint angles are observed, hence

$$H = \begin{pmatrix} I_3 & 0_{3\times 3} \end{pmatrix} \in \mathbb{R}^{3\times 6}, \qquad D = \sigma_\rho \, H \in \mathbb{R}^{3\times 6}.$$

The state cost used in the optimal control problem is diagonal:

$$Q_t = \mathrm{diag}(q_\theta, q_\theta, q_\theta, q_\omega, q_\omega, q_\omega), \qquad t = 1, \ldots, T,$$

with $q_\theta = 1.0$ and $q_\omega = 10^{-3}$. The control cost is

$$R_t = r\, I_9, \qquad r = 10^{-2}, \qquad t = 1, \ldots, T - 1,$$

with the last control cost being zero. Additive process and sensory noises are

$$\Sigma_\xi = \sigma_\xi^2 I_6, \qquad \Sigma_\omega = \sigma_\omega^2 I_3,$$

and internal noise is modeled as

$$\Sigma_\eta = \sigma_\eta^2 I_6.$$

In all simulations we use zero-mean, zero-covariance initial conditions:

$$\mathbb{E}[x_1] = 0_{6\times 1}, \qquad \mathbb{E}[z_1] = 0_{6\times 1},$$
$$\Sigma_{x_1} = 0_{6\times 6}, \qquad \Sigma_{z_1} = 0_{6\times 6}.$$

The parameters of the problem are listed in Table 7 (std = standard deviation).

Table 7: Parameters of the Redundant arm-control task

| Name | Description | Value |
|---|---|---|
| $\Delta t$ | time-step $(s)$ | 0.010 |
| $T$ | time steps | 300 |
| $\sigma_\xi$ | std of dynamics noise $\xi_t$ | 0.1 |
| $\sigma_\omega$ | std of the sensory noise $\omega_t$ | 0.1 |
| $\sigma_\varepsilon$ | std of the control-dependent noise $\varepsilon_t$ | 0.1 |
| $\sigma_\rho$ | std of the sensory-dependent noise $\rho$ | 0.1 |
| $\sigma_\eta$ | std of the additive internal noise $\eta_t$ | $\in [0.2, 0.5]$ |

**Additional Analyses** As described in Sec. 5.2, the M-Match solution channels internal variability into cost-irrelevant and unobserved state dimensions, thereby stabilizing the control output (in this task only joint angles are strongly penalized and observed, as specified by $Q$ and $H$). This can be seen by analyzing the principal components of the internal variable $z_t$. As internal noise increases, the first PC of $z_t$ (explaining more than $90\%$ of total variance) becomes aligned with the directions corresponding to the unobserved and cost-irrelevant components of the state (here the angular velocities). In Fig.7a, the first PC of $z_t$ has negligible loading on the first three (cost-relevant) dimensions and substantial loading only on the last three (cost-irrelevant) dimensions, indicating that variability is routed into the cost-irrelevant subspace. Notably, the first PC of $z_t$ maintains nearly identical direction as $\sigma_\eta$ increases (Fig.7b, red line, where the absolute projection with first PC at low noise level and all other first PCs at higher noise levels is computed).

Conversely, in the Model Mismatch framework, the first PC of $z_t$ substantially changes with internal noise (Fig.7b, purple curve), reflecting a noise-adaptive internal computation unavailable to the M-Match model.

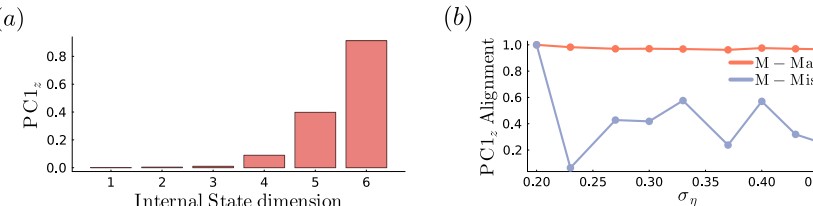

Figure 7: *Redundant Arm-Control Task: Additional Analyses.* **(a)** Components of the first principal component (PC1) of the internal state $z_t$ in the M-Match solution (computed over 500 trials) for $\sigma_\eta = 0.33$. **(b)** Alignment between the first principal components of $z_t$ across noise levels, computed as the normalized absolute scalar product (i.e absolute cosine similarity) between the reference PC1 at the smallest $\sigma_\eta$ and the PC1 at higher noise levels.

### A.4.6 Neural Population Steering via Model Mismatch Control: Model and Parameters

In Sec. 5.3, we showed how the Model Mismatch framework can be used to model a wider range of problems by going beyond the classical estimation–control setting. To illustrate this, we considered a task in which an unstable neural population is stabilized and steered toward a target state by another linear population. We model two populations of $N_{\text{units}} = 100$ linear neurons, each with sparse Gaussian recurrent connectivity, following standard assumptions from dynamical mean-field theory (Sompolinsky et al., 1988; Rajan et al., 2010). Here, the matrix $A$ represents the recurrent connectivity of the $x_t$ population, whereas $W$ represents the connectivity of the $z_t$ population. They are given by

$$A_{ij} \sim \mathcal{N}\left(0, \frac{g_A}{\sqrt{N_{\text{units}}}}\right), \quad i, j = 1, \ldots, N_{\text{units}},$$

and

$$W_{ij} \sim \mathcal{N}\left(0, \frac{g_W}{\sqrt{N_{\text{units}}}}\right), \quad i, j = 1, \ldots, N_{\text{units}}.$$

Note that internal dynamics is fixed over time, $W_{0,\ldots,T} = W$. The activity of the second population is linearly read out through a time-varying matrix $L_t$, which is optimized to steer the activity of the first population toward a desired target state while minimizing control effort (see Fig. 3a). The population $z_t$ receives input from $x_t$ through sparse random projections defined by

$$P_{ij} \sim \mathcal{N}\left(0, \frac{g_P}{\sqrt{N_{\text{units}}}}\right), \quad i, j = 1, \ldots, N_{\text{units}}.$$

Again we consider $P_{0,\ldots,T} = P$. To conform this setup to our control framework, we set $m = n = p = k = N_{\text{units}}$, and define

$$B = H = I_{N_{\text{units}}}$$
$$D = \Sigma_\omega = 0_{N_{\text{units}} \times N_{\text{units}}}.$$

The cost and noise structure of the problem are defined by the following matrices

$$C = \sigma_\varepsilon \cdot I_{N_{\text{units}}},$$
$$\Sigma_\xi = \sigma_\xi^2 \cdot I_{N_{\text{units}}},$$
$$\Sigma_\eta = \sigma_\eta^2 \cdot I_{N_{\text{units}}},$$
$$Q_{1,\ldots,T-1} = q_{<T} \cdot I_{N_{\text{units}}},$$
$$Q_T = q_T \cdot I_{N_{\text{units}}},$$
$$R_t = r \cdot I_{N_{\text{units}}}, \quad \text{for } t = 1, \ldots, T-1,$$
$$R_T = 0.$$

The initial conditions are given by:

$$\mathbb{E}[x_1] \sim \mathcal{N}\left(0, g_{x_1}^2 I_{N_{\text{units}}}\right),$$
$$\mathbb{E}[z_1] \sim \mathcal{N}\left(0, g_{z_1}^2 I_{N_{\text{units}}}\right),$$
$$\Sigma_{x_1} = 0_{N_{\text{units}} \times N_{\text{units}}},$$
$$\Sigma_{z_1} = 0_{N_{\text{units}} \times N_{\text{units}}}.$$

As stated above, the choice of Gaussian-distributed connectivity for the recurrent matrices $A$, $W$, and the feedforward matrix $P$ is grounded in principles from dynamical mean-field theory, which describes the macroscopic behavior of large, sparsely connected networks of rate neurons (Sompolinsky et al., 1988; Rajan et al., 2010). We set $g_A = 1.1$ to ensure that the state dynamics in $x_t$ are intrinsically unstable – this choice is deliberate, as our objective is to stabilize the system through control. Since we define the desired target state as zero, using it as a reference point, the initial condition effectively coincides with the goal. In this setting, a naturally decaying (stable) dynamics would trivially converge to the target without requiring active control. Instead, by inducing unstable dynamics, we create a scenario where control is essential to prevent divergence from the desired state. The internal dynamics gain $g_W = 0.9$ places the latent population $z_t$ in a subcritical regime,

supporting stable internal representations of the external dynamics. Lastly, the feedforward gain $g_P = 0.3$ models sparse and weak inter-population connectivity. These structured random matrices instantiate biologically inspired constraints that the Model Mismatch framework naturally accommodates while enabling effective control. The parameters of the problem are listed in Table 8 (std = standard deviation).

Note that the "dynamics noise" $\xi_t$ now represents the internal noise affecting the population $x_t$, analogous to the role of $\eta_t$ for the population $z_t$. We also observe that the initial condition of the population $z_t$ reflects spontaneous activity arising from internal fluctuations; accordingly, we set $g_{z_1} = \sigma_\eta$ to match the scale of this variability.

Table 8: Parameters of the Neural Steering task

| Name | Description | Value |
|------|-------------|-------|
| $T$ | Time steps | 50 |
| $r$ | Control cost scaling | 0.001 |
| $q_{<T}$ | Task-related cost scaling | 0.001 |
| $q_T$ | Task-related cost scaling | 0.1 |
| $g_{x_1}$ | Initial condition scaling for $x_1$ | 10.0 |
| $g_{z_1}$ | Initial condition scaling for $z_1$ | 0.2 |
| $g_A$ | Scaling of random connectivity of population $x_t$ | 1.1 |
| $g_W$ | Scaling of random connectivity of population $z_t$ | 0.9 |
| $g_P$ | Scaling of random connections from population $x_t$ to population $z_t$ | 0.3 |
| $\sigma_\xi$ | Std of dynamics noise $\xi_t$ | 0.5 |
| $\sigma_\varepsilon$ | Std of multiplicative control noise $\varepsilon_t$ | 0.0 |
| $\sigma_\eta$ | Std of additive internal noise $\eta_t$ | 0.2 |

Lastly, we note that although Sec. 5.3 highlights qualitative parallels with results from related RL-based approaches, our method is fundamentally different. In the linear–quadratic setting we study, the optimal solution is obtained analytically via fixed-point equations, yielding deterministic updates and very low computational cost. RL methods—both model-free and model-based—require Monte-Carlo roll-outs, which incur high sample complexity and high variance under multiplicative noise, making them far less efficient for this class of problems.

### A.4.7 COMPARISON WITH KALMAN FILTERING UNDER MULTIPLICATIVE NOISE

To compare our algorithm with alternative analytical approaches to stochastic optimal control, and to demonstrate that multiplicative and internal noises break the separation principle, we evaluated an alternative method in which the internal estimate $z_t$ is replaced with a Kalman filter that is optimal for estimation only. This allows us to directly test whether – as expected from theory (Todorov, 2005) – estimation and control cannot be optimized independently once we move beyond the classical LQAG setting.

To the best of our knowledge, there is no Kalman filtering theory that can optimally accommodate control-dependent multiplicative noise in the state dynamics or internal noise in the estimator dynamics. Nevertheless, we considered the Kalman-like filter proposed by Wu et al. (2016), which is specifically designed for linear systems with additive and multiplicative measurement noise, and thus most closely aligns with the subset of our problem where their assumptions hold. We implemented the filtering equations of Wu et al. (2016) in the simplest setting where they apply: no control-dependent noise and no internal noise. We then used the 1-D reaching task of Sec. A.4.1, with slightly adjusted parameters (see Table 9), and swept the magnitude of multiplicative sensory noise $\sigma_\rho$. We included a small but non-zero intermediate state cost by setting $Q_t = 0.0001 I_m$, $\forall t = 1, ..., T - 1$, where $I_m$ is the $m \times m$ identity matrix, and we considered process noise $\sigma_\xi$ affecting all components of the state.

For each value of $\sigma_\rho$, we computed the estimator gains $K_t$ using the algorithm of Wu et al. (2016) and then optimized the controller $L_t$ using our analytical M-Match update, and we compared with the full solution of our M-Match algorithm, where both control and filter gains are jointly optimized.

Our results show that when $\sigma_\rho = 0$, the methods behave identically, as expected from classical LQAG theory where the separation principle holds. However, as $\sigma_\rho$ increases, using the gains $K_t$ returned by Wu et al. (2016) leads to markedly sub-optimal control performance, even when $L_t$ is re-optimized using our M-Match algorithm. In contrast, the full M-Match solution achieves substantially lower expected cost (Fig. 8).

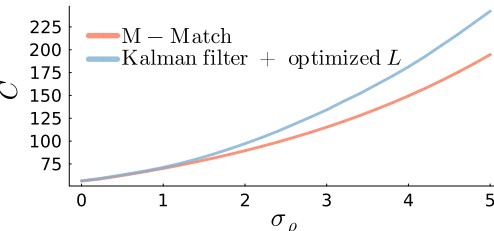

Figure 8: *Effect of Multiplicative Sensory Noise on Control Performance*. Expected cost for the M-Match solution (red) and for the Kalman-filter-based approach with re-optimized $L_t$ (blue), plotted as a function of sensory multiplicative noise $\sigma_\rho$. Curves show the analytically computed expected cost. The M-Match solution consistently achieves lower cost as $\sigma_\rho$ increases, demonstrating that a fixed Kalman estimator becomes suboptimal when multiplicative noise is present and joint estimation–control optimization is required.

These findings confirm the theoretical expectation: enforcing a fixed Kalman-filter structure (such as that of Wu et al., 2016) degrades performance once multiplicative or internal noise is present. In such settings, the estimator must adapt to the control law and vice-versa. Therefore, joint optimization is essential.

Table 9: Parameters of the single-joint reaching task for the Kalman filtering test

| Name | Description | Value |
|------|-------------|-------|
| $\Delta t$ | time-step ($s$) | 0.010 |
| $m$ | mass of the hand ($Kg$) | 1 |
| $\tau_1$ | first time constant of the second order low pass filter | 0.04 |
| $\tau_2$ | second time constant of the second order low pass filter | 0.04 |
| $r$ | Auxiliary variable for control-dependent cost | 0.001 |
| $w_v$ | Auxiliary variable for task-related cost | 0.2 |
| $w_f$ | Auxiliary variable for task-related cost | 0.01 |
| $T$ | time steps | 1000 |
| $x_1$ | Target position | 0.0 |
| $\sigma_x$ | Target position standard deviation | 0.0 |
| $\sigma_\xi$ | std of dynamics noise $\xi_t$ | 0.5 |
| $\sigma_\omega$ | std of the sensory noise $\omega_t$ | 0.5 |
| $\sigma_\varepsilon$ | std of the control-dependent noise $\varepsilon_t$ | 0.0 |
| $\sigma_\rho$ | std of the sensory-dependent noise $\rho$ | $\in [0.0, 5.0]$ |
| $\sigma_\eta$ | std of the additive internal noise $\eta_t$ | 0.0 |

### A.4.8 ROBUSTNESS TO NON-GAUSSIAN NOISE

As outlined in Sec. 3, the solutions derived through our M-Match or M-Mis algorithms depend only on 1st and 2nd order moments of the noise terms. Consequently, no distributional assumptions beyond finite covariance are required, and the method applies to any noise source with well-defined second moments. To validate this point empirically, we repeated the Monte-Carlo simulations of the 1D reaching task of Appendix A.4.1 – with the same parameters as Appendix A.4.1 – using three noise distributions for all noise terms with matched variance but strongly differing shapes. Besides the Gaussian baseline, we tested: (i) heavy-tailed Student-t noise ($\nu = 5$), introducing

occasional large outliers; and (ii) skewed $\beta(2, 5)$ noise, rescaled to zero mean and matched variance, introducing substantial asymmetry and bounded support. All control, filter, and internal parameters were kept fixed across conditions.

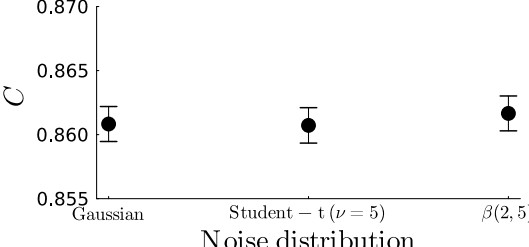

Figure 9: *Robustness to Non-Gaussian Noise*. Mean total cost ($\pm$ standard error of the mean across $50,000$ Monte-Carlo trials) obtained under three noise distributions with matched variance: Gaussian (baseline), heavy-tailed Student-t ($\nu = 5$), and skewed $\beta(2, 5)$. Despite strong differences in shape, tail behavior, and symmetry, all distributions yield nearly identical expected cost, confirming that—under linear dynamics and quadratic cost—performance depends only on second moments and not on Gaussianity.

Because the dynamics are linear and the cost is quadratic, the expected cost should depend only on second moments and therefore remain invariant across noise distributions. This prediction is confirmed in Fig. 9: the mean total cost is nearly identical for all three distributions, despite their markedly different shapes. This numerical result further supports the theoretical claim that the framework does not require Gaussian noise, and that performance depends solely on the covariance structure of the perturbations.

## A.5 LLM USAGE

Large Language Models (LLMs) were used exclusively to assist with writing clarity – specifically for grammar correction, wording suggestions, and improving readability. No part of the technical content (including research ideas, mathematical derivations, proofs, analyses, experiments, or results) was generated by an LLM. The authors take full responsibility for all scientific content in the manuscript.

