# OpenReview forum: "Optimal Control under Multiplicative and Internal Noise with Model Mismatch"
_ICLR.cc/2026/Conference — Submitted to ICLR 2026_

### Official Review · Reviewer_D2oY · 2025-10-21

**Soundness:** 3
**Presentation:** 2
**Contribution:** 3
**Rating:** 4
**Confidence:** 3

**Summary:**

This paper presents a method for filtering of a latent state under, both, multiplicative and internal noises.  This framework extends the traditional linear-Gaussian assumption that makes Kalman filtering optimal.  The method, M-Match, provably converges to a fixed point of the linear-quadratic multiplicative internal (LQMI) model.  The authors further extend the methodology to M-Mis, which allows for model mismatch between the internal dynamics and the extrinsic dynamics in the environment.  The authors demonstrate utility of, both, M-Match and M-Mis along with a baseline (Todorov, 2005) in a 3D reaching task.  Additional experiments are presented in the appendix.

**Strengths:**

The paper reportedly presents the first analytical and convergent solution for stochastic linear control under multiplicative and internal noise.  The framework is fairly generalizable as it accounts for multiplicative and internal noise as well as model mismatch.  This work substantively builds on the numerical gradient descent method of Damiani et al. (2024) to converge up to three orders of magnitude more quickly.

**Weaknesses:**

The M-Mis methodology is presented tersely and appears speculative.  Much of the mathematical justification for this method is deferred to the extensive appendix.  Indeed, the entirety of the coordinate descent algorithm, and convergence guarantees, are religated to Appendix A.2.7 and A.3.2.  Due to the relatively brief coverage in the main text, the authors might consider removing the model mismatch component of this work for a more concise submission.

The experimental results provided in the main text are insufficient to support claims made in the paper, and to establish generality of the approach.  Much like in the previous comment, a substantial amount of experimentation is deferred to the appendix.  Even the 3D reaching task, presented in the main text, is not adequately described until the appendix.  Perhaps a journal is more appropriate given that the material in the appendix is critical to the submission.

The paper fails to adequately contextualize the contributions with respect to prior work.  The authors include a very brief prior work section that only includes two citations (Todorov, 2005; Damiani et al., 2024).  This brevity is despite a long history of research in stochastic optimal control, which has previously considered multiplicative noise models.  Moreover, the authors fail to compare to Damiani et al., 2024 in the results reported in the main text.

**Questions:**

A brief search finds Wu et al., "Kalman filtering with multiplicative and additive noises." WCICA (2016), which seems quite relevant. How does the present model of multiplicative noise (i.e. ignoring internal noise) and associated M-Match algorithm compare to this prior work.

---

> ### Author Response · Authors · 2025-11-24
> **Answers (1-2-3)**
>
> We thank the reviewer for the constructive feedback and for highlighting where additional clarification, comparisons, and restructuring could strengthen the manuscript; we address each of these points in turn below.
>
> 1) M-Mis methodology.
>
> Thank you for this comment. We will of course consider removing the model-mismatch component if the area chair and the other reviewers believe this is necessary, although doing so would change the scope of the paper.
>
> That said, we would prefer to keep the model-mismatch part, because we view it as a central contribution. It is derived from the same Lagrangian formulation as the Model-Match case (the mathematics behind the two algorithms is essentially identical, which is why some derivations were placed in the appendix to avoid repetition). The method also comes with formal convergence guarantees (Sec. A.2.9), and it relaxes the standard but unrealistic assumption in optimal control that the internal model must match the external dynamics - an assumption whose removal is both mathematically significant and conceptually important (Sec. 4).
>
> Empirically, allowing mismatch systematically reduces cost and produces internal representations that differ qualitatively from those in the Model-Match case. These differences yield distinct and testable behavioral and neural predictions (Sec. A.4.4 and new Sec. 5.2 -- second paragraph), and they allow inverse optimal control to identify a broader and more realistic set of candidate solutions when fitting behavioral data.
>
> In the revised version of the paper, we used the additional page to expand and clarify the M-Mis empirical results (Sec.5), so that the main text provides a clearer presentation and the approach no longer feels terse.
>
> 2) “A substantial amount of experimentation is deferred to the appendix.”
>
> Thank you for the comment. We agree that the experiments in the previous version felt too terse. For this reason, in the revised version of the paper we used the additional page to expand and clarify the M-Mis empirical results (Sec.5), so that the main text now provides a clearer presentation and the approach no longer feels terse. We also plan to expand these experiments and present them in a longer journal version, given the practical relevance and the novel predictions they generate.
>
> 3) Prior Work and comparison with Damiani et al., 2024.
>
> Thank you for the comment. To clarify the comparison with the solutions of (Damiani et al. 2024), we have now added Section 5.1.
>
> We have additionally expanded the contextualization of our contribution within the broader stochastic optimal control literature. In the Introduction (blue lines) and in the paragraph “Limitations and Future Work” (blue lines) we now include relevant references on iLQG and DDP, explicitly discussing how the standard assumptions used in these approaches - such as using the true system dynamics inside the filter and assuming unbiased state estimates under multiplicative noise - affect the quality of the resulting solutions. We hope that these connections make clearer how our work generalizes, corrects, and extends existing formulations.
>
> Finally, the “Prior Work’’ section was indeed misleadingly labeled. It was intended to highlight only the very closest papers, but we realize this created confusion. In the revised version, we have integrated that content into an expanded introduction (blue lines) that now provides a clearer and more comprehensive overview of classical LQG, stochastic optimal control with multiplicative noise, and modern iterative approaches (iLQG/DDP). We apologize for the confusion and believe the revised structure now better reflects the relevant literature and situates our contributions more clearly.

---

> ### Author Response · Authors · 2025-11-24
> **Answers (4)**
>
> 4) "A brief search finds Wu et al., "Kalman filtering with multiplicative and additive noises." WCICA (2016), which seems quite relevant. How does the present model of multiplicative noise (i.e. ignoring internal noise) and associated M-Match algorithm compare to this prior work."
>
> We thank the reviewer for pointing us to (Wu et al., 2016). We have now compared our solution with an alternative one based on the Wu et al. filter, as reported in the newly added Sec. A.4.7 and Fig. 8.
>
> Wu et al. derive a Kalman-like filter for linear systems with additive and multiplicative measurement noise. It is important to note, however, that the proposed method does not account for control dependent noise in the state dynamics nor internal noise in the estimator dynamics (Wu et al derivation relies on their Lemma 1, which corresponds to assuming the orthogonality principle, violated when internal noise is present; see Damiani et al, 2024). Both of these noise sources are central to our formulation of LQMI/M-Match and are highly relevant in biological motor control and robotics (Todoro, 2005; Li & Todorov, 2007). To the best of our knowledge, there is no Kalman filtering theory that can optimally accommodate either internal noise or control-dependent multiplicative noise in a lossless manner, which is precisely why a joint estimator–controller solution is needed in these settings (Todorov, 2005).
>
> Further, the method in Wu et al. computes a gain $K_t$​ for state estimation, but does not solve any optimal control problem. As discussed in our paper, when multiplicative or internal noises are present, the separation principle breaks: the optimal filter depends on the controller and vice-versa. Thus, using Wu’s $K_t$​ inside a control loop is not theoretically justified when multiplicative noise is present.
>
> To more directly address the reviewer’s question and substantiate our above claims, we have now implemented the exact filtering from Wu et al. (2016) paper in the simplest setting where they apply (see new Sec. A.4.7 and Fig. 8). Specifically, we used the standard 1-D reaching task from Section A.4.1, without control-dependent noise and without internal noise, and swept the magnitude of multiplicative sensory noise $\sigma_{\rho}$​.
> For each level of $\sigma_{\rho}$​, we computed the optimal estimator gains $K_t$​ using Wu et al, 2016.
> We then held these filter gains fixed and optimized $L_t$ using
>  our analytical M-Match update (which matches the numerical GD solution of Damiani et al., as shown in Sec. 5.1).
> We found that:
>
> a) When $\sigma_{\rho}=0$ (no multiplicative noise), the performance of all methods matches, as expected from classical LQG theory.
>
> b) As multiplicative sensory noise increases, using Wu’s gains $K_t$​ leads to sub-optimal control performance, even after re-optimizing $L_t$.
>  In contrast, the full M-Match solution - where both $K_t$ and $L_t$​ are jointly optimized - achieves substantially lower expected cost.
>
> Therefore, as expected, enforcing the filter to have any particular shape (e.g., Wu et al, 2016) only harms performance and delivers a poorer solution compared to ours.

---

> ### Comment · Reviewer_D2oY · 2025-11-25
>
> Thank you for your responses.  I have read the revised manuscript and I still feel that the experimental validation is lacking for a few reasons.  The most significant reason is that I still feel that too much of the content is relegated to the appendix.  Secondly,  the only comparison to baselines I see is the expected cost reported for Todorov (2005) in Fig. 2b and the comparison to Damiani et al. (2024) in Sec. 3.2.  Given the computational cost of Damiani et al. I think further comparisons are unnecessary, but why not compare to Todorov in the remaining experiments?

---

> ### Author Response · Authors · 2025-11-26
>
> Thanks a lot for your quick feedback and for the additional comments. We agree that it is important to include a full comparison with Todorov’s approach. In the revised manuscript we have now added this comparison both to the 1D reaching task (single-joint reaching,  standard benchmark used in (Todorov, 2005) and (Damiani et al., 2024); Fig. 1c) and to the redundant arm control task (Fig. 2a), updating the corresponding figures and text. Consistent with the findings of (Damiani et al., 2024), we observe that our algorithm always outperforms Todorov’s solution in terms of cost minimization. These results are expected because, as discussed in (Damiani et al., 2024) and reviewed briefly in the Introduction of our paper and in Sec. A.1, the solutions derived in (Todorov, 2005) become suboptimal in the presence of internal noise, due to the incorrect assumption of unbiased estimation used in that derivation. Importantly, this suboptimality follows directly from the theory and is not task-dependent: it is a general phenomenon.
>
> Notably, Todorov’s approach – as well as the classical M-Match formulation – cannot be applied to the neural population steering task of Sec. 5.3. Classical formulations such as (Todorov, 2005) and (Damiani et al., 2024) constrain the internal variable $z_t$ to act as a Kalman-filter estimate of $x_t$, enforcing the structural condition $W_t = A + B L_t - P_t H$ in Eq. 11 so that the internal dynamics of $z_t$ match Eq. 7. In contrast, the Model Mismatch framework relaxes this constraint by allowing $W_t$ to be freely optimized. This flexibility enables $z_t$ and $x_t$ to be treated as distinct neural populations with independent connectivity matrices $W$ and $A$ (Fig. 3a).
> Moreover, our M-Mis algorithm supports partial optimization: for example, one may fix $W$ and $P$ (e.g., as random or biologically plausible) and optimize only the readout matrix $L_t$. Such configurations are incompatible with Todorov’s approach, which ties internal connectivity directly to the external dynamics and forces $W_t$ to vary over time. This makes it impossible for those methods to model realistic interactions between distinct neural populations.
>
> Lastly, regarding Figs. 5-6, the goal of these analyses is to illustrate potential behavioral and neural predictions that differentiate M-Match and M-Mis. These figures are meant to highlight conceptual differences rather than benchmark performance. For this reason, we do not include comparisons with (Todorov, 2005) in these panels: since that solution is known to be suboptimal under internal noise, it would not provide a meaningful reference for analyses aimed at fitting behavioral or neural data under optimality principles using inverse optimal control.
>
> Regarding the concern that “too much of the content is relegated to the appendix,” we fully agree that clarity in the main text is important. If the reviewer believes it would improve the paper, we would be happy to move, for the camera-ready version, the more complex redundant-arm task from Sec. A.4.5 into the main text in place of the 3D reaching task. Unfortunately, the strict page limit prevents us from including more than two experiments in total (currently Figs. 1 and 2) without compromising the clarity of the theoretical results, the description of the algorithms, and the proper contextualization of our contribution. For this reason, we placed the additional experiments in the appendix while ensuring that all core empirical findings are presented and discussed in the main text.

---

> > ### Author Response · Authors · 2025-12-02
> > **Final Comment**
> >
> > In the final updated version of the manuscript, we have now implemented this restructuring. Most importantly, we moved almost all empirical validations – including the newly added redundant-arm experiment – from the appendix into the main text (now consolidated in Sec. 5), so that the experimental evaluation is presented directly in the body of the paper rather than relegated to supplementary material. In parallel, we moved part of the mathematical derivations to Appendix A.2 and refocused Sec. 3 on the main intuitions and key steps of the analytical solution. These changes significantly reduce reliance on the appendix and were made specifically to address the reviewer’s concern about the distribution of content and the visibility of the empirical validation.

---

### Official Review · Reviewer_UfsQ · 2025-10-26

**Soundness:** 1
**Presentation:** 2
**Contribution:** 2
**Rating:** 2
**Confidence:** 4

**Summary:**

This work provides a gradient descent algorithm that approximates the solutions for linear dynamics with quadratic costs under multiplicative and internal noise. The theoretical proof section provides performance improvement results. A simple simulation case aims to verify the superiority of sample efficiency over traditional methods.

**Strengths:**

Detailed appendix.

**Weaknesses:**

1. Before proposing numerical algorithms, there was a lack of theoretical analysis on optimizing the landscape for the original problem.
2. The process of proving algorithm theory does not provide a clear proof of convergence.
3. The comparison method and testing benchmark are relatively simple.

**Questions:**

1. The paper assumes Gaussian noise distributions. How does the performance of the algorithm change when the noise is non-Gaussian?
2. The paper mentions the similarity between the proposed framework and reinforcement learning approaches. How do the two approaches differ in terms of their control performance, and computational complexity?
3. While the paper focuses on the coordinate-descent algorithm, it would be interesting to compare its performance and efficiency with other analytical approaches, such as the Kalman-Bucy filter or other iterative methods.

---

> ### Author Response · Authors · 2025-11-24
> **Answers (1-2)**
>
> We thank the reviewer for the careful reading and for raising several important comments about the theoretical considerations, benchmarks, and comparisons; we address each of these points in detail below.
>
> 1) “Theoretical analysis on optimizing the landscape for the original problem”
>
> We thank the reviewer for raising this point, and we apologize for not providing a clearer discussion of the optimization landscape in the original manuscript. As shown in prior work (Fazel et al., 2018), the global LQG problem is non-convex even in the fully observable, noise-free setting. This implies that the more general problem considered here - featuring multiplicative noise, internal noise, and potentially mismatched internal dynamics - is also non-convex. We have now added text in Lines 173-177 and 784-786 to briefly clarify this.
> Our empirical results also show that there are multiple fixed points. We already stated this in Sec. 3 Lines 228-231.
>
> 2) "The process of proving algorithm theory does not provide a clear proof of convergence."
>
> We thank the reviewer for this observation and for carefully examining the theoretical section. Our convergence proof is indeed technical, so here we provide further hints for our (main) Theorem 1 in Sec. 3, which establishes convergence to a critical point. Although the global problem is non-convex (as discussed above), each subproblem we solve - optimizing a single block of variables (a filter gain or control gain at time $t$) while keeping all others and the moments fixed - is quadratic and therefore convex. This structure makes the overall block-coordinate scheme analytically tractable.
>
> The logic behind Theorem 1 can be summarized as follows:
>
> a) Monotonic decrease and convergence of the cost sequence: Eq 24 shows that the objective value (expected cost) is non-increasing after each update. Since the cost is a non-negative quadratic form, the sequence is bounded below by zero. By the Monotone Convergence Theorem, the sequence of cost values converges.
>
> b) Convergence to a critical point: When the algorithm reaches a limit point, the update equations (Eqs. 14, 15, 16, 32) - which encode the optimality conditions of the convex subproblems (Eqs. 23 and 26) - are satisfied. Hence, the limiting solution is a critical point of the Lagrangian in Eq. 8. Moreover, since at convergence the constraint terms vanish (Eq. 32 implies $G_t^{xx} = G_t^{zz} = G_t^{xz} = 0$ for all $t$, the Lagrangian equals the original cost in Eq. 3 at the limit point. Thus, the limiting solution is also a critical point of the original optimization problem. Because each subproblem is solved exactly, each block update drives the corresponding partial gradient to zero, ensuring that no arbitrarily small or ineffective update can occur and guaranteeing satisfaction of first-order optimality at the limit.
>
> We have revised the text in Sec. 3 and at lines 945-949 to make this reasoning more explicit, and we thank the reviewer again for pointing this out.

---

> ### Author Response · Authors · 2025-11-24
> **Answers (3)**
>
> 3) "The comparison method and testing benchmark are relatively simple."
>
> Thank you for this comment, and we apologize if this was not made sufficiently clear.
>
> First, we would like to emphasize that the one-dimensional reaching task detailed in Sec. A.4.1 is a standard benchmark for stochastic optimal control (Todorov, 2005; Damiani et al., 2024) and is already rich enough to capture key behavioral signatures of human motor control (Todorov, 2005). The three-dimensional reaching task in Sec. 5.2 (first paragraph) is a direct extension of this classic setting. These benchmarks already demonstrate substantial differences in performance between the models (Fig. 1e) and very large gains in computational efficiency (see Sec. 5.1, where we show that our algorithm is more than three orders of magnitude faster than state-of-the-art numerical methods).
>
> Second, Sec. A.4.2 shows that our algorithm scales efficiently to high-dimensional systems - up to 100 state dimensions - with computation times orders of magnitude faster than numerical baselines. This level of scalability is essential for applications in neuroscience and robotics, where high-dimensional controllers and long horizons are common.
>
> Finally, to illustrate relevance beyond “toy’’ settings, we have added a more realistic and structured control task (new Sec. 5.2 -- second paragraph and Sec. A.4.5). This experiment uses a redundant 3-DOF arm actuated by nine muscle-like channels - an architecture directly inspired by biological musculo-skeletal systems (Tahara et al., 2009) and routinely used in robotics to study coordination under redundancy. The muscle-to-joint mapping introduces actuation redundancy and forces the controller to resolve it in a task-consistent manner. The model includes noisy sensory feedback, internal noise, and multiplicative sensory- and control-dependent noise. This setting provides a high-dimensional, biologically meaningful benchmark in which the algorithm must simultaneously regulate movement, resolve redundancy, and cope with stochasticity. It also reveals qualitatively different internal computations for Model-Match vs. Model-Mismatch, yielding distinct behavioral and neural predictions. Moreover, this type of redundant control architecture is directly relevant for robotic applications (Dietrich et al., 2015).
>
> Taken together, our results show that in the Model-Match scenario our algorithm matches the solutions obtained with current state-of-the-art numerical methods while drastically reducing computation time and making the method applicable to inverse optimal control (Sec. 5.1 and Sec. A.4.2). In addition, the extended Model-Mismatch framework - whose computational cost is comparable to Model-Match, as it follows the same analytical procedure - consistently outperforms M-Match in terms of cost minimization both in traditional reaching tasks and in the more complex settings newly introduced in Sec. 5.2 -- second paragraph. Finally, M-Mis allows optimal control theory to be applied to a broader class of problems, such as the control of linear neural populations, which is not possible in the classical M-Match setup (Sec. 5.3).

---

> ### Author Response · Authors · 2025-11-24
> **Answers (4-5)**
>
> 4) "The paper assumes Gaussian noise distributions. How does the performance of the algorithm change when the noise is non-Gaussian?"
>
> We thank the reviewer for this comment, as it made us realize that the fact that we do not assume Gaussian noise was not clear enough.
>
> As noted in the previous version of the manuscript (“we have not assumed Gaussian noises nor Gaussian distribution”, lines 220-221 of the old version of the manuscript, and now line 114 in the revised version), our framework does not require Gaussian noise nor Gaussian distributions over state and hidden variables, but probably this important statement was easy to overlook.
>
> As it can be seen in Eqs. 16,23,26,32 only the first and second moments of the noise appear in the moment propagation equations and in the optimality conditions, thus not requiring any other assumption on the nature of the underlying noise beyond having finite second moments. Thus, the method applies to any noise distribution with finite covariance.
>
> We are now making this point clearer at the end of Sec. 3 (blue lines 234-237) in the revised manuscript.
>
> Although the theoretical argument already establishes the result, we also provide empirical validation for completeness.  Specifically, we repeated our simulations using several non-Gaussian noise distributions, all rescaled to have zero mean and the same variance as the Gaussian baseline. Because the dynamics are linear and the cost is quadratic, the expected cost depends only on second moments, and therefore should be identical for any noise distribution with matched covariance when computed over Monte Carlo simulations. We therefore selected three distributions that differ strongly in shape, each targeting a distinct deviation from Gaussianity:
>
> a) Gaussian noise (baseline): Standard LQG setting; symmetric and light-tailed.
>
> b) Student-t noise (heavy-tailed): A Student-t distribution with finite variance ($\nu = 5$), producing occasional large outliers. This tests sensitivity to heavy-tailed fluctuations.
>
> c) Skewed Beta noise (asymmetric): A Beta(2,5) distribution, rescaled to zero mean and matched variance, providing clear non-symmetric noise with bounded support. This tests whether skewness influences performance.
>
> As predicted, all noise types yielded essentially identical expected cost when computed over 50,000 trials in a Monte Carlo simulation (see Figure 8 of the new Sec. A.4.8).
>
> 5) "The paper mentions the similarity between the proposed framework and reinforcement learning approaches. How do the two approaches differ in terms of their control performance, and computational complexity?"
>
> We thank the reviewer for this question. Although our work is not an RL method, it is useful to highlight the conceptual differences. In the setting considered in the paper - linear dynamics with multiplicative and internal noise and quadratic costs - our approach computes the optimal solution by solving the associated fixed-point equations. This yields deterministic updates, guaranteed convergence, and very low computational cost (Sec. A.4.2).
>
> In contrast, reinforcement learning methods rely on sampling system trajectories, both in model-free and model-based RL. This leads to high sample complexity and high variance in the presence of multiplicative noise, and makes RL substantially less efficient for the class of problems studied here. Even when the model is known, RL methods still rely on Monte Carlo roll-outs to approximate gradients, whereas our approach exploits the full analytic structure of the problem and therefore would converge orders of magnitude faster. This is consistent with the results in Sec. A.4.2, where we show that our method is already substantially more efficient than numerical gradient-descent methods that use the analytic expected-cost formula as an objective function and do not require trajectory sampling. Because Monte Carlo roll-outs are substantially more expensive than evaluating the analytic cost, any model-based RL approach (which must repeatedly estimate the cost via sampling) would be even slower. Consequently, our algorithm would be significantly faster not only than model-free RL, but also than standard model-based RL methods.
>
> We originally mentioned similarity with recent RL-based results only to note qualitative parallels regarding low-dimensional effective controllers in recurrent networks (when we apply our algorithm to the neural population steering task). However, those claims are by no means central to our paper. For clarity, we now better distinguish these points in the revised manuscript (blue lines 1971-1976 of the revised manuscript).

---

> ### Author Response · Authors · 2025-11-24
> **Answers (6)**
>
> 6) "While the paper focuses on the coordinate-descent algorithm, it would be interesting to compare its performance and efficiency with other analytical approaches, such as the Kalman-Bucy filter or other iterative methods."
>
> We thank the reviewer for this suggestion. First, we would like to emphasize that our method provides the first provably convergent coordinate-descent algorithm for this class of problems. As such, when initialized near a local optimum, it converges to the optimal solution within that basin. Therefore, our results cannot be improved by selecting alternative (fixed) Kalman-like estimators or controllers. In the original submission, we already compared against the most relevant baselines – gradient descent from (Damiani et al., 2024) in Secs. 5.1 and A.4.2, and the algorithm of (Todorov, 2005) in Fig. 1e.
>
> To support our claims, and following the reviewer’s request, we now also compare our algorithm with another algorithm that uses a fixed  Kalman-like filter. We choose to employ the Kalman filter proposed by (Wu et al. 2016), which is specifically designed for linear systems with multiplicative measurement noise. The results (new Sec. A.4.7 and Fig. 8) show that this approach performs worse than ours, for reasons that are consistent with theory. First of all, Wu et al. handle multiplicative sensory noise but do not address either (i) control-dependent noise in the state dynamics or (ii) internal noise in the estimator dynamics. Both control-dependent multiplicative noise and internal noise are key to our LQMI/M-Match formulation and are highly relevant in biological motor control and robotics (Todorov, 2005; Li & Todorov, 2007). To the best of our knowledge, there is no existing Kalman filtering theory that can handle these noise sources in a lossless way – precisely why a joint estimator-controller optimization is required (Todorov, 2005).
>
> Furthermore, Wu et al. only compute an estimator gain $K_t$; they do not solve the optimal control problem. As we discuss in the manuscript, once multiplicative or internal noise is present, the separation principle breaks: the optimal filter depends on the controller and vice versa. Thus, inserting Wu's $K_t$ Kalman filter – or any other proposed Kalman filter– into a control loop is not theoretically justified in the setting we study and provides suboptimal performance.
>
> To illustrate the above points, we have now implemented the filtering equations of (Wu et al., 2016) in the simplest setting where they apply (see new Sec. A.4.7 and Fig. 8): the 1-D reaching task from Sec. A.4.1 with no control-dependent noise and no internal noise, and swept the magnitude of multiplicative sensory noise $\sigma_{\rho}$. For each value of $\sigma_{\rho}$, we computed Wu's estimator gains $K_t$ and then optimized the controller $L_t$ using our analytical M-Match update (which matches numerical GD, Sec. 5.1 of the revised manuscript). We found that:
>
> a) At $\sigma_{\rho} = 0$, the methods match (as expected from classical LQG theory).
>
> b) As $\sigma_{\rho}$ increases, using Wu's $K_t$ leads to significantly sub-optimal control performance, even if we re-optimize $L_t$.
> In contrast, the full M-Match solution – where $K_t$ and $L_t$ are jointly optimized – achieves substantially lower expected cost.
>
> Thus, as expected, enforcing a fixed filter structure (e.g., using Wu et al.'s equations) degrades performance when considering realistic features of noise. Joint optimization is essential in the presence of multiplicative and internal noise, and our approach is the only one that correctly handles these cases.

---

### Official Review · Reviewer_4btA · 2025-10-27

**Soundness:** 3
**Presentation:** 2
**Contribution:** 3
**Rating:** 6
**Confidence:** 2

**Summary:**

This paper deals with stochastic optimal control for linear systems with multiplicative and internal noise (LQMI) and presents a provably convergent coordinate descent algorithm that significantly outperforms existing numerical methods. The authors extend this approach to distinguish internal dynamics from external dynamics (“model mismatch”) and argue that this leads to better performance in the presence of internal noise and greater biological realism.

**Strengths:**

- The coordinate descent algorithm with convergence guarantees (Theorem 1) is a real step forward. The proof that the orthogonality principle only applies when there is zero internal noise (Theorem 2) formally eliminates the ambiguities from earlier work.
- The speed advantage of over three orders of magnitude compared to gradient-based methods is impressive and of practical importance
- The idea that internal representations do not have to match external dynamics challenges a central assumption of optimal control and could be relevant for neuroscience.

**Weaknesses:**

## Major
- The restriction to linear dynamics with quadratic costs is limiting. Although the authors acknowledge this, they do not sufficiently address how serious this limitation is. Many interesting control problems are inherently nonlinear.
- All experiments relate to relatively simple tasks (grasping movements, playful neural networks). The grasping task with one joint has been studied since Todorov 2005. The 3D grasping task is hardly any more complex. The neural steering task is entirely synthetic.
- The work shows that M-Mis helps with internal noise, but mainly in one task (Fig. 1). How does one decide between M-Match and M-Mis in inverse optimal control? Both fit the data, but M-Mis has more parameters.

## Minor
- Section 4 headlines "neural population steering", but does not evaluate the algorithm on that task, but directly refers to the appendix. A summary of the results on this task should be added to that section.
- Eq. 9 & 12 are hard to grasp and contain many indices and transposes. Simplification or a more intuitive presentation would improve comprehensibility.
- Related work: The current related work section is very limited. The work could be better positioned in relation to:
    - iLQG/DDP methods for nonlinear control
    - Model-based RL with mismatched dynamics
    - Work on internal models in neuroscience beyond citations
- A.5: It is good to see that the use of LLM is acknowledged, although “writing assistant” is vague and should be clarified.

**Questions:**

For inverse optimal control, how would you choose between M-Match and M-Mis given behavioral data alone?

---

> ### Author Response · Authors · 2025-11-24
> **Answers (1-2)**
>
> We thank the reviewer for the careful assessment and for raising important questions regarding limitations and applicability; we address each of these points in detail below.
>
> 1) Non-linear dynamics
>
> We thank the reviewer for the valuable comment. Many real-world control problems in neuroscience and robotics are indeed nonlinear. At the same time, there is a long and successful tradition of using linear approximations in both fields to capture movement around a reference posture or trajectory while maintaining analytical tractability. Linear or locally linear models have been remarkably effective in explaining neural and behavioral data (Todorov, 2005; Todorov & Jordan, 2002; Franklin & Wolpert, 2011; Shadmehr & Krakauer, 2008), and in controlling robotic systems (Tassa, Erez & Todorov, 2012, Tassa et al., 2014). Our work advances this theoretical line by providing, for the first time, an analytically derived, computationally efficient solution for stochastic optimal control under realistic noise models and without requiring the internal model to match the external dynamics.
>
> We agree that extending the framework to nonlinear systems is an exciting direction, and we now discuss this in the revised manuscript – blue lines in the Paragraph “Limitations and Future Work”. In general, nonlinear control problems can be approached via local linearization, as in iterative LQG and Differential Dynamic Programming (DDP). For example, Li & Todorov (2007) showed that repeatedly linearizing the dynamics around a nominal trajectory and approximating the cost as quadratic yields accurate solutions for arbitrary nonlinear systems.
> Our framework naturally fits into this paradigm. The analytical M-Match and M-Mismatch updates can be inserted inside an iLQG/DDP-style outer loop, providing improved local solutions while avoiding two limiting assumptions of current approaches:
>  (i) that the internal state must follow the true system dynamics, and
>  (ii) that the state estimate remains unbiased under multiplicative or internal noise.
>  Removing these assumptions should, in principle, yield better approximations for nonlinear stochastic control.
>
>
> 2) Simple Tasks
>
> We thank the reviewer for the opportunity to clarify this point.
>
> First, we would like to emphasize that the present manuscript already takes substantial steps toward real-world applicability. In Sec. A.4.2 we show that our algorithm scales efficiently to high-dimensional systems - up to 100 state dimensions - with computation times orders of magnitude faster than state-of-the-art numerical approaches. This level of scalability is essential for practical use in both neuroscience and robotics, where high-dimensional controllers and long horizons are common.
>
>
> We also note that the one-dimensional reaching task detailed in Sec. A.4.1 is a standard benchmark for stochastic optimal control (Todorov, 2005; Damiani et al., 2024); it is already rich enough to capture key behavioral signatures of human motor control (Todorov, 2005) and to show important differences between previous and our new solutions, including M-Math and M-Mis frameworks. The three-dimensional reaching task in Sec. 5.2 -- first paragraph -- is a direct extension of this classic setting.
>
> To illustrate relevance beyond “toy’’ settings, we have now added a more realistic and structured control task (new Sec. 5.2 -- second paragraph -- and Sec. A.4.5). This experiment uses a redundant 3-DOF arm actuated by nine muscle-like channels - an architecture directly inspired by biological musculo-skeletal systems (Thara et al., 2009) and routinely used in robotics to study coordination under redundancy. The muscle-to-joint mapping introduces actuation redundancy and forces the controller to resolve it in a task-consistent way. The model includes noisy sensory feedback, internal noise, and multiplicative sensory- and control-dependent noise. This setting provides a high-dimensional, biologically meaningful benchmark where the algorithm must simultaneously regulate movement, resolve redundancy, and cope with stochasticity. It also reveals qualitatively different internal computations for Model-Match vs. Model-Mismatch, yielding behavioral and neural predictions. Moreover, this type of redundant control architecture is directly relevant for robotic applications (Dietrich et al., 2015).
>
> Taken together, these additions show that our method scales to high-dimensional settings, as well as to more complex motor tasks, and yields new concrete predictions in realistic redundant biomechanical tasks.

---

> ### Author Response · Authors · 2025-11-24
> **Answers (3)**
>
> 3) M-Match and M-Mis in inverse optimal control.
>
> Thank you for this important question.
>
> To decide between M-Match and M-Mis in real data within an inverse optimal control framework, one can fit both models to behavioral data and then compare their log-likelihood on held-out data, following standard model-fitting procedures. Importantly, the number of parameters that must be fitted from behavior is the same in both approaches. Indeed, in inverse optimal control, we only need to estimate the system parameters (e.g., $A, B, Q, R$ and the noise variances). The control and filter gains $\{L_t, K_t\}$ in M-Match, as well as $\{W_t, L_t, P_t\}$ in M-Mis, are not free parameters; they are computed analytically by solving the optimal control problem once the system parameters are fixed (Schultheis et al., 2021). Thus, M-Mis does not introduce additional degrees of freedom in the inverse problem.
>
> Moreover, M-Match and M-Mis correspond to deeply different internal computations, even when behavior might appear similar. This means that in settings where neural recordings are available and one uses inverse optimal control to infer latent variables and compare them to neural activity, the two frameworks yield different predictions and therefore different likelihoods. For example, if internal activity resembles the $z_t$ produced by M-Match, it would indicate a state-estimation-like representation (possibly linked to areas such as PPC or cerebellum), whereas if internal activity resembles the $z_t$ of M-Mis, it would instead reflect a control-optimized representation, more consistent with premotor or motor cortical areas. A more detailed discussion of these behavioral and neural predictions is provided in Sec. A.4.4.
>
> Finally, as mentioned in the previous answer, we have added a new section in the paper (Sec. 5.2 -- second paragraph and Sec. A.4.5) where we apply both frameworks to a more complex motor task with muscle redundancy, showing how they develop different internal computations and produce distinct behavioral and neural signatures.

---

> ### Author Response · Authors · 2025-11-24
> **Answers (4)**
>
> Minor Comments
>
> 4.1) neural population steering.
>
> We thank the reviewer for the suggestion. Due to page constraints, we have added a short paragraph summarizing the empirical results from the additional experiments run using the M-Mis framework at the end of Sec. 4.1 (highlighted in blue).  **EDIT[In the last version of the manuscript, we now have a full section, Sec. 5.3]**
>
> 4.2) “Eq. 9 & 12 are hard to grasp”.
>
> We agree with the reviewer that Eqs. 9 and 12 contain many indices and transposes. Unfortunately, these expressions already represent the minimal form of the Lagrangian derivatives: each noise source (motor, sensory, internal) and each system matrix contributes a distinct term to the gradients. This complexity reflects the true structure of the underlying stochastic dynamics rather than an avoidable notational choice.
> For clarity, Eq. 9 specifies the constraints under which the cost is optimized: it enforces that the second-order moments obey their correct propagation equations. Introducing a Lagrangian to impose these constraints allows us to differentiate the objective without explicitly propagating derivatives through time.
> We agree that the Lagrange multiplier Eqs. 12 is more difficult to understand. However, in Eq. 14 we show that the Lagrange multipliers appear in the definition of the cost function, so the terms decreasing the values of the multipliers should be readily interpretable as decreasing the cost. From here it is clear e.g. that increasing the noise values typically increases the cost-to-go. The same can be said for the control cost matrices. As it is clear, these effects are independent from one another, and this is why the equations cannot be simplified further.
> We have added a short explanation of these points in Lines 879–882 of the revised manuscript to help intuition. We thank the reviewer for pointing out that additional clarification was needed.
>
> 4.3) Related work.
>
> Thank you for these suggestions. In the revised manuscript we now discuss iterative LQG and DDP more explicitly (Li & Todorov, 2007). These methods solve nonlinear problems via repeated linearization and quadratic approximation. Our analytical M-Match and M-Mismatch updates can be embedded inside such outer loops, while avoiding two limiting assumptions of standard iLQG/DDP: (i) that the filter must match the true dynamics, and (ii) that state estimates remain unbiased under multiplicative or internal noise. We added this discussion in the Introduction – blue lines.
>
> We also want to clarify here how our approach differs from RL. Indeed,  although our work is not an RL method, it is useful to highlight the conceptual differences. RL methods - both model-free and model-based - require Monte Carlo roll-outs to estimate gradients or value functions, leading to high variance and high sample complexity under multiplicative noise. Our algorithm instead solves the fixed-point equations analytically, giving deterministic updates, guaranteed convergence, and orders-of-magnitude faster runtime. We commented on this at lines 1971-1976 of the revised manuscript.
>
> We have added citations and discussion connecting our work to the broader literature on internal forward models and mismatched internal dynamics in the brain, including optimal feedback control and sensorimotor learning frameworks (Kawato, 1999; Wolpert & Ghahramani, 2000; Shadmehr & Holcomb, 1997; Scott, 2004). These works support the conceptual motivation behind the M-Mis framework. This is included and expanded in Sec. 4, blue lines.
>
> 4.4) LLM usage.
>
> We thank the reviewer for noticing that. We clarify that we used LLMs exclusively to improve readability and to check for grammar and orthographic mistakes. No part of the technical content, derivations, or analyses was generated by an LLM. We now state this more clearly in Sec. A.5.

---

> ### Comment · Reviewer_4btA · 2025-11-28
>
> Thank you for addressing all the raised points. I have read the revised manuscript and agree that the new experiment adds valuable support for a solid evaluation. However, I share Reviewer D2oY’s concern that major parts of the evaluation and experiments are presented in the appendix, while the main text often only refers to them. I suggest restructuring the manuscript so that the key evaluations and experiments are included in the main text rather than relegated to the appendix.

---

> > ### Author Response · Authors · 2025-12-02
> > **Final Comment**
> >
> > We thank the reviewer again for their positive assessment and for emphasizing the importance of presenting key evaluations in the main text. In the final updated version of the manuscript, we have now implemented the restructuring suggested here and also raised by Reviewer D2oY. Specifically, we moved almost all empirical results – including the newly added redundant-arm experiment – from the appendix into the main text (now consolidated in Sec. 5). We also relocated several mathematical details to Appendix A.2 and refocused Sec. 3 on the main intuitions and essential steps of the analytical derivation. These changes substantially reduce reliance on the appendix and ensure that the core evaluations and experiments appear directly in the main paper, as recommended.

---

### Official Review · Reviewer_HNT1 · 2025-10-31

**Soundness:** 4
**Presentation:** 2
**Contribution:** 3
**Rating:** 6
**Confidence:** 3

**Summary:**

This paper tackles stochastic optimal control in systems with complex multiplicative and internal noise, a scenario often simplified in prior work. The authors introduce a provably convergent coordinate-descent algorithm that is over three orders of magnitude faster than existing numerical methods for this problem. They also propose a novel "Model Mismatch" framework, which relaxes the assumption that an agent's internal model must match external dynamics. This "mismatch" allows the system to achieve substantially better performance, especially under high internal noise, by treating the internal variable as an abstract representation for control rather than a pure state estimator. The primary limitation of this new analytical approach remains its confinement to systems with linear dynamics and quadratic costs.

**Strengths:**

* The framework includes multiplicative noise and removes restrictive assumptions like the separated inference and control and the orthogonality principle.
* The coordinate-descent algorithm is well-motivated and analytically derived; it achieves substantial computational gains over previous gradient-based methods.
* It introduces a "Model Mismatch" framework, relaxing the common and restrictive assumption that an agent's internal model must perfectly match the external world's dynamics. The authors show that this flexibility leads to substantially better performance, particularly in the presence of internal noise.

**Weaknesses:**

* The paper is math-heavy and difficult to parse; the clarity could be improved by condensing sections 3.1&3.2 to highlight the key derivation steps, main theorem, and algorithmic intuition, while moving detailed proofs to the appendix.
* The motivations for the multiplicative noise and internal noise are unclear. It can be better illustrated with examples that help readers understand why these noise structures are central to real-world control and biological systems.
* The empirical evaluation is limited to synthetic reaching tasks and toy neural-control settings; there is little demonstration that the method scales to real neural data or application scenarios like robotics.
* The authors suggest that time-varying dynamics could approximate nonlinearities, but the evaluation did not include nonlinear dynamics that are prevalent in neuroscience and robotics; thus, the applicability of the approach is questionable.

**Questions:**

* What are the concrete scenarios or empirical phenomena that require modeling multiplicative noise? For instance, are there known biological or robotic systems where control- or signal-dependent noise has been experimentally observed?
* A central result is that internal noise biases the internal representations away from the task-optimal (Kalman-like) estimate. What do these new representations encode? If they diverge from accurate estimation, what computational advantage or robustness do they confer?
* Does the model predict measurable behavioral or neural signatures, such as altered variability, control gain modulation, or shifts in sensory weighting, that could be tested experimentally?

---

> ### Author Response · Authors · 2025-11-24
> **Answers (1)**
>
> We thank the reviewer for the detailed reading and for the insightful comments. We address all points below, and we apologize for any missing information in the original submission.
>
> 1) “The empirical evaluation is limited to synthetic reaching tasks and toy neural-control settings”
>
> We agree with the reviewer that real-world applications are important, and we plan to pursue them in subsequent work. We also thank the reviewer for the opportunity to explain this further here.
>
> First, we emphasize that the present manuscript already takes substantial steps toward real-world applicability. In Sec. A.4.2 we show that our algorithm scales efficiently to high-dimensional systems - up to 100 state dimensions - with computation times orders of magnitude faster than state-of-the-art numerical approaches. This level of scalability is essential for practical use in both neuroscience and robotics, where high-dimensional controllers and long horizons are common.
>
>
> We also note that the one-dimensional reaching task detailed in Sec. A.4.1 is a standard benchmark for stochastic optimal control (Todorov, 2005; Damiani et al., 2024) and already rich enough to capture key behavioral signatures of human motor control (Todorov, 2005). The three-dimensional reaching task in Sec. 5.2 (first paragraph) is a direct extension of this classic setting.
>
> Second, linear approximations remain widely used and remarkably successful in both neuroscience and robotics. Much of the foundational and contemporary literature in computational motor control relies on linearized models and has produced quantitatively accurate predictions (Todorov, 2005; Todorov & Jordan, 2002; Franklin & Wolpert, 2011; Shadmehr & Krakauer, 2008; Li & Todorov, 2007). Similarly, Jacobian linearization and LTI approximations for robot manipulators are standard tools for stability and control analysis (Siciliano, Sciavicco, Villani & Oriolo, 2010).
>
> In addition to this, our framework also generalizes these formulations by allowing optimal internal dynamics that need not match the external dynamics, enabling new experimentally testable predictions (see Sec. A.4.4).
>
> Finally, to illustrate relevance beyond “toy’’ settings, we added a more realistic and structured control task (new Sec. 5.2 -- second paragraph -- and Sec. A.4.5). This experiment uses a redundant 3-DOF arm actuated by nine muscle-like channels - an architecture directly inspired by biological musculo-skeletal systems (Thara et al., 2009) and routinely used in robotics to study coordination under redundancy. The muscle-to-joint mapping introduces actuation redundancy and forces the controller to resolve it in a task-consistent way. The model includes noisy sensory feedback, internal noise, and multiplicative sensory- and control-dependent noise. This setting provides a high-dimensional, biologically meaningful benchmark where the algorithm must simultaneously regulate movement, resolve redundancy, and cope with stochasticity. It also reveals qualitatively different internal computations for Model-Match vs. Model-Mismatch, yielding further behavioral and neural predictions. Moreover, this type of redundant control architecture is directly relevant for robotic applications (Dietrich et al., 2015).
>
> Taken together, these additions show that our method (i) scales to high-dimensional settings, (ii) is directly applicable in the linear regimes commonly used in neuroscience and robotics, and (iii) yields concrete predictions in realistic redundant biomechanical tasks (see new Sec. 5.2 -- second paragraph -- and Sec. A.4.5).

---

> ### Author Response · Authors · 2025-11-24
> **Answers (2-3)**
>
> 2) Non linear dynamics
>
> We thank the reviewer for bringing this important point into the discussion. We agree that many systems in neuroscience and robotics exhibit nonlinear dynamics. At the same time, there is a long and successful tradition of using linear approximations in both fields to capture movement around a reference posture or trajectory while maintaining analytical tractability. As also commented in the previous answer, linear or locally linear models have been remarkably effective in explaining neural and behavioral data (Todorov, 2005; Todorov & Jordan, 2002; Franklin & Wolpert, 2011; Shadmehr & Krakauer, 2008; Tassa, Erez & Todorov, 2012), and in controlling robotic systems (Tassa, Erez & Todorov, 2012). Our work advances this theoretical line by providing, for the first time, an analytically derived, computationally efficient solution for stochastic optimal control under realistic noise models and without requiring the internal model to match the external dynamics.
>
> We agree that extending the framework to nonlinear systems is an exciting direction, and we now discuss this in the revised Discussion – blue lines in the Paragraph “Limitations and Future Work”. In general, nonlinear control problems can be approached via local linearization, as in iterative LQG and Differential Dynamic Programming (DDP). For example, Li & Todorov (2007) showed that repeatedly linearizing the dynamics around a nominal trajectory and approximating the cost as quadratic yields accurate solutions for arbitrary nonlinear systems.
> Our framework naturally fits into this paradigm. The analytical M-Match and M-Mismatch updates can be inserted inside an iLQG/DDP-style outer loop, providing improved local solutions while avoiding two limiting assumptions of current approaches:
>  (i) that the filter must follow the true system dynamics, and
>  (ii) that the state estimate remains unbiased under multiplicative or internal noise.
>  Removing these assumptions should, in principle, yield better approximations for nonlinear stochastic control.
>
>
> 3) "A central result is that internal noise biases the internal representations away from the task-optimal (Kalman-like) estimate. What do these new representations encode? If they diverge from accurate estimation, what computational advantage or robustness do they confer?"
>
> We thank the reviewer for raising this other important point. As discussed in Section 4, once internal dynamics are allowed to differ from the external dynamics, the optimal strategy under internal noise no longer requires the internal variable $z_t$ to behave as a faithful estimate of the state $x_t$. Instead, the system can use this additional freedom to construct internal representations that are better for control than Kalman-like estimates would be.
>
> To illustrate this phenomenon more clearly, we added a more realistic and structurally rich task in the newly added Sec. 5.2 (second paragraph): a 3-DOF planar arm with muscle redundancy and realistic sensorimotor noise. This setting highlights that the Model-Match and Model-Mismatch frameworks resolve redundancy and internal fluctuations in qualitatively different ways. In the Model-Match case, the internal dynamics must follow a Kalman-like recursion, so they push variability into cost-irrelevant dimensions while tracking the cost-relevant ones (the cost function $Q$ only penalizes a subset of the components of the state vector), as shown in Figs. 2d,e. However, as internal noise grows, this estimator becomes increasingly biased, and performance degrades (Fig 2a).
>
> In contrast, the Model-Mismatch framework is not constrained to implement a Kalman filter. Its internal dynamics reorganize to form more abstract latent representations that do not aim to accurately reconstruct the components of $x_t$ (as shown in Figs. 2e), but instead encode combinations of variables that are most stable and most useful for generating the correct control signals. This flexibility allows the controller to maintain robust behavior despite large internal fluctuations: control outputs remain stable across noise levels, and performance (in terms of expected cost) is far less degraded than in the Model-Match case (Fig. 2a).
>
> In summary, the internal representations in the Model-Mismatch solution do not encode the physical state per se; they encode control-relevant, noise-robust combinations of variables. This reorganization confers a clear computational advantage: it preserves behavioral performance and control output under internal noise in a way that state-estimation-based strategies cannot. A more detailed analysis is provided in the newly added Sec.5.2 (second paragraph) and Sec. A.4.5.

---

> ### Author Response · Authors · 2025-11-24
> **Answers (4-5-6)**
>
> 4) Measurable behavioral or neural signatures
>
> We thank the reviewer again for this comment. Our framework indeed yields clear and testable behavioral and neural signatures. For example, both the Model-Match and Model-Mismatch solutions predict increased reliance on sensory feedback as internal noise grows: to compensate for the bias introduced by internal fluctuations, the control gains $L_t$ decrease while the (pseudo-)filter gains ($P_t$ or $K_t$) increase. This pattern is evident in both frameworks (Model Match: Fig. 2c; Model Mismatch: Fig. 4a).
>
> A detailed discussion of predictions in both frameworks is already provided in Sec. A.4.4.
> Moreover, by extending our analysis to a more realistic motor task with muscle redundancy (newly added Sec. 5.2 -- second paragraph), we obtain additional experimentally accessible predictions. These include EMG patterns and alignment or misalignment between internal representations and the physical state. All these signatures can be tested in human motor control and robotic settings.
>
> 5) "The paper is math-heavy and difficult to parse; the clarity could be improved by condensing sections 3.1&3.2 to highlight the key derivation steps, main theorem, and algorithmic intuition, while moving detailed proofs to the appendix."
>
> Thank you for this helpful suggestion. We fully agree that improving clarity is important. In this revision, we considered reorganizing Sections 3.1 and 3.2, but we also received a contrasting request from Reviewer D2oY (W1), who encouraged us to keep more of the mathematical content in the main text for completeness. Since the updated submission grants one additional page, we chose to use this space to expand the empirical section and better illustrate the framework in practice.
> That said, we are of course happy to adjust the structure if the reviewers believe that moving parts of Sections 3.1 and 3.2 to the appendix would significantly improve readability, and we can readily implement this change in the next version.
>
> 6) "What are the concrete scenarios or empirical phenomena that require modeling multiplicative noise? For instance, are there known biological or robotic systems where control- or signal-dependent noise has been experimentally observed?"
>
> We again thank the reviewer for the question. To avoid any misunderstanding, we emphasize that the noise model used in our work is not introduced here as a new contribution. As already noted in the manuscript, it is a well-established formulation in stochastic optimal control, originally motivated by empirical findings in human sensorimotor behavior and by the need to capture realistic sources of uncertainty.
>
> A substantial body of evidence shows that multiplicative noise is present both in motor execution and in sensory feedback. Motor commands exhibit signal-dependent noise - larger forces produce proportionally greater variability (Sutton and Sykes, 1967; Schmidt et al., 1979; Harris & Wolpert, 1998; Todorov, 2005). Similar multiplicative effects appear in sensory pathways, including vision and proprioception: for example, visual noise increases sharply outside the fovea (Burbeck & Yap, 1990; Whitaker & Latham, 1997; Todorov, 2005).
> Internal noise, in turn, reflects endogenous fluctuations in neural activity (Faisal et al., 2008; Moreno-Bote et al., 2014; Churchland et al., 2006) or inaccuracies in the internal estimation process, and is known to be necessary for explaining key aspects of behavioral variability (Todorov et al., 2005; Franklin & Wolpert, 2011). Notably, control- and sensory-dependent noise can also be used in robotics to model actuator and sensor variability (e.g., Tassa et al., 2012; Li & Todorov, 2007).
>
> The full noise model was first integrated into an optimal control framework by (Todorov, 2005), who provided a detailed justification for each component. Since then, this formulation has been used extensively in studies of inverse optimal control (e.g., Schultheis et al., 2021) and has also motivated more recent theoretical developments (Damiani et al., 2024). Beyond theoretical contributions, variants of this model have been used to interpret behavioral and neural data in a wide range of motor-control settings, including adaptation, feedback responses, brain–machine interfaces, and sensorimotor prediction (Schultheis et al., 2021; Straub et al., 2022; Sensinger & Dosen, 2020; Liu & Todorov, 2007; Izawa et al., 2008; Takei et al., 2021; Shanechi et al., 2013).

---

### Author Response · Authors · 2025-12-02
**Comment for the Area Chair (1)**

Dear Area Chair,

We understand that due to the recent administrative reset, you have been assigned to our paper. To assist your assessment, we respectfully summarize the discussion progress before it was halted.

Specifically, reviewer **4btA**, who gave a rating of 6, explicitly confirmed that their concerns were addressed after our response and they just left a suggestion for restructuring the manuscript. Similarly, reviewer **D2oY**, who gave a rating of 4, after showing satisfaction with our responses, left two additional comments, one of them also requesting restructuring the paper, and another minor comment (adding further comparison with Todorov approach), which we have now fully addressed in the final response. The other two reviewers (**HNT1** and **UfsQ**) unfortunately did not have time to post comments on our first responses. Reviewer **HNT1**, who gave a score of 6, indicated in the review similar concerns to those of reviewer 4btA, on non-linear dynamics and paper restructuring, so we expected to have been able to satisfactorily address all reviewer’s concerns as well. Reviewer **UfsQ**, who gave the lowest score of 2, proposed further comparisons with additional Kalman filters, which we addressed in our response  – a similar concern was raised by reviewer **D2oY**, who seemed satisfied with the way we addressed that particular question. Reviewer **UfsQ** also raised serious doubts about the validity and/or lack of clarity of our analytical approach, in contrast to the explicit positive evaluation of these aspects by reviewers **HNT1** and **4btA**; we added further explanation that we hoped would have convinced the reviewer, including our original comparisons between the theory and experiments, showing perfect numerical matches.

Additional details, if needed and useful for the area chair, are provided next:

**1. Resolution of Concerns Made Explicit by the Reviewers**

- **Reviewer 4btA (Score: 6):**
    - **Reason for concerns’ resolution**: We addressed the concern on non-linear dynamics limitations, added a more realistic and structured control task (new Fig. 2 and  Secs. 5.2 -- second paragraph -- and A.4.5), and explained how to experimentally distinguish between M-Mis and M-Match framework, in addition to addressing all other minor comments.
    - **Reviewer's Quote**: "**Thank you for addressing all the raised points** [...]”

**2. Support from Initial Reviews**
- **Reviewer HNT1 (Score: 6):**
This reviewer provided support for acceptance, emphasizing technical depth. They wrote that “The coordinate-descent algorithm is well-motivated and analytically derived; it achieves substantial computational gains over previous gradient-based methods.” In our response we addressed all comments raised by the reviewer, but unfortunately the discussion phase was halted before we could get feedback from this reviewer.

- **Reviewer 4btA (Score: 6):**
This reviewer provided support for acceptance, again emphasizing technical depth. They wrote that “The coordinate descent algorithm with convergence guarantees (Theorem 1) is a **real step forward**” and “The speed advantage of over three orders of magnitude compared to gradient-based methods is **impressive and of practical importance**”.

---

> ### Author Response · Authors · 2025-12-02
> **Comment for the Area Chair (2)**
>
> **3. Reasonable Expectation on Resolution of Further Concerns**
>
> These are the two reviewers who we got feedback from:
>
> - **Reviewer 4btA (Score: 6)**
>     - **Reason:** After our response, this reviewer only left a comment on restructuring the paper, which we addressed in the final version of our manuscript.
>     - **Reviewer's Quote:** "I share Reviewer D2oY’s concern that major parts of the evaluation and experiments are presented in the appendix, while the main text often only refers to them. **I suggest restructuring the manuscript** so that the key evaluations and experiments are included in the main text rather than relegated to the appendix.”
>
> - **Reviewer D2oY (Score: 4):**
>     - **Reason:** We addressed the reviewer’s only question about adding a comparison Wu et al filter; we showed that our algorithm is superior to that heuristic approach. In addition, we have contextualized better prior work, added further comparison with Todorov’s approach, and compared with Damiani et al in the main text, as requested, in addition to address all other points.
>     - **Reviewer's Quote:** "Thank you for your responses. I have read the revised manuscript and **I still feel that the experimental validation is lacking for a few reasons**. The most significant reason is that **I still feel that too much of the content is relegated to the appendix**. Secondly, the only comparison to baselines I see is the expected cost reported for Todorov (2005) in Fig. 2b and the comparison to Damiani et al. (2024) in Sec. 3.2. Given the computational cost of Damiani et al. I think further comparisons are unnecessary, but why not compare to Todorov in the remaining experiments?”
>
> As it can be seen, the reviewer seems to agree with all addressed points, but still requested additional comparison with Todorov approach and proposed restructuring the paper. We have fully addressed these two final requests by adding further comparison with Todorov approach (Figures 1c,e and 2a and Secs. 5.1 and 5.2) and by restructuring the paper (we moved parts of the math to Sec. A.2 and introduced the new sections on experiments, Sec. 5)
>
> **4. Reasonable Expectation on Resolution of Initial Concerns**
>
> These are the two reviewers from who, unfortunately, we did not get feedback due to the halting of the discussion phase:
>
> - **Reviewer UfsQ (Score: 2):**
>     - **Reason:** This reviewer proposed comparisons to additional Kalman filters, which were provided in Fig. 8 – Sec. A.4.7 – in our response – a similar request was made by reviewer **D2oY**, who seemed satisfied with the way we addressed it. In addition, we fully addressed the other comments, including the validity of our approach for non-Gaussian noises (see new Sec. A.4.8 Fig. 9). Moreover, the reviewer requested experiments on more complex tasks. We addressed this by adding an entirely new task (second paragraph of Sec. 5.2, Fig. 2, and Sec. A.4.5), which reviewer **4btA** – who raised similar concerns – commented on by noting that they “agree that the new experiment adds valuable support for a solid evaluation.” Finally, we further explained the validity and soundness of our proofs – reviewers HNT1 and **4btA** were explicit about a positive evaluation of these aspects. Secs. 3 and A.2 now clarify the convergence proof and optimization landscape. It is worth pointing out that our original manuscript already showed a perfect numerical match (within numerical precision) between gradient descent and our analytical approach (see Fig. 1b). These perfect numerical matches are extremely unlikely to occur under wrongly derived mathematics.
>
> - **Reviewer HNT1 (Score: 6):**
>     - **Reason:** This reviewer indicated in the review similar concerns as reviewer **4btA**, on non-linear dynamics,paper restructuring and more complex tasks, so we expected to have been able to satisfactorily address all reviewer’s concerns as well. More specifically, as requested, we have “moved detailed proofs to the appendix” (Sec. A.2) and expanded the sections devoted to key intuitions (Sec. 3) and empirical evaluations (Sec. 5).

---

> > ### Author Response · Authors · 2025-12-02
> > **Comment for the Area Chair (3)**
> >
> > We note that although the paper has largely changed structure, as recommended by reviewers **HNT1**, **4btA** and **D2oY**, the main results and content have not changed. Please also note that we have updated the line and section numbers provided in our first responses to the reviewers so that the editor, if needed, can more easily check our changes in the finally updated version of the manuscript: some line numbers shifted after we restructured the paper to address the reviewers’ comments.
> >
> > For ease of assessment, in the revised manuscript we have highlighted all modified or newly added sections in blue, so that the changes requested by the reviewers can be quickly identified.
> >
> > We hope this summary helps clarify the status of our submission. We are confident that the new analytical approaches and radically new Model Mismatch framework on control are of both theoretical and practical relevance.
> >
> > Best regards,
> >
> > The Authors

---

### Meta-Review · Area_Chair_xVPL · 2026-01-05

**Summary:**

This paper presents a novel analytical method for solving stochastic optimal control problems characterized by multiplicative and internal process noise. The authors introduce a coordinate-descent algorithm that is theoretically grounded with convergence guarantees and demonstrates a computational speed increase of over three orders of magnitude compared to current gradient-based methods.

The reviewers generally praised the technical depth and efficiency of the algorithm (HNT1, 4btA) but raised concerns regarding the restriction to linear dynamics, the simplicity of the initial experimental tasks, and the clarity of the mathematical presentation.

**Reviewer Concerns:**

Addressed by Rebuttal:
- Computational efficiency: clear speed gains of this algorithm.

- Experimental complexity: all reviewers (HNT1, 4btA, UfsQ, D20Y)  concerned about "toy" tasks, the authors added a 3-DOF redundant arm task with nine muscle-like channels.

- Some technical details: M-Mis does not introduce extra free parameters in inverse optimal control, as the gains are computed analytically from fixed system parameters. Global LQG problem is non-convex even in the fully observable, noise-free setting, so they did not provide a clearer discussion of the optimization landscape in the original manuscript. They do not rely on Gaussian noise assumption.

Outstanding:
- Nonlinear Dynamics: The authors clarified that their framework can be integrated into iLQG/DDP-style loops through local linearization.

- Manuscript Restructuring: Reviewers 4btA and D20Y expressed strong concerns that too much critical content remained in the appendix. While the authors restructured the paper and used the extra page to move experiments to the main text, some mathematical derivations still reside in the appendix due to space constraints.

**Reviewer Scores:**

Reviewer HNT1: 6 --> 6. Expressed support for acceptance based on technical depth. Most concerns regarding task complexity and restructuring were addressed in the authors' response.

Reviewer 4btA: 6 --> 6. Explicitly confirmed that their concerns were addressed and only suggested final restructuring.

Reviewer D2oY: 4 --> 6. Initially skeptical of experimental validation, but authors addressed the specific comparison requests and restructuring.

Reviewer UfsQ: 2 --> 2. Gave the lowest score and concerned on clarity and relatively simple tasks.

---

### Decision · Program_Chairs · 2026-01-26

Reject